# Pan-cancer copy number analysis identifies optimized size thresholds and co-occurrence models for individualized risk stratification

Minh P. Nguyen[1,2,3], William C. Chen [1,2,3] ✉, Kanish Mirchia [1,2,3], Abrar Choudhury[1,2,3], Naomi Zakimi[1,2,3], Vijay Nitturi[4], Tiemo J. Klisch[4], Stephen T. Magill [5], Calixto-Hope G. Lucas [6,7], Akash J. Patel[4] & David R. Raleigh [1,2,3] ✉

Chromosome instability leading to aneuploidy and accumulation of copy number gains or losses is a hallmark of cancer. Copy number alteration (CNA) signatures are increasingly used for cancer risk stratification, but size thresholds for defining CNAs across cancers are variable and the biological and clinical implications of CNA size heterogeneity and co-occurrence are incompletely understood. Here we analyze CNA and clinical data from 691 meningiomas and 10,383 tumors from The Cancer Genome Atlas to develop cancer- and chromosome-specific size-dependent CNA and CNA co-occurrence models to predict tumor control and overall survival. Our results shed light on technical considerations for biomarker development and reveal prognostic CNAs with optimized size thresholds and co-occurrence patterns that refine risk stratification across a diversity of cancer types. These data suggest that consideration of CNA size, focality, number, and co-occurrence can be used to identify biomarkers of aggressive tumor behavior that may be useful for individualized risk stratification.

Chromosome instability and aneuploidy contribute to the genomic complexity of cancer[1] and are implicated in tumorigenesis, progression, metastasis, and resistance to therapy[2–5]. Copy number alteration (CNA) signatures, which are comprised of amplifications or deletions of chromosomes that are markers of aneuploidy and chromosome instability, are increasingly used for clinical risk stratification of diverse cancer types[6,7]. Pan-cancer databases such as The Cancer Genome Atlas (TCGA)[8] have been used to derive prognostic models based on CNAs[7,9]. Despite the importance of CNAs in molecular characterization and risk stratification of cancer, there is no consensus on the optimal size threshold for defining CNAs to predict clinical outcomes. Moreover, copy number analyses in cancer are typically focused on individual CNAs rather than combinations of CNAs, except in cancers such as low-grade glioma and glioblastoma[10], breast cancer[11], colorectal cancer[12], and prostate cancer[13]. Thus, the importance of size-thresholds for defining CNAs and CNA co-occurrence patterns for optimizing prognostic models of cancer outcomes are incompletely understood.

To test the hypothesis that size-dependent CNA models and co-occurrence patterns may improve clinical risk stratification of cancer, we investigated meningioma, a tumor type that is not represented in TCGA datasets but which is associated with recurrent CNAs that can be used for risk stratification[14,15]. Loss of chromosomes 1p, 6q, 14q, 22q, and others distinguish biologically aggressive meningiomas[14,15], but

[1]Department of Pathology, University of California San Francisco, San Francisco, CA, USA. [2]Department of Neurosurgery, University of California San Francisco, San Francisco, CA, USA. [3]Department of Radiation Oncology, University of California San Francisco, San Francisco, CA, USA. [4]Department of Neurosurgery, Baylor College of Medicine, Houston, TX, USA. [5]Department of Neurological Surgery, Northwestern University, Chicago, IL, USA. [6]Department of Pathology, Johns Hopkins University, Baltimore, MD, USA. [7]Department of Neurosurgery, Johns Hopkins University, Baltimore, MD, USA. ✉e-mail: william.chen@ucsf.edu; david.raleigh@ucsf.edu

published meningioma risk stratification models that incorporate CNAs have used inconsistent size thresholds, ranging from 5% to 50% to 80% of individual chromosome arms, and have applied those thresholds uniformly across all chromosomes in the genome[14–20]. To test the broader implications of size-dependent CNA models and co-occurrence patterns, we also investigated 33 cancer types from TCGA[8]. Our results reveal that meningioma CNAs have chromosome- and outcome-dependent optimal size-thresholds that capture focal or broad regions of recurrent deletion.

In this work, we find that canonical meningioma CNAs such as loss of chromosome 1p or 22q may not reliably predict meningioma outcomes on their own, but CNA co-occurrence signatures such as co-deletion of chromosomes 1q and 22q, or increased CNA burden, may serve as reliable biomarkers for meningioma control and overall survival. We identify related patterns of chromosome-specific CNA size-dependence, co-occurrence, and overall burden in 29 of 33 cancer types from TCGA, suggesting widespread applicability of size-dependent CNA models and co-occurrence patterns as biomarkers for individualized risk stratification of cancer.

## Results

### Meningiomas encode size-dependent prognostic CNAs

A summary of the study design and experimental workflow for meningioma investigations that were subsequently applied to TCGA datasets in this project is provided in Fig. 1. To study CNA size-dependence and co-occurrence patterns in meningiomas, CNAs were defined in a previously described integrated cohort of 565 meningiomas with available DNA methylation profiling, RNA sequencing, and long-term clinical outcomes data[16] (Table 1) using iterative size thresholds ranging from 1-99% of each chromosome arm in 1% increments to analyze the prognostic significance of focal versus broad CNAs, both of which are common in meningiomas (Supplementary Fig. 1). The prognostic significance of each CNA using each iterative size threshold was assessed using univariate Cox proportional hazards

models for postoperative local freedom from recurrence (LFFR) or overall survival (OS). The optimal size thresholds for predicting either LFFR or OS were heterogeneous across individual chromosomes, as measured using the area under the curve (AUC) of a time-dependent receiver operating characteristic (ROC) curve to predict 5-year LFFR or OS. These analyses identified loss of chromosomes 1p, 3q, 4p, 4q, 6p, 6q, 9p, 10q, 12q, 14q, 18q, and 22q, and gain of chromosome 1q as "size-dependent" CNAs (Fig. 2a), which were defined as having (1) an AUC of at least 0.60 at any CNA threshold, (2) an AUC standard deviation of at least 0.01 across CNA size thresholds, and (3) prevalence in at least 2.5% of samples when defined using the optimal size threshold (Supplementary Data 1, 2). These criteria were used to maintain sensitivity to any potential prognostic size-dependent CNA. As a comparison, extent of resection, which is a well described prognostic variable for meningioma[21,22], achieved a 5-year LFFR AUC of 0.59 in this cohort. Use of a higher AUC of 0.65 resulted in only 2 prognostic CNAs for LFFR and OS: loss of chromosomes 1p and 6q. Many of the CNAs that were not captured when increasing the AUC cutoff to 0.65 (such as loss of 3q, 4p/q, 6p, 9p, 10q, 12q, 14q, 18q, or 22q, and gain of 1q) are known to be important for identifying aggressive meningiomas[14,15]. Optimal thresholds were minimally impacted by the use of a step size of 1% compared to 5% (median change in % threshold of 3%, interquartile range 1.5–5.5%). Thus, an AUC cutoff of 0.60 and a step size of 1% of each chromosome arm were used for all subsequent analyses.

The implications of CNA size-dependence for meningioma risk stratification were investigated using 2 published models that rely on CNAs to predict postoperative meningioma LFFR. The first, Integrated grade, is based on copy number losses of chromosomes 1p, 3p, 4p/q, 6p/q, 10p/q, 14q, 18p/q, and 19p/q at a uniform threshold of 50% of each chromosome arm, plus *CDKN2A* loss and histological mitotic count[14]. The second, Integrated score, is based on copy number losses of chromosomes 1p, 6q, and 14q at a uniform threshold of 5% of each chromosome arm plus DNA methylation family[23] and World Health Organization (WHO) histological grade[15,20]. Each model was tested on

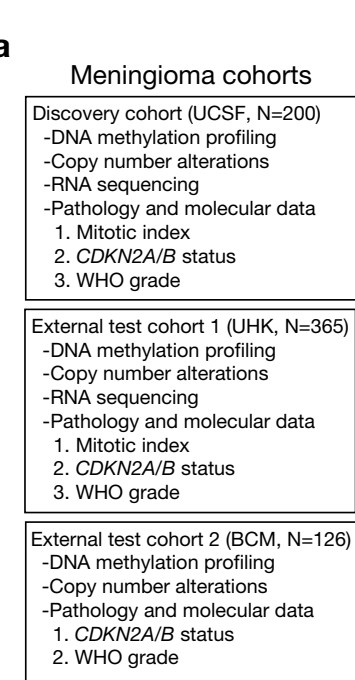

**a**
**Meningioma cohorts**

Discovery cohort (UCSF, N=200)
-DNA methylation profiling
-Copy number alterations
-RNA sequencing
-Pathology and molecular data
  1. Mitotic index
  2. *CDKN2A/B* status
  3. WHO grade

External test cohort 1 (UHK, N=365)
-DNA methylation profiling
-Copy number alterations
-RNA sequencing
-Pathology and molecular data
  1. Mitotic index
  2. *CDKN2A/B* status
  3. WHO grade

External test cohort 2 (BCM, N=126)
-DNA methylation profiling
-Copy number alterations
-Pathology and molecular data
  1. *CDKN2A/B* status
  2. WHO grade

**b**
**Copy number alteration analyses**

Size-dependent CNAs    WHO grade, mitotic index, *CDKN2A/B*

LASSO    Elastic Net    Gradient Boosting

Consensus model

Focal prognostic CNA and gene expression
-2q, 3q, 4p/q, 5p/q, 7p, 8p, 10q, 12q, 13q, 14q
-9p21 (*CDKN2A/B*, IFN gene locus)
-STAT signaling, type I interferon signaling
-*NUAK, UBE2C, CTSC, NEK11, FANCA*

Prognostic CNA co-occurences
-1p/22q co-deletion
-9p/22q co-deletion
-1p/9p/22q triple deletion
-1p/7p/22q triple deletion
-1p/22q co-deletion and 1q gain

**Fig. 1 | Study design and experimental workflow. a** Three meningioma cohorts, comprising three different medical centers, were used for meningioma copy number analyses. The discovery cohort was used for training of all models, and the external test cohorts were used for model validation. **b** Workflow diagram of analyses performed, illustrating the identification of size-dependent CNAs, model building and validation, sliding window binned CNA analyses, and co-occurrence analyses. Source data are provided as a Source Data file.

**Table 1 | Characteristics of meningioma discovery cohort and external testing cohort 1**

| | Integrated cohort, N = 565[a] | Discovery, N = 200[a] | External Testing 1, N = 365[a] |
|---|---|---|---|
| Follow-up (years) | 5.6 (2.0–9.6) | 6.3 (2.5–11.3) | 5.3 (1.9–8.5) |
| Local recurrence at follow-up | 161 (28.5) | 73 (36.5) | 88 (24.1) |
| Local freedom from recurrence, years | 3.8 (1.2–8.0) | 11.8 (7.3—not reached) | Not reached |
| Deceased at follow-up | 137 (24.2) | 59 (29.5) | 78 (21.4) |
| Sex | | | |
| F | 374 (66.2) | 127 (63.5) | 247 (67.7) |
| M | 191 (33.8) | 73 (36.5) | 118 (32.3) |
| Age at surgery | 58 (47–67) | 57 (45–66) | 58 (48–68) |
| Extent of resection | | | |
| Gross total | 394 (70.7) | 118 (59.0) | 276 (77.3) |
| Subtotal | 158 (28.4) | 77 (38.5) | 81 (22.7) |
| Biopsy | 5 (0.9) | 5 (2.5) | 0 (0.0) |
| Unknown | 8 | 0 | 8 |
| Recurrent | 109 (19.3) | 43 (21.5) | 66 (18.1) |
| WHO 2016 grade | | | |
| 1 | 388 (68.7) | 87 (43.5) | 301 (82.5) |
| 2 | 142 (25.1) | 83 (41.5) | 59 (16.2) |
| 3 | 35 (6.2) | 30 (15.0) | 5 (1.4) |
| Adjuvant radiotherapy | 89 (15.8) | 45 (22.5) | 44 (12.1) |
| CDKN2A/B loss | 37 (6.5) | 22 (11.0) | 15 (4.1) |
| Mitoses per 10 HPF | 0.00 (0.00–1.00) | 0.00 (0.00–2.00) | 0.00 (0.00–1.00) |
| DNA methylation group | | | |
| Immune-enriched | 216 (38.2) | 65 (32.5) | 151 (41.4) |
| Merlin-intact | 192 (34.0) | 72 (36.0) | 120 (32.9) |
| Hypermitotic | 157 (27.8) | 63 (31.5) | 94 (25.8) |

[a]n (%); Median (IQR).

**Table 2 | Characteristics of patients and meningiomas in external test cohort 2**

| | N = 126[a] |
|---|---|
| Sex | |
| F | 77 (61) |
| M | 49 (39) |
| Age | 60 (48-68) |
| EOR | |
| GTR | 102 (81) |
| STR | 24 (19) |
| Grade | |
| 1 | 101 (80) |
| 2 | 25 (20) |
| Local recurrence | 28 (22) |
| Median local freedom from recurrence (95 % CI) | 8.38 (7.93—Not reached) |
| Death | 6 (4.8) |
| Median overall survival | Not reached |
| CDKN2A/B Loss | 1 (0.8) |

[a]n (%); Median (IQR).

our integrated cohort of 565 meningiomas (Table 1) using CNA thresholds ranging from 1-99% of each chromosome arm (Fig. 2b). Integrated grade reached a maximum AUC for 5-year LFFR of 0.79 at a CNA threshold of 17% for all chromosomes, and a maximum AUC for OS of 0.77 at a threshold of 29% for all chromosomes. Integrated score reached a maximum AUC of 0.77 for both LFFR and OS at CNA thresholds of 4% and 2%, respectively, for all chromosomes. The performance of each model degraded when varying CNA size thresholds, suggesting that CNA size heterogeneity can influence risk stratification for the most common primary intracranial tumor[24].

**Meningioma size-dependent CNA models identify distinct prognostic CNA signatures**

To further explore the importance of varying CNA size thresholds across chromosomes in meningioma, LASSO and Elastic Net regularized Cox regression models using CNAs defined at chromosome-specific optimal size thresholds to predict LFFR or OS were trained and tested using a discovery cohort (N = 200, University of California, San Francisco), an external test cohort (N = 365, University of Hong Kong) (Table 1), and a secondary external test cohort (N = 126, Baylor College of Medicine) (Table 2). The resulting "size-dependent" CNA models identified several prognostic CNAs including loss of chromosomes 1p, 6q, 7p, 9p, 12q, 18q, and 22q and gain of chromosome 1q, and were well-calibrated to predict LFFR and OS in the test cohort (Supplementary Fig. 2a, b). Of these, chromosome 1q gain, 7p loss, 12q loss, and 22q loss were not included in either Integrated grade or Integrated score (Fig. 2c), suggesting that size-dependent CNA models can identify new CNAs that are associated with clinical outcomes. Consensus models comprised only of size-dependent CNAs appearing in both LASSO and Elastic Net models for meningioma LFFR or OS achieved AUCs for 5-year LFFR and OS of 0.77 and 0.67, respectively, in external test cohort 1, and 0.74 and 0.70, respectively, in external cohort 2. Elastic Net or LASSO Cox models trained on size-dependent CNAs at optimized size thresholds in the absence of histological or molecular features demonstrated maximum 5-year LFFR and OS AUCs of 0.73 and 0.71, respectively, in external test cohort 1 (N = 365). In external test cohort 2 (N = 126), the maximum AUCs for LFFR and OS were 0.76 and 0.84, respectively, using size-dependent CNA models. In comparison, the CNAs in Integrated grade, as defined using the published threshold of 50%, achieved LFFR and OS AUCs in external test cohort 1 of 0.66 and 0.60, respectively, and in external test cohort 2 of 0.63 and 0.48, respectively. The CNAs in Integrated score, as defined using the published threshold of 5%, achieved LFFR and OS AUCs in external test cohort 1 of 0.78 and 0.73, respectively, and in the external test cohort 2 of 0.74 and 0.87, respectively. These data suggest that existing models for clinical risk stratification of meningiomas should be interpreted and defined carefully (and precisely) in the context of CNA size thresholds.

To more comprehensively test the prognostic significance of size-dependent CNAs, we developed models that incorporated size-dependent CNAs with histological and molecular features using similar machine-learning approaches as were used to develop Integrated grade and Integrated score[14,15]. Size-dependent CNAs, WHO grade, CKDN2A/B loss, and histological mitoses per 10 high powered fields were used as inputs in a gradient boosting-based model, resulting in a CNA size-dependent risk score. Gain of chromosome 1q, which contains the USF1 locus[16] and MDM4, loss of chromosome 12p, and loss of the CDKN2A/B locus on chromosome 9p were the most important features selected by these models for LFFR, while loss of chromosome 12p and loss of the CDKN2A/B locus were the most important features for OS (Supplementary Fig. 2b, c). In external test cohort 1, the CNA size-dependent risk score achieved AUCs of 0.83 and 0.78 for LFFR and OS, respectively. External test cohort 2 lacked histological mitotic data, but a version of the CNA size-dependent risk score without mitoses achieved AUCs for LFFR and OS of 0.78 and 0.88, respectively (Supplementary Fig. 2d). In multivariate regression combining the CNA

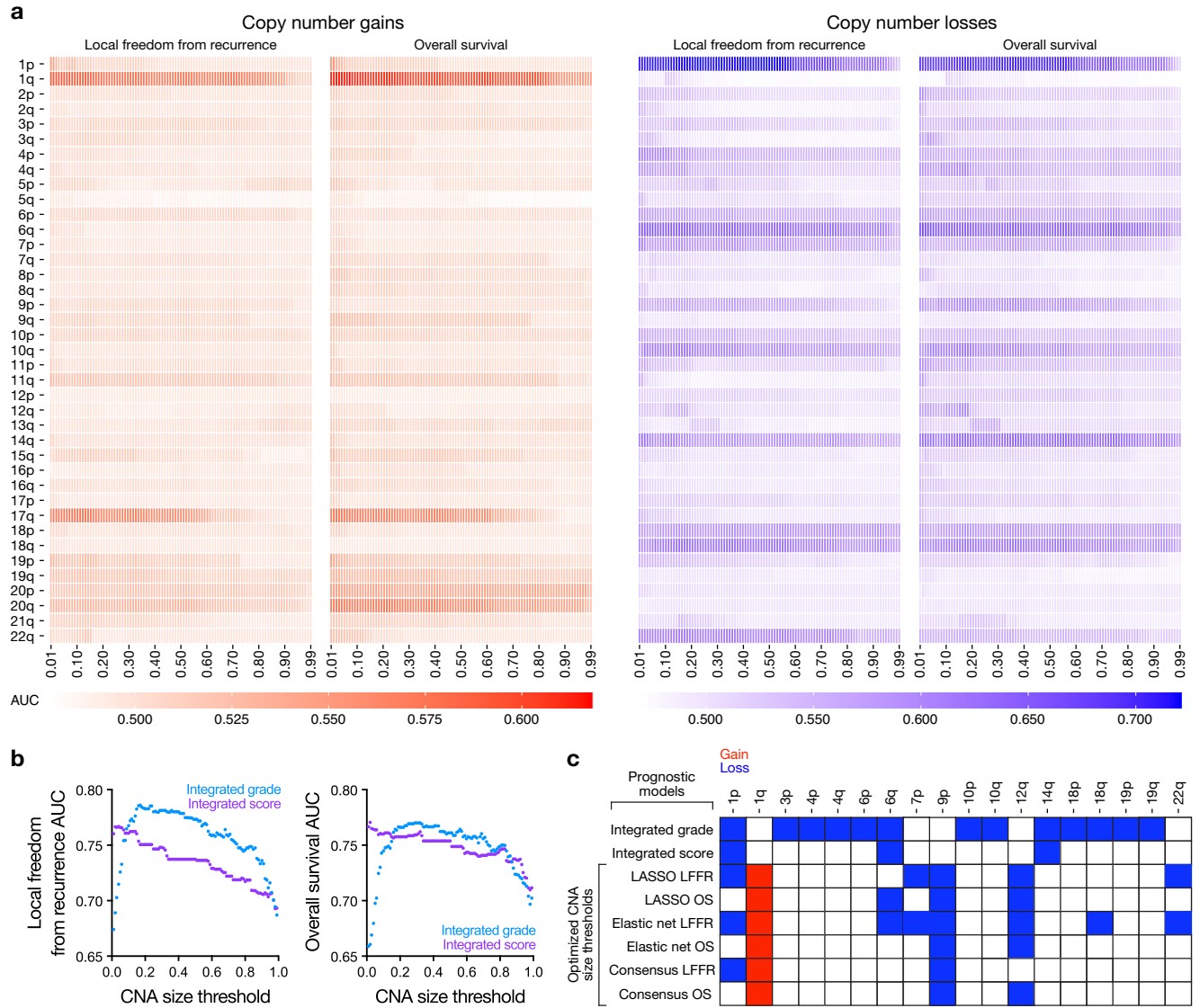

**Fig. 2 | Meningioma risk-stratification models demonstrate chromosome-specific CNA size-dependence. a** Heatmaps showing area under the curve for univariate Cox models of LFFR or OS based on individual copy number gains (left, red) or individual copy number losses (right, blue). Models were trained using sequential size thresholds requiring ≥1% to ≥99% of each chromosome arm required to be gained or lost to define CNAs. $n = 565$ meningiomas. The threshold at which each CNA reached achieved maximum AUC is shown in Supplementary Data 1. **b** Previously published meningioma risk-stratification models incorporating CNAs (Integrated grade, based on histological features and a ≥50% threshold for

defining CNAs, or Integrated score, based on histological features, DNA methylation families, and a ≥5% threshold for defining CNAs). $n = 565$ meningiomas. **c** CNAs from previously published meningioma risk-stratification models incorporating CNAs, or from newly-derived chromosome-specific CNA size-dependent LASSO or Elastic Net models for meningioma LFFR or OS, or from newly-derived consensus models comprised only of size-dependent CNAs appearing in both LASSO and Elastic Net models for meningioma LFFR or OS. $n = 565$ meningiomas. Source data are provided as a Source Data file.

size-dependent risk score with patient sex, extent of resection (defined as gross total or subtotal based on postoperative magnetic resonance imaging), newly diagnosed versus recurrent presentation, WHO histological grade[25], and adjuvant radiotherapy, the CNA size-dependent risk score remained prognostic ($P < 0.0001$) for both LFFR and OS in both external test cohorts (Supplementary Fig. 3, Supplementary Data 3). In external test cohort 1, likelihood ratio tests demonstrated that the addition of the CNA size-dependent model to Integrated grade in a Cox proportional hazards model significantly improved prediction of LFFR ($P = 0.0055$), but reciprocal addition of Integrated grade to the CNA size-dependent model did not improve prediction of LFFR ($P = 0.32$). Addition of the CNA size-dependent model to Integrated score also improved prediction of LFFR ($P = 0.00088$), and the reciprocal was also true, albeit with a more modest $P$ value ($P = 0.013$). This same pattern was also observed for prediction of OS, where adding the

CNA size-dependent model significantly improved performance of Integrated grade ($P = 7.01 \times 10^{-5}$), but the reciprocal was not true ($P = 0.18$), and adding the CNA size-dependent model significantly improved performance of Integrated score ($P = 5.0 \times 10^{-6}$), and the reciprocal was also true, albeit with a more modest $P$ value ($P = 0.014$). Size-dependent CNA models additionally provided further risk stratification within the low- and intermediate-risk groups and some high-risk groups identified by WHO grade, Integrated grade, and Integrated score (Supplementary Fig. 4a–c). When incorporated with Integrated grade or Integrated score in multivariate regressions, the CNA size-dependent model remained independently prognostic in both external test cohorts (Supplementary Data 3). In addition, incorporation of DNA methylation groups in a multivariate Cox analysis showed that the CNA size-dependent model remained independently prognostic for LFFR ($P = 0.000136$) and OS ($P = 7.24e-08$).

## Meningiomas encode prognostic focal CNAs

One interpretation of these data is that the size-dependence of some CNAs in prognostic models may reflect biologically significant focal CNAs that are not captured using larger thresholds. Thus, to explore the prognostic significance of focal CNAs, a sliding window of 200 Kb was used to examine the 5-year LFFR AUC of focal copy number gains and losses across chromosomes in the integrated cohort (Fig. 3a, Table 1). A bin size of 200 Kb was chosen as it approximates the size of the *HLA* locus, which is focally gained or lost on chromosome 6p in different DNA methylation groups of meningiomas[16]. Focal prognostic CNAs were defined by AUC of at least 0.04 higher than the median AUC for CNAs affecting the entire chromosome arm. This threshold was chosen based on the magnitude of the AUC peak corresponding to loss of the *CDKN2A/B* locus on chromosome 9p, a molecular feature that is sufficient for diagnosis of anaplastic meningioma, WHO grade 3[10]. Several chromosomes harbored focal regions of recurrent copy number deletion with prognostic value according to these "focal criteria", including chromosomes 2q, 3q, 4p/q, 5p/q, 7p, 8p, 10q, 12q, 13q, and 14q (Fig. 3a). Copy number deletions of chromosomes 1p and 22q, which are known prognostic markers in meningioma, did not demonstrate clear focal prognostic regions, although loss of the distal end of chromosome 1p had a higher AUC than loss of the proximal segment (Supplementary Fig. 5a). Interestingly, regions with focally recurrent copy number deletions did not consistently align—and were sometimes negatively associated—with regions of focal prognostic value, such as those on chromosomes 8p, 10q, and 18q. This suggests that random copy number changes alone may not worsen clinical outcomes, and that focal prognostic losses could impact chromosomal regions critical to meningioma biology.

Genes mapping to focal prognostic regions on chromosomes 9p and 14q had concordantly decreased expression by matched RNA sequencing of the same samples, while gene expression on the remaining selected chromosome arms had low levels of expression at baseline (Fig. 3b, Supplementary Data 4, 5). Ontology analysis of genes contained in these focal prognostic CNAs revealed STAT signaling, response to exogenous nucleic acids, and type I interferon-mediated signaling (Fig. 3c). Manual inspection of the genes contained in focal prognostic regions suggested that these ontologies were primarily driven by interferon-related genes on chromosome 9p (Supplementary Data 5). Visualization of regions of focally recurrent loss on chromosomes 9p and 14q stratified by DNA methylation groups revealed that the focal regions of loss on 14q were most enriched compared to broad losses in Immune-enriched meningiomas (Supplementary Fig. 5b).

LASSO Cox regression of the expression of genes located on recurrent focally lost CNAs identified several as being prognostic for LFFR, such as *NUAK2, UBE2C, CTSC, NEK11, and HSPB7* (AUC 0.85 for 5-year LFFR in external test cohort 1), and several as being prognostic for OS, such as *FANCA, UBE2C, CACHD1*, and *SYNPO2* (AUC 0.76 for 5-year OS in external test cohort 1) (Supplementary Fig. 5c). Narrowing this analysis to genes located only in regions with focal prognostic value as determined by focal increases in AUC across 200 Kb bins identified *NUAK2* as important for LFFR (AUC 0.61), and *NUAK2, DSP, and PRKAR2B* as most important for OS (AUC 0.72) (Supplementary Fig. 5d), suggesting that these genes may drive the prognostic value of their respective CNAs in meningioma.

To address the potential impact of greater degrees of copy number amplification or deletion on meningioma outcomes, such as trisomy or homozygous loss, we generated sample-wise CNA segment plots by chromosome position plotted against the amplitude of loss or gain as measured by segment intensity. Other than identifying the same regions of focal loss or gains as our sliding window approach (Fig. 3a), this analysis did not reveal clear regions of decreased or increased intensity to suggest focal deep deletion or high level amplification, except for the *CDKN2A/B* locus on chromosome 9p,

which is expected to be homozygously deleted in some samples[18,26], and the *HLA* locus on chromosome 6p, which is expected to be homozygously deleted or amplified in some samples[16]. Plotting segment mean intensities for the chromosomes most associated with copy number gains, 1q and 17q, revealed no clear patterns of focal amplification, suggesting that a uniform intensity threshold was sufficient to capture prognostic CNA gains. As an additional sensitivity analysis, we examined whether varying the intensity threshold by 0.05 increments significantly altered the prognostic value of each chromosome and found no optimal intensity threshold for defining any chromosome-specific CNA. Additional descriptions of these analyses can be found in the "Methods" section.

## Meningiomas encode prognostic co-occurrent CNAs

Size-dependent CNAs tended to co-occur in individual meningiomas (Supplementary Fig. 6a). Thus, to test the hypothesis that CNA co-occurrence patterns could be used to refine meningioma risk stratification, LASSO regularized Cox regression models using co-occurrent CNA pairs were developed using the discovery cohort (Table 1). These models identified chromosome 1p/22q co-deletion, 9p/18q co-deletion, and 1q amplification with 22q deletion as prognostic for LFFR (Supplementary Fig. 6b). Each of these CNA pairs remained prognostic when defining CNAs at different size thresholds, including the optimal size thresholds identified in Supplementary Data 1, a threshold of 5% uniformly applied to each chromosome arm, or a threshold of 80% uniformly applied to each chromosome arm, using samples in external test cohort 1 (Fig. 4a). These findings remained significant when accounting for the total number of CNAs per meningioma as defined using the optimal threshold for each chromosome ("CNA burden") on multivariate modeling (Supplementary Data 6). Repeating this analysis with co-occurrent triplets identified chromosome 1p/22q/9p deletion, 1p/22q/7p deletion, and 1p/22q co-deletion with 1q gain as prognostic for LFFR, and 1p/9p/14q deletion as prognostic for OS (Supplementary Fig. 6c). These CNA triplets were associated with worse clinical outcomes than any of co-occurring CNA pairs or individual CNAs (Supplementary Fig. 6d, Supplementary Data 6).

## Clinically aggressive meningiomas are associated with increasing CNA burden

Loss of chromosome 22q is a common early alteration in meningioma[27], but the prognostic significance of this CNA is limited as subsequent genomic alterations lead to divergent meningioma phenotypes, such as immune infiltration or cell cycle misactivation[16]. Thus, we hypothesized that progressive accumulation of CNAs beyond loss of chromosome 22q loss could coincide with more aggressive meningioma behavior. In support of this hypothesis, hierarchical clustering of meningiomas binned by CNA burden using optimized size-thresholds revealed 3 groups (Fig. 4b, Supplementary Fig. 7a–d). Cluster 1 CNAs, such as loss of chromosomes 1p, 14q, and 22q, were prevalent regardless of total CNA burden. Cluster 3 CNAs, such as loss of chromosome 9p or gain of chromosome 1q, were enriched in meningiomas with higher total CNA burden. Cluster 2 CNAs were uncommon and did not correlate with total CNA burden. Increased CNA burden was associated with a higher proportion of WHO grade 2 and 3 meningiomas, and meningioma CNA burden was associated with worse clinical outcomes (Fig. 4c). Thus, CNA burden as a marker of aneuploidy, which has been demonstrated to be important in other cancers[28], and enrichment of key CNAs may be a useful biomarker of high-risk meningiomas.

## Pan-cancer analyses identify size-dependent CNAs across 29 cancer types

To test the broader implications of CNA size thresholds and co-occurrence patterns for cancer risk stratification, SNP array-derived CNA profiles and clinical outcome data were obtained for 10,383

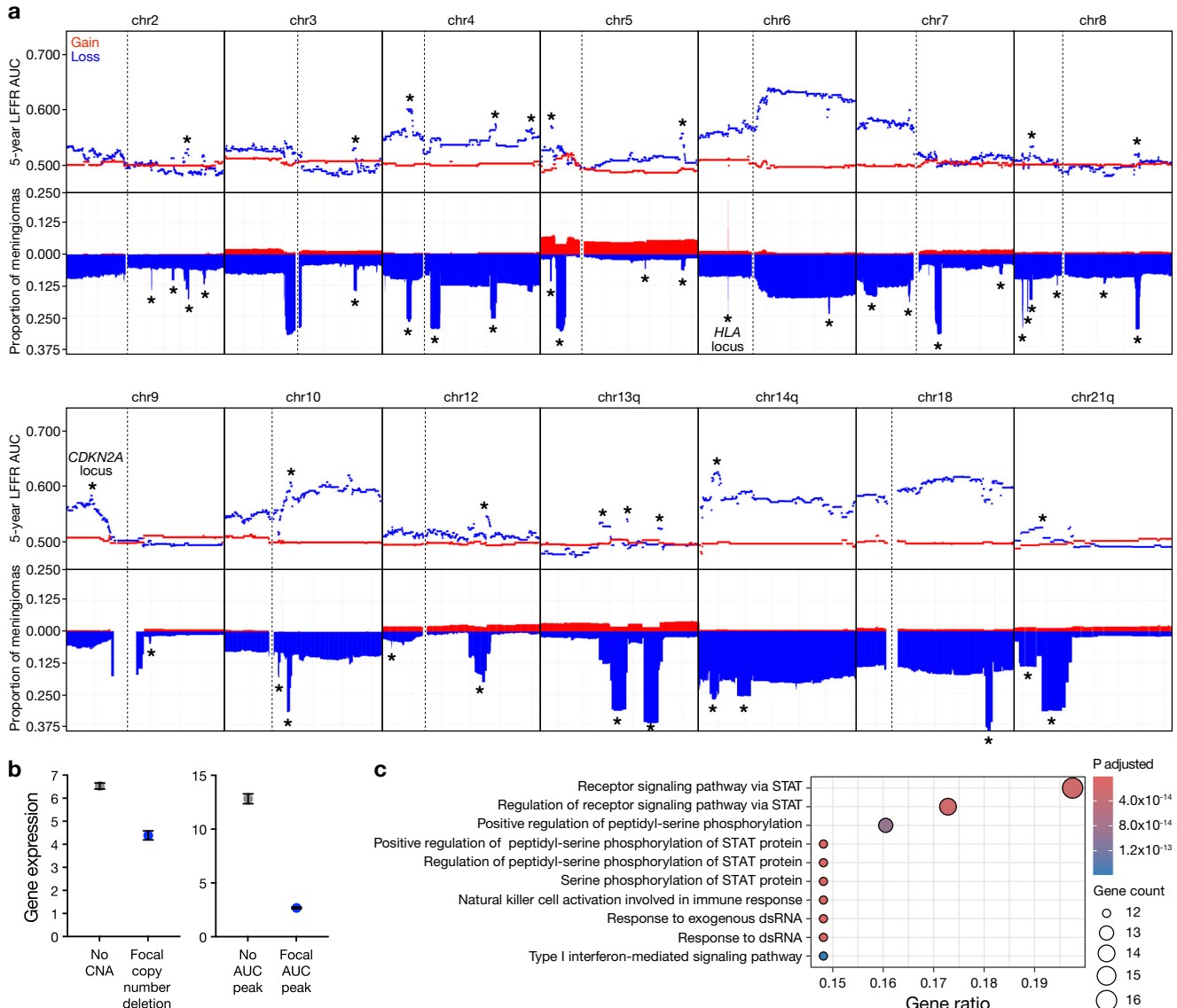

**Fig. 3 | Meningiomas encode prognostic, focal copy number alterations. a** CNA profile plots (bottom rows) demonstrating focal copy number losses (blue) or gains (red) in meningioma, and 5-year LFFR AUC plots (top rows) demonstrating prognostic significance associated with focal regions of loss (blue) or gain (red). All chromosomes with peaks on either CNA profile plots or AUC plots are shown included. Dashed lines show centromeres. The x-axis of each plot shows the entire length of the corresponding chromosome or chromosome arm. Asterisks denote focal copy number and prognostic regions analyzed in Fig. 3b and Supplementary Data 4 and 5. *n* = 565 meningiomas. **b** Sample-matched RNA sequencing expression (log$_2$ transcripts per million) of all genes mapping to focal regions of recurrent loss on CNA profile plots from Supplementary Data 4 (left) or AUC plots from Supplementary Data 5 (right). Genes were identified by cross-referencing positions along the chromosome with the Ensembl database. Both plots are shown as peaks on CNA prolife plots did not always correspond to peaks on AUC plots, and vice versa. Dots and error bars show mean and standard error of the mean, respectively. *P* values (<0.0001 for each plot) are from two-sided Wilcoxon rank-sum tests comparing expression in meningiomas with deletion of each region (blue) compared to samples with no deletions anywhere in the cognate chromosome arm (gray). *n* = 502 meningiomas. **c** Dotplot of gene ontology analysis using genes contained in focal regions of AUC z-score plots with a univariate Cox AUC at least 0.04 higher than the median for their respective chromosome arms. Source data are provided as a Source Data file.

tumors in TCGA datasets that encompassed 33 cancer types[8]. The majority of cancers analyzed demonstrate prognostic CNA size-dependence using the same criteria as for meningioma: (1) an AUC of at least 0.60 at any CNA threshold when using univariate Cox proportional hazards models to predict progression-free survival (PFS) or OS, (2) an AUC standard deviation of at least 0.01 across CNA size thresholds, and (3) prevalence in at least 2.5% of samples when defined using optimal size thresholds for each chromosomes (Fig. 5a, Supplementary Data 7, Supplementary Data 8). There were no clear differences in overall CNA burden as measured by the aneuploidy score[9] or using CNA heterogeneity[9] between CNA size-dependent and CNA

size-independent TCGA cancers (Supplementary Data 7). Esophageal carcinoma (ESCA) had 31 size-dependent chromosome arms, the highest among all cancer types. Bladder urothelial carcinoma (BLCA), head and neck squamous cell carcinoma (HNSC), acute myeloid leukemia (LAML), lung squamous cell carcinoma (LUSC), and thyroid carcinoma (THCA) had no identifiable size-dependent prognostic CNAs, indicating that uniform size thresholds could likely be applied with minimal loss in sensitivity in these cancer types. More broadly, heterogeneity of size-dependent CNAs across cancer types suggests that a disease-specific approach may be necessary and beneficial to optimize the use of CNAs as biomarkers for clinical outcomes.

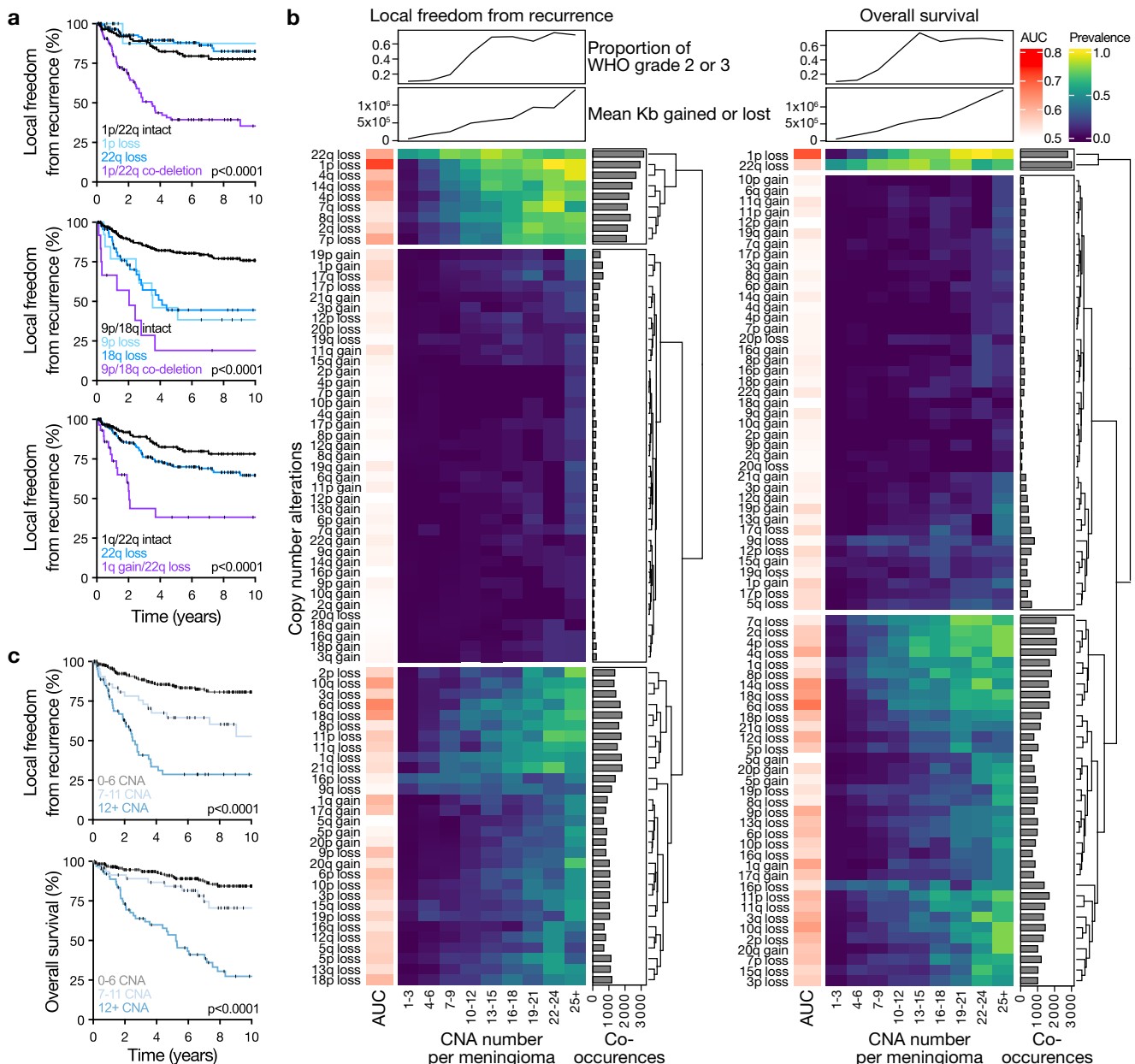

**Fig. 4 | Size-dependent CNA co-occurrence patterns and total CNA burden predict meningioma outcomes. a** Kaplan-Meier curves comparing meningioma LFFR according to individual CNAs versus co-occurrent CNA pairs identified as the most important predictors of clinical outcomes in LASSO Cox models from Supplementary Fig. 6b using chromosome-specific CNA size thresholds. Log-rank tests. *n* = 365 meningiomas. **b** Heatmap showing unsupervised hierarchical clustering of individual CNAs according to the total number of CNAs per meningioma. CNAs were defined using optimal size thresholds for LFFR or OS. Bar plots on the right side of each heatmap measure the total number of co-occurrent pairs including each of the CNAs, across all samples containing that CNA. **c**, Kaplan-Meier curves showing meningioma LFFR or OS according to CNA burden. Log-rank tests. *n* = 565 meningiomas. Source data are provided as a Source Data file.

As in meningioma, co-occurrent pairs of size-dependent CNAs were found in many TCGA cancers and were associated with significantly worse outcomes even when accounting for CNA burden (Fig. 5b, Supplementary Data 9). Unlike meningioma, several TCGA cancers had size-dependent co-occurrent CNA pairs that were associated with better outcomes, including ACC, BLCA, HNSC, LAML, LUSC, and THCA (Supplementary Data 9). As the cancer type with the most available samples, breast cancer (BRCA) was selected for co-occurring CNA triplet analysis, which revealed chromosome 10q/18p/18q co-deletion and 11p/18q/9p co-deletion as significantly prognostic for PFS and OS, respectively (Supplementary Fig. 8a, b, Supplementary Data 9).

**Cancer subtypes demonstrate distinct size-dependent CNAs and CNA co-occurrence signatures**

To examine whether patterns of CNA size-dependence and co-occurrence were consistent across subtypes of TCGA cancers, we selected BRCA and cervical squamous cell carcinoma and endocervical adenocarcinoma (CESC), two TCGA cancer types for which molecularly distinct subtypes could be discerned from available histological data (a full list of identifiable subtypes is available in Supplementary Data 10). CNA size-dependence and co-occurrence analyses (performed using LASSO regularized Cox regression with 10-fold cross-validation across the entirety of samples within each subtype to predict PFS or OS, as described in the "Methods" section) revealed further

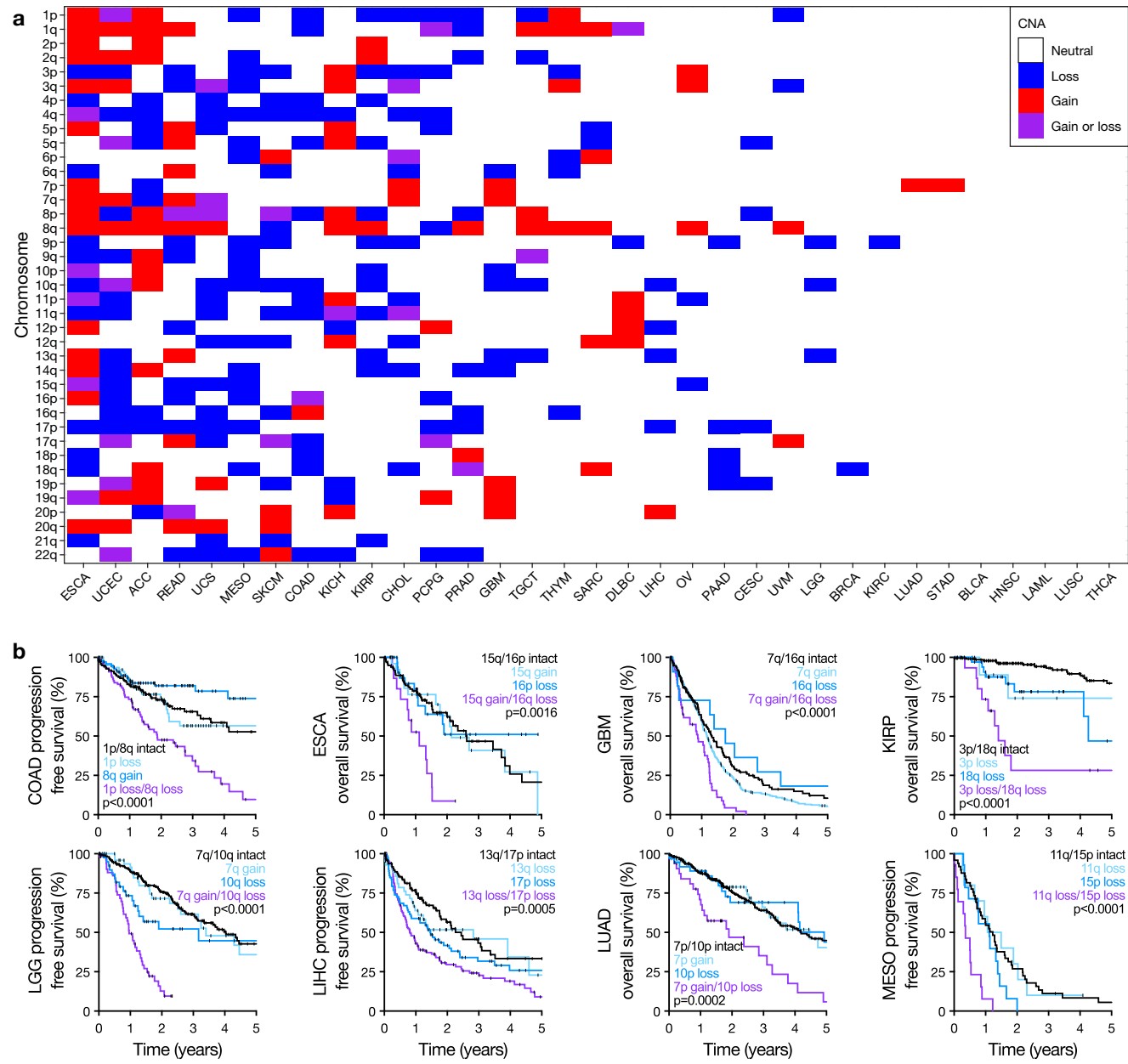

**Fig. 5 | Pan-cancer analyses reveal chromosome-specific size-dependent CNA co-occurrence risk-stratification patterns. a** Heatmap showing chromosome-specific size-dependent CNAs across TCGA cancer types. Size-dependent CNAs were defined as having (1) a univariate Cox AUC for either PFS or OS of at least 0.60, (2) a standard deviation for AUC across CNA size thresholds of at least 0.01, and (3) presence in at least 2.5% of samples for a given TCGA cancer type. A full list of optimal size thresholds and AUCs are provided in Supplementary Data 8. **b** Selected Kaplan-Meier curves showing prognostic co-occurrent CNA pairs for PFS or OS identified using LASSO Cox models of size-dependent CNAs for TGCA cancer types. Log-rank tests. *n* = 426 COAD, 182 ESCA, 571 GBM, 283 KIRP, 510 LGG, 367 LIHC, 500 LUAD, and 87 MESO. Source data are provided as a Source Data file.

heterogeneity in size-dependent CNAs and prognostic co-occurrence patterns within histological subgroups of BRCA and CESC. Infiltrating lobular breast carcinoma had a higher number of size-dependent CNAs than infiltrating ductal carcinoma (Supplementary Fig. 9a), and CESC adenocarcinomas had a higher number of size-dependent CNAs than CESC squamous cell carcinomas (Supplementary Fig. 9b). Moreover, prognostic size-dependent CNA co-occurrent pairs were distinct between histological subtypes of BRCA and CESC (Supplementary Fig. 9c, d). Thus, use of CNA size thresholds and co-occurrence patterns as prognostic markers may need to be refined with respect to histological subtypes of particular cancers, especially for those known to be associated with distinct molecular profiles.

Additional analyses were performed on select TCGA cancers with previously described prognostic copy number co-occurrence patterns. Although low-grade glioma (LGG) can be stratified by co-deletion of chromosomes 1p and 19q to identify less aggressive oligodendrogliomas[10], neither of these CNAs met size-dependent criteria and this pair was excluded from co-occurrence models. Nevertheless, chromosome 1p/19q co-deletion was associated with better clinical outcomes, as expected (Supplementary Fig. 10a). Although gain of chromosome 7 and loss of chromosome 10 are common in glioblastoma (GBM)[10], this co-deletion was not associated with any difference in clinical outcomes than either of these CNAs alone (Supplementary Fig. 10b), suggesting that widespread CNAs or CNA

co-occurrent patterns may not necessarily correlate with prognostic value. The presence of other co-occurrent pairs in GBM (such as chromosome 7p loss and chromosome 16q gain) were rare, which limited prognostic analyses. In sum, these data reinforce the interpretation that a cancer-specific approach may be necessary and beneficial to optimize the use of CNAs as biomarkers for clinical outcomes.

## Discussion

Here we show that the performance of prognostic models for meningioma outcomes that incorporate CNAs are sensitive to variation in CNA size, focality, number, and co-occurrence, and that use of optimized size thresholds can identify additional prognostic CNAs, improve prognostic models, and identify focal CNAs and CNA co-occurrence patterns with prognostic value. Extending these findings to TCGA samples, we demonstrate that the majority of human cancers encode prognostic size-dependent CNAs and co-occurrence patterns.

Recent meningioma molecular classification systems have incorporated CNAs to enhance clinical risk stratification. These include Integrated grade, which is based on copy number losses of chromosomes 1p, 3p, 4p/q, 6p/q, 10p/q, 14q, 18p/q, and 19p/q, defined at a uniform threshold of 50% of each chromosome arm plus *CDKN2A* loss and mitotic count from histology[14]. The second, Integrated score, is based on copy number losses of chromosomes 1p, 6q, and 14q, defined at a uniform threshold of 5% of each chromosome arm plus DNA methylation family[23] and WHO histological grade[15]. Our finding that each of these systems achieved optimal performance using CNA size thresholds close to the values used for training, and that performance degraded with variation of size thresholds, underscores the sensitivity of CNA models to size thresholds. At the level of model training, we find that size thresholds determine which CNAs form model inputs. Thus, the selection of CNA size thresholds represents a crucial branch point that influences the ultimate characteristics of resulting prognostic models. While Integrated grade and Integrated score incorporate overlapping CNAs, the optimal CNA size threshold and performance across thresholds varies significantly between these systems. These models each use uniform CNA size thresholds across chromosomes, but our results indicate that the optimal CNA size threshold can vary from chromosome to chromosome. By allowing for chromosome-specific variation in CNA size thresholds, we show that additional prognostic CNAs can be identified, and that a size-dependent CNA model may provide additional prognostic information in comparison to, or in combination with, Integrated grade and Integrated score.

These findings emphasize the importance of context when interpreting prognostic CNAs in meningiomas with respect to size, location, and co-occurrence with other CNAs. For example, interpretation of 10% loss of chromosome 1p in a proximal location, in the absence of any other CNAs, differs dramatically from 30% loss of distal chromosome 1p, co-occurring with loss of chromosomes 22q, 9p, or 14q, or co-occurring with gain of chromosomes 1q. Our data suggest the optimal prognostic threshold for loss of chromosome 1p is ~20%. One possible explanation for this threshold is that smaller losses tended to occur sporadically throughout chromosome 1p in our cohorts, whereas broader losses preferentially impacted the distal chromosome arm. Concordantly, our analysis of focal prognostic CNAs showed that AUC peaked in the distal region of chromosome 1p. Indeed, loss of distal chromosome 1p has long been shown to be among the most prognostic CNAs in meningioma[29]. More broadly, the optimal size thresholds for defining chromosome-specific CNAs ranged from 1 to 32% of individual chromosome arms, suggesting that for some chromosomes, such as loss of 1p (optimal LFFR threshold 23%) or loss of 12q (optimal LFFR threshold 18%), smaller size thresholds may not be ideal for identifying meningiomas with poor clinical outcomes. These thresholds may reflect the cutoff needed to separate broader amplification or deletion from sporadic CNAs that are not associated with a worse

clinical prognosis. However, recent work has demonstrated that loss of as little as 6% of chromosome 1p is prognostic in meningioma, and that even loss of 1% could be useful in predicting risk of recurrence after resection[20], suggesting that any deletion of a critical chromosome arm may be sufficient to affect clinical outcomes.

Our CNA size-dependent models identified multiple prognostic CNAs that were not included in Integrated grade or Integrated score. While some of the CNAs identified by our models, such as loss of chromosome 12p, displayed marginal individual prognostic power, the same can also be said of CNAs included in existing models (such as loss of chromosomes 3p or 19p/q in Integrated grade). Despite strict criteria for including CNAs in models to limit skewed prediction of LFFR or OS, the models we developed and tested still identified different CNAs as important for either outcome. It is difficult to understand how a CNA may be important for LFFR or OS, but not for LFFR and OS, other than through speculation about genes that may be affected and their importance in response to salvage therapy after local recurrence. When comparing LASSO to Elastic Net regularization models, Elastic Net tends to identify more predictors as it places less emphasis on identifying a sparse model, which may have contributed to some incongruity between models.

Thus, incorporation of CNAs with low individual prognostic value into integrated prognostic models may suggest the importance of looking beyond the size and location of chromosome-specific CNAs. In support of this hypothesis, our findings indicate that CNA co-occurrence patterns also matter. For example, presence of chromosome 1p loss alone did not portend a worse prognosis for meningioma, but co-occurrence of chromosome 1p loss with chromosome 22q loss was associated with substantially worse clinical outcomes. Interestingly, our analyses identified chromosome 1p/22q co-deletion, 1q gain/22q deletion, and 9p/18q co-deletion as prognostic co-occurrent CNA pairs in meningioma. All three of these prognostic pairs remained significant predictors of LFFR even when accounting for CNA burden, suggesting that the prognostic significance of these pairs is not simply a marker of aneuploidy. Further investigation of CNA pairs or triplets may improve understanding of the biological importance of meningioma copy number profiles for clinical risk stratification.

When examined more globally, we identified clusters of co-occurrent CNAs that were prevalent at all levels of CNA burden (chromosomes 1p, 14q, 4p), while some prognostic CNAs (chromosome 1q gain, 9p loss, 18q loss) became more common only at higher levels of meningioma aneuploidy. The presence of chromosome 22q loss may be a marker for meningiomas with loss of *NF2*/Merlin expression[16]. Some *NF2*-inactivated meningiomas may also be marked by accumulation of CNAs, and the acquisition of co-occurrent CNAs involving chromosomes 1p, 1q, 9p, or 18q may be particularly prognostic in this context. Notably, chromosome 1q gain has been shown to enhance oncogenesis via overexpression of *USF1*[16] or *MDM4*[30], which may phenocopy *TP53* loss and promote tolerance of DNA damage resulting from chromosomal instability. Chromosome 9p contains the *CDKN2A/B* locus, which is a key tumor suppressor of the RB pathway, and even heterozygous loss of the *CDKN2A/B* locus has been shown to portend a poor prognosis in meningioma[18]. Interestingly, our analyses of gene expression data mapping to focally prognostic CNAs identified an enrichment of the STAT pathway and type I interferon signaling, possibly driven by a cluster of interferon related genes contained on chromosome 9p that are recurrently lost along with the *CDKN2A/B* locus in meningioma. Suppression of type I interferon signaling may represent a mechanism of immune evasion in the setting of chromosomal instability[31], which has been shown to induce a cGas-STING mediated type I interferon response. Indeed, loss of the chromosome 9p interferon gene cluster correlates with reduced T cell infiltrate and poorer prognosis in melanoma and HPV-negative head and neck cancer[32,33]. In meningioma, we identified focal prognostic regions on chromosomes 2q, 3q, 5p, 5q, 8p, 9p, 10q, 12q, 13q, and 14q, including a

focal peak at the *CDKN2A/B* and interferon gene locus on 9p. Of these, only 3q, 9p, 12q, and 14q were identified as prognostic and size-dependent when using percent-arm-altered thresholds rather than a sliding window approach, suggesting that a more granular approach to defining CNAs could identify additional focal gains and losses of prognostic significance. However, a granular approach such as this may be too complex to be implemented and routinely used in clinical practice as compared to a percent-arm-altered threshold.

Higher grade meningiomas are known to encode higher CNA burden and have worse clinical outcomes compared to WHO grade 1 meningiomas. To account for the effect of WHO grade and to improve the clinical relevance and comparability of our CNA size-dependent and co-occurrence models to previously published prognostic models, we combined CNA size-dependent and co-occurrence models with WHO grade, mitoses per 10 high-power fields, and *CDKN2A/B* loss (the additional variables that were included alongside CNAs during the development of Integrated grade). Adjuvant radiotherapy was a significant predictor of LFFR but not OS in these models. Currently, the criteria to prescribe adjuvant radiotherapy for intermediate risk meningiomas are controversial, and models that can better stratify these patients are of considerable value[34,35]. The models described here, along with other prominent meningioma risk stratification systems[14,15,36], were designed with a focus on guiding postoperative therapy and management based on data obtained from resected or biopsied tissue. As radiotherapy is the only standard treatment for meningioma beyond surgery, it is perhaps unsurprising that molecular features could be used to guide postoperative management of meningioma. Nevertheless, the aim of this work was not to propose a new meningioma risk stratification model that uses different size-thresholds for each chromosome or to propose a universally applicable CNA threshold across chromosome arms across cancer types. Rather, our objective was to understand if consideration of CNA size thresholds and co-occurrence patterns could be used to identify new predictors of clinical behavior. The use of external cohorts served to validate our initial findings regarding CNA size-dependence and co-occurrence, but further investigation testing these models in modern series of meningiomas that better model contemporary clinical practice will be necessary before adoption into clinical practice. Moreover, the exploration of these phenomena in other cancers, as suggested by our analyses of TCGA samples, presents a foundation to validate our results and perhaps translate these findings to clinical practice.

To our knowledge, ambiguity of optimal size thresholds for defining chromosome-specific CNAs is not a meningioma-specific problem and appears to be a problem for the majority of cancers. Among the size-dependent cancers we identified from TCGA datasets, several have been reported in pan-cancer analyses to have CNA patterns of prognostic significance, including KIRP, STAD, LIHC, OV, and SKCM[37]. For others such as COAD, OV, BRCA, HNSC, PRAD, CESC, and NLSC, aneuploidy is associated with worse clinical outcomes[38]. Here we demonstrate that all but 4 of the 33 cancer types in TCGA have prognostic CNAs that are sensitive to variation in size threshold. As we show in meningioma, consideration of chromosome-specific thresholds across these 29 cancers identifies prognostic CNAs distinct from those reported in the literature. Moreover, prognostic CNA co-occurrence patterns are rarely reported in the literature, but several have been identified, including chromosome 1p/19q co-deletion in LGG[10], 1q gain and 16q loss in BRCA[11], 7 gain and 10 loss in GBM[10], 18q loss and 8q gain in COAD[12], and 8p loss and 8q gain in prostate cancer[13]. We also identify prognostic CNA co-occurrence patterns across most TCGA cancers, and by restricting co-occurrence model inputs to size-dependent CNAs, we identify pairs that differ from the few that have been previously reported. While each co-occurrent CNA pair remained significantly prognostic when combined in multivariate analyses with CNA burden, CNA burden was still prognostic for many cancers with size-dependent CNAs, in accordance with prior pan-cancer investigations[28]. Thus, as with meningioma, consideration of size thresholds for defining chromosome-specific CNAs identified prognostic co-occurrence patterns for most cancers.

The findings of this study should be interpreted in the context of its limitations. Clinical data for all meningioma cohorts and TCGA samples were obtained retrospectively, and all analyses derived from such data are subject to biases that are inherent to retrospective research. We assess copy number changes in meningioma using DNA methylation profiling, as opposed to SNP array or whole-exome sequencing, and probe-based approaches are inherently limited by probe density and distribution compared to whole genome sequencing. However, CNAs identified using DNA methylation profiling have previously been demonstrated to have 99.12% concordance with those identified through exome sequencing approaches across multiple platforms[16]. Meningioma samples tended to have either very small or large copy number changes, which may have limited investigation of the prognostic value of intermediately sized copy number changes. Thus, the choice of bin size to identify prognostic focal CNAs in meningioma (200 Kb) may limit the accuracy and granularity with which regions of importance were identified. This size was chosen as it approximates the size of the *HLA* locus, an important focal CNA in meningioma[16]. Many meningiomas exhibited either broad arm-level loss or multiple noncontiguous regions of deletion, making it difficult to examine the prognostic importance of focal CNAs in isolation. While manual inspection of focal CNAs is the preferred method of identifying deletion of the *CDKN2A/B* locus[39,40], this was previously only done in the context of identifying a single prognostic marker, rather than a generalized approach for identifying focal CNAs of prognostic significance across chromosomes. For the sake of consistent analyses, the time-dependent AUC for clinical outcomes was calculated over 5 years, and focused investigation of cancer types other than meningioma may benefit from a more tailored selection of endpoints. Subgroup analysis of TCGA samples was limited due to lack of molecular data. We used histological data to assign samples to distinct subgroups, which in the cases of BRCA and CESC captured differences in size-dependent CNAs and co-occurrence patterns. Despite being present in at least 2.5% of the cohort, the co-occurrent CNA pairs that were important for OS in meningioma were rare overall and risk stratification became difficult, especially at higher thresholds. Thus, it is possible that rarer CNA pairs occurred stochastically in patients with shorter OS, leading models to overestimate their importance. Moreover, the low rate of death in our cohorts and in meningioma cohorts in general represents a challenge to analyses involving so many possible predictors without a higher event rate.

In sum, our results demonstrate that chromosome-specific CNAs exhibit size-dependence with respect to their prognostic value across most cancer types. We find cancer risk stratification systems using CNAs with chromosome-specific size thresholds and co-occurrence patterns may refine risk stratification across a diversity of human cancers.

## Methods

### Inclusion and ethics

This study complied with all relevant ethical regulations and was approved by the UCSF Institutional Review Board (13-12587, 17-22324, 17-23196 and 18-24633) and by The University of Hong Kong (HKU) Institutional Review Board (UW 07-273 and UW 21-112). As part of routine clinical practice at both institutions, patients signed a written waiver of informed consent to contribute deidentified data to research at the time of specimen collection given the minimal risk posed to subjects, as was previously described[16]. For samples from Baylor College of Medicine, patients provided written informed consent, and tumor tissues were collected under an institutional review board (IRB)–approved protocol at BCM by the Human Tissue Acquisition and Pathology Core (protocol H-14435), as was previously described[19,41].

## Meningioma samples and clinical data

Meningioma samples for the integrated cohort were collected from two sites, UCSF and Hong Kong University. Samples from the UCSF discovery cohort ($n = 200$) were selected from the UCSF Brain Tumor Center Biorepository and Pathology Core in 2017 and comprised all available WHO grade 2 and 3 meningioma frozen samples, and WHO grade 1 frozen samples with clinical follow-up greater than 10 years ($n = 40$) or those with the longest available clinical follow-up less than 10 years ($n = 47$). The electronic medical record was reviewed for all patients in late 2018, and paper charts were reviewed in early 2019 for patients treated before the advent of the electronic medical record. The Hong Kong University external test cohort 1 ($n = 365$) was comprised of consecutive meningiomas from patients treated at Hong Kong University from 2000 to 2019 with frozen tissue that was sufficient for DNA methylation profiling. The medical record was reviewed for all patients in late 2019. For both cohorts, meningioma recurrence was defined as new radiographic tumor on magnetic resonance imaging after gross total resection, or progression of residual meningioma on magnetic resonance imaging after subtotal resection.

External cohort 2 was comprised of meningioma samples with available DNA methylation profiling data from Baylor College of Medicine and included previously published ($n = 110$)[41] and unpublished ($n = 16$) samples with clinical outcomes data. Sample collection, preparation, and analysis for this cohort were previously described[41]. For all samples, patient sex was collected from the electronic health record and was included where applicable as a covariate in regression analyses.

## Meningioma DNA methylation profiling and analysis

DNA methylation profiling was performed as previously described[16] using the Illumina Methylation EPIC 850k Beadchip (WG-317-1003, Illumina) according to manufacturer instructions. Pre-processing and β-value calculations were performed using the SeSAMe (v1.12.9) pipeline with default settings (BioConductor 3.13). All DNA methylation profiling was performed at the Molecular Genomics Core at the University of Southern California.

## TCGA SNP and clinical outcomes data

TCGA data were collected from the TCGA (https://gdc.cancer.gov/about-data/publications/pancanatlas)[8,42] and copy number information was obtained using the Copy Number Dataset (broad.mit.edu_PANCAN_Genome_Wide_SNP_6_whitelisted.seg). Only primary tumor samples were included by filtering TCGA Biospecimen Core Resource (BCR) barcodes for sample numbers containing the "01" and "03" designator. Clinical information was obtained from the TCGA-Clinical Data Resource (CDR) Outcome Dataset (TCGA-CDR-Supplementary Data 1.xlsx) and was matched to CNA data by BCR barcode. TCGA histological type designations were reviewed by a board-certified pathologist (KM) and were adjusted to distinguish between molecularly distinct cancer subtypes based on current WHO criteria.

## CNA analysis

CNA profiles were generated from meningioma DNA methylation data using the SeSAMe package. As previously described[16], the "cnSegmentation" command with default settings and the 'EPIC.5.normal' dataset as a copy-number normal control were used.

For both meningioma DNA methylation data and TCGA SNP array data, chromosome segments with mean intensity values less than −0.1 were defined as lost, and mean intensity values greater than 0.15 were defined as gained, as previously described[16]. CNA profiles excluded sex chromosomes and p arms of acrocentric chromosomes (13p, 14p, 15p, 21p and 22p). CNA threshold analysis for each CNA profile was performed by measuring the mean intensity value at intervals of 30,000 bases along each chromosome arm and summing nonconsecutive gains and losses. Intensity value thresholds were selected based on manual inspection for local minima among the distribution of mean values across all segments in the 565 meningiomas comprising the integrated cohort, as previously described[16]. These values were stable across meningiomas from the two sites used to build the integrated cohort, but to test this we examined whether varying the intensity threshold by 0.05 increments significantly altered the prognostic value of each chromosome arm and found no clear optimal intensity threshold for defining any CNA events. Similar manual inspection was performed for the SNP-array data used to generate CNA results from TCGA. As a sensitivity analysis, we repeated the process of defining CNAs using incremental intensity value thresholds for gain and loss and did not identify an intensity that significantly improved prognostic value of individual CNAs beyond those described above for losses (−0.1) or gains (0.15).

The total number of CNAs that met thresholds for loss or gain from 1% to 99% by 1% increments of each chromosome arm were counted. 5-year AUC for a time-dependent ROC curve for meningioma LFFR and OS, and for TGCA PFS and OS, were used as primary measures of prognostic value for individual CNAs. Prognostic models throughout this study were calculated for each threshold using the *survivalROC* package (v1.0.3.1) in R, and the optimal threshold for each CNA was chosen based on the highest AUC for each clinical endpoint. AUC standard deviation was calculated for each CNA across 1-99% thresholds, and "size-dependent" CNAs were defined as (1) an AUC of at least 0.60 at any CNA threshold, (2) an AUC standard deviation of at least 0.01 across CNA size thresholds, and (3) prevalence in at least 2.5% of samples when defined using the optimal size threshold. As a comparison, extent of resection, which is a well described prognostic variable for meningioma, achieved a 5-year LFFR AUC of 0.59. Thus, a univariate AUC of 0.60 was felt to be a reasonable benchmark for a prognostic CNA. Use of a higher AUC threshold of 0.65 resulted in only 2 prognostic CNAs. The cutoff of a standard deviation of 0.01 was chosen by estimating the false discovery rate (FDR) of the standard deviation of AUC across size thresholds using both permutation and bootstrap methods. Sample-wise percent-arm altered analyses were either permuted without replacement or resampled with replacement (bootstrap) for each chromosome arm, maintaining the chromosome arm specific background distribution but scrambling the relationship between percent-arm alteration and sample. This was repeated 500 times to generate a distribution of standard deviations which serves as an estimate of the FDR. The standard deviation cutoff of 0.01 corresponded to an FDR of 5.3% or 5.4% by permutation or bootstrap, respectively. A cutoff of CNA presence in 2.5% of samples was used to exclude rare CNAs to improve the stability of our analyses.

CNA network plots were constructed using the *igraph* package (v1.5.1) in R. Plots were constructed using CNAs selected from regression models for meningioma. CNAs were defined using chromosome-specific optimized size thresholds for co-occurrence analyses.

Cluster analysis was performed using CNAs defined at chromosome-specific optimized size-threshold for predicting LFFR or OS. Clusters were built on the presence of each CNA, and binned CNA burden was across the x-axis for visualization. Clustering was done using the *factoextra* (v1.0.7) and *cluster* (v2.1.4) packages in R and visualized with the *ComplexHeatmap* package (v2.15.4). The optimal number of clusters was determined using K-means and corroborated with the Density-based spatial clustering of applications with noise (DBSCAN) algorithm using the *dbscan* (v1.1-12) package in R. CNA burden was defined as the number of CNA events per sample. Recursive partitioning analysis to stratify samples by CNA burden and predict LFFR or OS in meningioma was performed using the *rpart* (v4.1.23) package in R.

## Survival analysis and modeling

CNAs using chromosome-specific optimized size threshold were used to train regression models on meningioma samples for feature selection. LASSO and Elastic net regularized Cox regression models were

trained on the discovery cohort (UCSF) with the concordance index (c-index) for each target endpoint, using the *glmnet* and *cv.glmnet* functions from the *glmnet* package (v4.1-8) in R. Elastic net model selection was performed by selecting an optimal alpha value from a range of 0.05 to 0.95. Model training was performed using 10-fold cross validation. Prognostic size-dependent CNAs for each model were identified within 1 standard error of the model achieving maximal c-index to reduce over-fitting.

Cox proportional hazards models were trained on the discovery cohort using selected size-dependent CNAs from LASSO and Elastic net regression models, either alone or in iterative combinations with WHO 2016 histological grade, *CDKN2A/B* loss, and mitoses per 10 high-power fields. Prognostic risk scores were obtained from these models in external test cohort 1 and external test cohort 2 and used for subsequent comparisons to Integrated grade and Integrated score. To build a model with improved prognostic performance, size-dependent CNAs were used as inputs into a gradient boosting model (XGBoost), either alone or in iterative combinations with WHO 2016 histological grade, *CDKN2A/B* loss, and mitoses per 10 high-power fields, using the *xgboost* (v1.7.7.1) package in R. These variables have been previously demonstrated to boost prognostic performance of risk stratification models incorporating meningioma CNAs[14]. Parameters for the XGboost models were selected to optimize performance and included using the *gblinear* booster, the "survival:cox" objective, a learning rate of 0.009, and maximum depth of 6, with a total of 2 decision trees included in final models. XGboost risk scores were stratified into three discrete groups (low, intermediate, and high risk) using the 25th and 50th percentiles of the score distribution in the training set.

Nomograms for models were constructed using the predicted risk scores in the discovery cohort linearly rescaled from 0 to 1 in combination with extent of resection and newly diagnosed versus recurrent presentation using the *rms* (v6.8-0) package in R.

Integrated grade[14] was assigned to meningioma samples using CNAs, mitoses per 10 high-power fields, and *CDKN2A/B* loss. Integrated score[15] was assigned using CNAs, WHO grade, and DNA methylation family[23], the latter which had been previously assigned independently by the authors who developed of this system.

Individual model performance was measured using 5-year LFFR AUC and 5-year OS AUC. Prognostic risk scores for CNA size-dependent models were stratified into quantiles for Kaplan-Meier analysis, which was performed using the *survminer* (v0.4.9) package in R. Likelihood ratio testing was performed with the *lmtest* (v0.9-40) package to assess model performance in combination with other models. Multivariate Cox proportional hazards analysis was performed using the *survival* package (v3.5-7) in R.

## CNA co-occurrence analysis

Co-occurrent CNA pairs including at least one size-dependent CNA were included as variables in LASSO or Elastic net regularized Cox models to identify prognostic pairs. Co-occurrence analysis was largely restricted to CNA pairs as sample size was insufficient to analyze the high number of predictors involved when using 3 or more CNAs. Model training was performed using 10-fold cross validation and similar parameters as for single-CNA analyses described above. For meningioma, models were first trained in the discovery cohort and then validated in external test cohorts 1 and 2. For TCGA cancers, models were trained and cross-validated using the entirety of each cancer cohort. The most important pairs identified in models were selected for Kaplan-Meier and multivariate Cox regression analyses, stratified by the presence of each pair, either of the two component CNAs, or neither CNA, in external test cohort 1.

## Sliding window binned CNA analysis

Chromosome arms were divided into bins of 200,000 bases, which was chosen as it approximates the size of the *HLA* locus, a prominent

distinguishing marker that is gained or lost in different DNA methylation groups of meningiomas[16]. For each bin a weighted average of the mean intensities of all segments overlapping with the bin was calculated, weighted to account for length of the overlap and the number of methylation probes in each segment. Weighted average values less than −0.1 were defined as lost, and those greater than 0.15 were defined as gained, as described above. The prognostic value of each focal bin was measured using AUC for 5-year LFFR or AUC for 5-year OS. Focal CNAs with prognostic value were defined as those for which the AUC was at least 0.04 higher than the median for the entire chromosome arm, which was chosen as it approximates the magnitude of the AUC peak corresponding to the *CDKN2A/B* locus, another well-known recurrently lost region with significant prognostic value in meningioma.

## Focal genomic and ontology analyses

CNA pileup plots demonstrating the proportion of tumors with losses or gains at each position along each chromosome arm were constructed using the *ggplot2* (v3.4.3) package in R. Focal regions of loss were selected by manual inspection of regions along the chromosome arm with a notably higher proportion of samples demonstrating deletion compared to the surrounding regions. This method was chosen to better account for subtle variability in the baseline frequency of loss across each chromosome arm or artifactual peaks or valleys due to uneven distribution of methylation probes. Manual inspection of focal CNAs has been previously reported as a preferred method of identifying hemizygous and homozygous deletion of the *CDKN2A/B* locus[39,40]. Genes present in focal regions of loss and in prognostic regions identified on sliding window binned CNA analysis were identified by cross-referencing positions along the chromosome with the Ensembl (release 109)[43] database using the *biomaRt* (v2.54.1) package in R.

## Meningioma gene expression analysis

Meningioma gene expression analysis was performed using sample-matched RNA sequencing data, as previously described[44]. Briefly, RNA sequencing was performed on all 200 of the UCSF discovery cohort samples and 302 of the Hong Kong University external test cohort 1 samples meeting quality metrics. For UCSF samples, library preparation was performed using the TruSeq RNA Library Prep Kit v2 (RS-122-2001, Ilumina), sequencing was performed on an Illumina HiSeq 4000 to a mean of 42 million reads per sample at the UCSF IHG Genomics Core, quality control of FASTQ files was performed with FASTQC (v0.11.9), and 50 bp single-end reads were mapped to the human reference genome GRCh38 using HISAT2 (v2.1.0) with default parameters. For Hong Kong University samples, library preparation was performed using the TruSeq Standard mRNA Kit (20020595, Illumina) and 150 bp paired-end reads were sequenced on an Illumina NovaSeq 6000 to a mean of 100 million reads per sample at MedGenome Inc. Analysis was performed using a pipeline comprised of FastQC for quality control and Kallisto for reading pseudo alignment and transcript abundance quantification using default settings (v0.46.2). Comparison of gene expression levels was performed using the Wilcoxon rank-sum test on log2 transformed transcripts per million.

Gene ontology and interaction analysis were performed using the *clusterProfiler* (v4.10.1) package in R. The enrichGO() function was used to map genes contained in focal regions of interest to known gene ontologies using the "org.Hs.eg.db" human annotation. Dotplots to visualize ontology results were generated using the *enrichplot* (v1.20.3) package in R.

## Statistics

All experiments were performed with independent biological replicates and repeated, and statistics were derived from biological replicates. Biological replicates are indicated in each figure panel or figure

legend. No statistical methods were used to predetermine sample sizes, but sample sizes in this study are similar, or larger to those reported in previous publications[16,23,45,46]. Data distribution was assumed to be normal, but this was not formally tested. Investigators were blinded to conditions during clinical data collection and analysis. Bioinformatic analyses were performed blinded to clinical features, outcomes, and molecular characteristics. The clinical samples used in this study were retrospective and nonrandomized with no intervention, and all samples were interrogated equally. Thus, controlling for covariates among clinical samples is not relevant. No data points were excluded from the analyses. Statistical analyses were conducted in R (v4.2.2).

### Reporting summary

Further information on research design is available in the Nature Portfolio Reporting Summary linked to this article.

## Data availability

DNA methylation and RNA sequencing for meningiomas in Discovery cohort 1 and External validation cohort 1 have been deposited in the NCBI Gene Expression under the accessions GSE183656 (DNA methylation, $n = 565$; RNA sequencing, $n = 185$) [https://www.ncbi.nlm.nih.gov/geo/query/acc.cgi?acc=GSE183656], GSE101638 (RNA sequencing, $n = 42$) [https://www.ncbi.nlm.nih.gov/geo/query/acc.cgi?acc=GSE101638], and GSE212666 (RNA sequencing, $n = 302$) [https://www.ncbi.nlm.nih.gov/geo/query/acc.cgi?acc=GSE212666]. DNA methylation and RNA sequencing for meningiomas in External validation cohort 2 have been deposited under the accession number GSE189521 ($n = 110$) [https://www.ncbi.nlm.nih.gov/geo/query/acc.cgi?acc=GSE189521]. The publicly available GRCh38 (hg38, https://www.ncbi.nlm.nih.gov/assembly/GCF_000001405.39/), and Kallisto index v10 (https://github.com/pachterlab/kallisto-transcriptome-indices/releases) datasets were used in this study. TCGA data was collected from the publicly available TCGA PanCanAtlas (https://gdc.cancer.gov/about-data/publications/pancanatlas). Copy number information was obtained using the Copy Number dataset (broad.mit.edu_PANCAN_Genome_Wide_SNP_6_whitelisted.seg). Clinical information was obtained from the TCGA-Clinical Data Resource (CDR) Outcome dataset (TCGA-CDR-Supplementary Data 1). Source data are provided with this paper.

## Code availability

Relevant scripts used for the core analyses of this manuscript, along with meningioma copy number segmentation data for our cohort of 565 meningiomas are available on Zenodo (https://zenodo.org/doi/10.5281/zenodo.11501566, https://zenodo.org/doi/10.5281/zenodo.11501381)[47,48], licensed under the MIT License, allowing reuse with attribution.

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

## Acknowledgements
This work was supported by funding from American Brain Tumor Association Jack & Fay Netchin Medical Student Summer Fellowship in memory of Rose Digang and NIH TL1/UCSF Yearlong Research Fellowship to M.P.N., K12 CA260225 and the Chan Zuckerberg Biohub Physician Scientist Fellowship to W.C.C., and R01 CA262311, P50 CA097257, the UCSF Wolfe Meningioma Program Project and the Trenchard Family Charitable Fund, to D.R.R. Research reported in this publication was supported by the National Cancer Institute of the National Institutes of Health under Award Number K12CA260225 (W.C.C.). The content is solely the responsibility of the authors and does not necessarily represent the official views of the National Institutes of Health.

## Author contributions
M.P.N. and W.C.C. designed and performed the experiments and analyses; M.P.N., W.C.C., and D.R.R. designed the study; K.M., A.C., and N.Z. provided data and performed analyses; V.N., T.J.K., S.T.M., C.G.L., and A.J.P. provided data and were involved in study design. M.P.N., W.C.C., and D.R.R. wrote the manuscript. All authors discussed the results and commented on the manuscript.

## Competing interests
The authors declare no competing interests.
