## [Transparent Peer Review file · Nature Communications]

Pan-cancer copy number variant analysis identifies optimized size thresholds and co-occurrence models for individualized risk-stratification

Corresponding Author: Dr David Raleigh

Version 0:

Reviewer comments:

Reviewer #1

(Remarks to the Author)

Summary and Significance

The authors have developed an approach to improve the prognostic value of copy-number by systematically optimizing the size thresholds used for inclusion in prognostic criteria. Applying this approach in a cohort of meningioma patient samples, the authors identify several optimized prognostic CNAs, as well as co-occurrent CNAs, that are not used in current prognostic criteria for meningioma. They then extend this approach to optimize size thresholds for CNAs across 9 TCGA cohorts. This approach seems actionable and may improve prognostic criteria, thus providing real clinical value.

Major Limitations

While the core analysis of this study has value, this manuscript, and some aspects of the study, are incomplete. The interpretation of the data is insufficient in many areas and discussion of the context, significance, and limitations of this study are missing. Additionally, while the presence of focal alterations is acknowledged, no attempt is made to determine if these, or any other chromosomal region, or any confounding factors (e.g., subtype), are the source of the prognostic value of size-dependent CNAs.

Major Issues

Data Analysis and Validity

1. Do size-optimized CNA improve risk stratification models? How much? This study does not directly answer that question aside from identifying how CNA size thresholds change their prognostic value (Fig 1B). Please provide direct comparison of the prognostic value of these model vs the current standard (integrated grade/score). An example would be to provide the ROC curves for the standard vs optimized models.
2. Does subtype influence the prognostic value of size dependent CNAs? Is the contribution of single or co-occurring CNAs to prognosis consistent across a cancer type or are these picking up residual signal from subtype-specific CNAs that may not necessarily be prognostic? This should be explored analytically (and discussed) to the extent possible and will significantly improve the clinical significance of these results.
3. The presence of focal alterations within size dependent CNAs in both datasets raises the question that these may be solely responsible for their prognostic value. This study is expanded to characterize which genes are present in these areas, thus additional analysis of their sole prognostic value is warranted.
4. Along these lines, focusing on the percentage of a chromosome arm is gained/lost overlooks the potential prognostic value of individual regions of the chromosome as well as the magnitude of the CNA (particularly for gains). For example, consider a whole-chromosome gain, which would be included in both a 5% or 75% threshold. In this setting, would a focal amplification on the same chromosome with real, prognostic value be overlooked? Would the magnitude of this amplification be overlooked? Varying additional inclusion criteria, including using a sliding window across each chromosome (e.g., using only CNAs within a 20Mb bin) or copy number-based thresholds (e.g., using only CNAs with signal intensity above 0.1, 0.2, 0.4, etc.) may help identify the source of the prognostic value (e.g., specific regions or cumulative effect?). This additional analysis would significantly improve this study.

Clarity, Presentation, and Context

5. Discussion of the context, significance, and limitations of this study is missing. This manuscript is incomplete.

Minor Issues

Data Analysis and Validity

6. Size-dependent and co-occurring CNAs are identified in about 50% of cancers in TCGA. However, characterization of

these (within Figure 3 and the text) is very sparse despite additional data being present in the supplement. Please expand the summary of this dataset past GBM and CESC. For example, what prognostic size dependent CNAs were identified that are or are not included in current prognostic criteria? Do any of these align with known, relatively aggressive subtypes as identified in the literature?

7. Line 64 — Gene ontology analyses using genes included in focal deletions is not sufficient to support dysregulation of these pathways. This conclusion oversteps what the analysis can support and should be corrected.

8. Line 78 — Hierarchical clustering performed in this way is not sufficient to independently gauge the relative timing of CNAs across individual cancers. While it is encouraging that many early CNAs are captured by your “early” cluster, it is important to note here that the original analysis (Magill et al.), as the authors know, was done in spatially distinct samples within the same tumor. Here, the conclusion that this analysis reveals early vs late CNAs seems overstated and this caveat should be addressed. Inclusion of tumor-grade may bolster this claim.

Clarity, Presentation, and Context

Nature Communications is a multi-disciplinary journal. Therefore, care should be taken to make results broadly accessible. Please consider the following changes to improve the readability of this manuscript.

9. The first paragraph of the main section refers to CNA signatures as a “marker of chromosome instability”. CNA signatures alone are not a reliable measure of CIN, but are a measure of aneuploidy, the product of CIN.

10. It is not immediately evident that AUC is used to assess the quality of the model rather than an orthogonal metric for survival. Please clarify this in the text.

11. Please clarify the rationale of the chosen criterion for size dependent CNAs: “a maximum AUC for 5-year LFFR or OS of at least 0.6 that decreased by at least 5% from the maximum AUC as CNV threshold varied”.

12. Please provide one or more examples of ROC curves from which AUC was calculated and how these curves change with size thresholds.

13. Figure 1F (and Extended Data Figure 4) — The interpretability of these figures is limited by exclusion of the names of the sub-processes involved (blue nodes). Please provide a key.

14. Extended Data Figure 1 — As presented, the scales for each plot are different causing some chromosomes to appear larger or smaller. Either show the same chromosomes for each sample or provide the x-axis scale. Also please consider an alternative color scheme as green/red on a grey background is not accessible for some.

15. Extended Data Figure 2 — Interpretation of these results is lacking. Also, there appear to be data off the scale in Extended Data Figure 2b which need to be explained and/or the scale needs to be expanded.

16. Please comment on potential sources of differences between CNAs that are prognostic in OS vs LFFR and those identified exclusively using the LASSO or elastic net models. This consensus approach, using two methods has value but should be examined accordingly.

17. Though citation of reference #21 is encouraged as it is relevant to this study (e.g., does this study identify the same novel prognostic CNAs?), it appears to be cited incorrectly in the text in reference to the PanCancer Atlas. Please provide the correct citation for the PanCancer Atlas in place of this.

18. Copy number changes, as described in this manuscript, are more accurately named “copy number alterations” (CNAs) rather than “copy number variants” (CNVs). See PMID: 23412801.

Reviewer #2

(Remarks to the Author)

In their article “Pan-cancer copy number variant analysis identifies optimized size thresholds and co-occurrence models for individualized risk-stratification”, Nguyen and Chen et al. present an analysis of meningiomas whereby they test the hypothesis that copy-number variants (CNVs) can have an optimal size threshold for prognostic power and then apply this analysis to The Cancer Genome Atlas (TCGA) cohort to uncover similar cancer-specific size-thresholds.

CNVs are a hallmark of many cancer types and using them as leverage for prognostic power has been an active area of research for many years. However, the difficulty in the problem comes from the fact that whole chromosome arms are rarely fully intact, and instead the amplified or deleted regions tend to be more fragmented, so the question of how we quantify the amount of DNA gained or lost is somewhat sensitive to the resolution one uses. As such, the paper is concerned with an important problem that can have significant clinical implications. However, I have some major concerns about the paper in its current form that should be addressed before deciding about publication.

I have broken down my major comments into categories corresponding to the three main analyses of the paper.

Major:

General Remarks:

1. I have concerns about the paper not including any of the analyses performed. The authors state that they did not develop any software packages or tools in the “Code Availability” section, but clearly their computational analyses are extensive and as such they must have produced some code. In the spirit of transparency and reproducibility, it is vital that the authors include the analysis scripts. As a bare minimum, the code to reproduce the figures in the paper should be included.

2. Similarly, the most important derived datasets must be made publicly available, in particular the copy-number profiles of the meningiomas. Since CNVs are non-identifiable data I would recommend uploading them together with the source code (see point 1 above) to Zenodo.

3. The main body of the article is extremely dense. There is barely any introduction and as such the work is not thoroughly placed in the existing literature. There is little to no discussion of their intermediate results and the summary paragraph feels somewhat short and underdeveloped. Where does their work stand in the field? Do they have any suggestions for future work or for further research directions?

4. Starting with the TCGA analysis (l.85 ff), there are virtually no references to existing literature. Are the TCGA results novel? Also for the analysis of GBM and CESC samples, there are zero references to the previously-found features of gains and losses (arm-level or otherwise). It is again unclear if the authors' work is entirely novel or if they have recapitulated previous (unmentioned) results.

Size-dependent CNAs:

5. Fig. 1a demonstrates the prognostic capabilities of individual gains and losses for local freedom from recurrence (LFFR) and overall survival (OS), where the authors claim there are size-dependent CNA thresholds. There are a number of boxes highlighting specific thresholds on a subset of chromosome arms. Given the text and the figure caption, it appeared at first glance that there are no other size-dependent CNAs (though Supp. Table 1 contains a list of the "Optimal CNV threshold[s]"). Are the size-dependent CNVs the same as the optimal CNV thresholds? If so, why not highlight all such optima in Fig. 1a?

6. Further to Fig. 1a, the authors are attempting to understand the size-dependence of CNAs, but does this not depend on the location or contiguity of the CNA? This is hinted at by the authors' noting that there are focal amplifications and deletions in a number of chromosomes (Fig. 1d). The authors do not motivate why they only consider size in this analysis and not where CNAs can occur. I would expect that a priori 5% gain in areas of high density of oncogenes is (presumably) not the same as 5% gain dispersed across areas of no major function.

7. The summary of Fig. 1a to me is that the prognostic power of any single CNV gain or loss on a single chromosome is not particularly good (with the exception of an extremely small select subset). Why do the authors choose an AUC of ≥ 0.6 as the benchmark? What happens to the analysis if the benchmark is set higher? Also, the area under what curve was even used for the AUC? Precision/recall? Receiver/operator? From the name of the R package I assume the latter, but this needs to be stated explicitly. Further, assuming a ROC curve, an AUC of around 0.6 is not particularly impressive, given that an AUC of 0.5 is no better than random guessing.

8. Why are the thresholds (l.38 / Fig 1a) varied at 5% intervals? There is no justification in the text for this. How stable are the optimal CNV thresholds with respect to the threshold step size? What happens if one picks e.g. 8% or 13% thresholds? Are the specific optima used in the paper simply artifacts of the choice of bin size?

9. Further to this point, for the 6q losses, there is a double peak from what I can tell: a peak at the $\geq 5\%$ threshold (the size-dependent CNA), and then a peak for a pair of bins (at $\geq 25-35\%$ for LFFR and $\geq 15-25\%$ for OS). The authors do not comment on the significance of this, and could it be attributed to the fact that the analysis is binned in 5% increments?

10. The authors state the performance (i.e. the AUC) for the LASSO and elastic net models for LFFR and OS (l.58 ff.), but do not comment on the fact that these measures seem to have roughly the same performance as the conventional integrated grade and score (judging from Fig 1b). The authors need to include a thorough comparison between the integrated grade and score model and their new model based on optimized CNV thresholds, to underpin the statements made in the title and the rest of the paper. They do note that their trained models identify some chromosome arms excluded by the conventional methods, but do not comment on the fact that some of these arms (such as 5p/q) showed poor individual prognostic power. To me, this could actually make a stronger case for the second part of their analysis: the co-occurrence of certain "size-dependent" CNVs.

Co-occurrence in meningiomas:

11. The size-dependent CNA co-occurrence analysis (l.67 ff / Fig 2) is interesting, though it is unclear if this is novel or a recapitulation of previous results found in the literature. If the results are novel, the authors may benefit from stating this more strongly. I would also like to know the predictive power of non-optimal CNA thresholds for both 1p/22q and 9p/14q curves. Does the size-dependence still have a significant effect as the authors claim?

12. The authors show the impressive results for 1p/22q and 9p/14q (Fig 2b), which are the predominant components of the LASSO and elastic net models for LFFR and OS (Extended data figure 3a). However, the authors make no mention of another relatively high weighted component: 22q/9p. Why is this absent?

13. The authors do not at all discuss the differences between co-occurrent CNA pairs for LFFR and OS and the two models used (LASSO vs Elastic net, Extended data figure 3a) For example, for LASSO LFFR vs OS, there are a number of pairs for the OS model that do not appear in the LFFR model. Similarly with the elastic net model. Can you posit any reason as to why this would be?

14. The number of clusters used in the unsupervised clustering is 3 for both LFFR and OS (the evidence the authors use is the "elbow plot" in Extended Data Fig. 3b and d). But why have they picked 3 and not 4? Three clusters already demonstrate some level of distortion of the score. How does the hierarchy look if there are instead 4 clusters and how drastically does the

heatmap of Fig. 2c change? If there are no significant changes with 4 clusters, then I would side with the authors and choose the lesser number, but I have no current a priori reason for seeing that it should be 3, other than the authors arguing for “early”, “late”, and “miscellaneous” CNAs.

15. A more general point to the k-means clustering: The method has a number of large drawbacks, for example, the assumption that the underlying data can be clustered into separable spherical clusters, and clusters should be of a similar size. Are these assumptions fulfilled? How sensitive is the data to the choice of clustering method? Have the authors tried alternatives, such as DBScan? Is there any other particular reason for using k-means clustering?

CNV size-dependence and co-occurrence in other cancers:

16. The finding that some cancers have “size-dependent CNAs” while others don’t (l.85 ff) is interesting, but the authors do not discuss potential reasons for this. In general, there is a lack of discussion about how the authors’ results fit into the current state of the field.

17. There is no mention of two quirky features in the co-occurrence analysis (Fig. 3c): 1. in GBM, 10q loss + 19q gain has a negative coefficient, and 2. CESC only has 2 components, much fewer than the other LASSO models the authors have shown in the rest of the paper. What does the Kaplan-Meier curve look like for 10q loss + 19q gain in GBM? Do the authors have any reason for CESC being well-described by just two covariants?

Minor:

1. In Fig. 1a, the heatmaps should have the same scale for all 4 plots, or at the very least the same scale for gains and for losses.

2. Fig. 1b: The effects of varying CNA thresholds on the integrated grade and integrated score are interesting, but then the authors do not comment on the fact that the integrated score is maximized at its conventional score.

3. What does Fig. 1d have to do with the LASSO and elastic net models? The order to the chromosomes is also odd. Why not order the plots by chromosome number? Why is chr20 included in the subfigure (while not being covariate in the models), but chr1 isn’t? Why does chr6 have the text “HLA” inscribed, though this is never mentioned in the text?

4. L.68: “Regularized Cox regression models using co-occurrent CNV pairs identified...” Are all possible pairs of co-occurrent CNV pairs used as possible hazard inputs?

5. L.96: Is CESC the abbreviation used for both subtypes of cervical cancer? Why are both included into the same dataset?

6. L.147: In the section “CNV analysis”, why are losses and gains defined with a different absolute mean intensity value? Is this conventional?

7. In Supplementary Table 5, what does the value “Preserved” mean for 20p loss in “size-dependent cancers”? Do the authors mean “Not Lost”?

Reviewer #3

(Remarks to the Author)

I co-reviewed this manuscript with one of the reviewers who provided the listed reports. This is part of the Nature Communications initiative to facilitate training in peer review and to provide appropriate recognition for Early Career Researchers who co-review manuscripts

Reviewer #4

(Remarks to the Author)

In this manuscript the authors applied statistical methods to determine the optimized size thresholds for arm-level CNV events across meningiomas for individualized risk stratification. They also applied the same method to TCGA cohort composed of various cancer types for a similar stratification.

While the manuscript presents some intriguing insights, several significant concerns detract from its clarity and the main conclusions.

Here are my major concerns:

1. While the manuscript’s central argument is valid, it is frequently misrepresented both conceptually and literally throughout the text. Specifically, the manuscript consistently refers to the cutoff for defining arm-level CNV events as “size thresholds for defining CNVs,” conflating two distinct terms and potentially leading to misinterpretation.

2. The considerable variability in cutoffs identified across both the Meningioma and TCGA cohorts underscores the manuscript’s inability to establish a universally applicable cutoff with prognostic value for risk stratification. The identified cutoffs vary between 0.05 -0.71 with the highest 0.95* Optimal AUC. The same variability is observed in the TCGA cohort (5% for chr7p gain vs 95% for chr7q gain in GBMs).

3. I find the conclusion to propose a superficial classifier, making it challenging to generalize, especially considering the

potential for clinical utility. However, it remains unclear whether the ultimate goal of the study was to define a classifier specifically intended for clinical application. Are the authors suggesting using ~78 distinct cutoffs to define chromosome level events? The manuscript briefly mentions an effort to define a "uniform" threshold by testing their approach against the previously published "integrated grade" and "integrated score" approaches. However even with this approach, the manuscript still comes up a very wide range of cutoff values (20-30% vs. 5%), which is not an improvement on the currently used varying cutoff values.

4. The identification of potentially new prognostic markers for new chromosome level CNV events are mentioned, but not shown how these specific events are "prognostic". Are these events prognostic alone or together with other events?

5. It is not clear how clustering based on CNV burden leads to a conclusion about timing of CNV occurrence. The presented analysis merely reveals co-occurrence patterns, rather than "timing" of occurrence. Moreover, it is not clear what authors mean by "burden" and "total number of CNVs". Is this only counting arm-level events determined by in the manuscript? Or does this include focal CNV events as well?

6. The definition and significance of "focal" CNVs require clarification. Since the large-scale/arm level CNV identification varies across the chromosomes in the study, does the cutoff of focal CNVs vary across the chromosomes as well? This is critical and should be better described. Also why did the authors choose to test the expression of genes in focal CNV regions against the other genes on the same chromosome arm rather than comparing to "same" region/genes in samples without the CNV events?

7. Detailed cohort description of both cohorts is missing and is not consistent. While Grade 1-2-3s are included in UCSF cohort, no such information is provided for the Hong Kong cohort? Furthermore, it is now known (and published many times) that higher grade meningiomas have higher CNV burden compared to grade 1 cases. Do the higher grade samples in this cohort indeed have higher CNV burden, when their optimized cut-off approach is applied to define arm-level events? The recurrence prediction in Grade-1 meningiomas vs. higher grade meningiomas should be calculated separately or should be corrected in the multivariate analysis. Furthermore, the treatment status of any of the higher grades is not reported, as they sometimes receive radiation treatment. The study does not take these clinically relevant variables into consideration.

8. The identified co-occurrence patterns in Meningiomas are quite intriguing and important.

Minor concerns:

1. The lack of descriptive clarity in figures and legends, particularly in Figure 1d-e-f, limits understanding. Clearer annotations would enhance the interpretability of the presented data.

2. It is interesting that chr7p amplification and chr10 deletion is not identified as a co-occurring event in GBMS. Even though it looks like it was identified in Figure 3b, it is not mentioned in the manuscript and the Kaplan-Meier curve was plotted for chr7p gain-chr16q loss, which seems to have a lower co-occurrence score. It is not clear why this co-occurrence pattern was selected.

Reviewer #5

(Remarks to the Author)

The study investigates optimal size thresholds for copy number variations (CNVs) and their co-occurrence patterns in outcome assessment models using DNA methylation arrays across meningiomas and other tumors. By identifying optimal chromosome size thresholds in various tumors, the authors developed predictive models that unveiled prognostically relevant and novel individual and co-occurring CNVs not detected by previous CNV burden-based models.

The subject is relevant, the paper concise, well-written and overall clear, except for the lack of a couple of definitions as delineated below. However, the advantage of their approach remains unclear as the authors do not compare the predictive performance of their novel approach with existing methods or demonstrate its generalizability to an external validation meningioma cohort. Furthermore, the absence of a detailed plan for a potential clinical deployment workflow renders the clinical significance of this work elusive.

Suggestions to strengthen the study include:

1. Comparing prediction performance with other CNV-based models.
2. Conducting a comparative analysis of outcomes between 1p/19q and 7p/16q for gliomas.
3. Investigating the influence of factors such as sex, tumor location, primary and recurrence status, and radiotherapy on size-dependent model performance.
4. Assessing the generalizability of findings to other meningioma cohorts.
5. Clarifying the definition of "early" and "late" CNV alterations.
6. Providing figures illustrating the three k-cluster differences and LFFR and/or OS.
7. Clarifying the weight of CNV burden in driving the identified clusters.
8. Detailing potential plans for CNV validation and scalability.
9. Developing a schematic workflow for model deployment and clinical application (from tissue obtention and profiling, model application through clinical utilization).
10. Inclusion of relevant references – Ma et al. 2020, Sybren et al. 2021, Sahm et al. 2017

Minor:

11. Downplaying statements suggesting a trend of co-occurrence of prognostic CNVs across all models, as only a few CNVs were common.

12. Temper the claim regarding the clinical relevance of CNV size-dependence and co-occurrence in human cancer, considering its applicability to only half of them.

Version 1:

Reviewer comments:

Reviewer #1

(Remarks to the Author)

I commend the authors on thoroughly addressing each point of my review. This manuscript is significantly improved. The study's scope and limitations are now defined, and the authors' have filled major gaps in their original analysis, interpretation, and reporting. Their methods are conveyed more clearly, their conclusions are supported by evidence, and their findings are contextualized by discussion of other studies.

Please see the remaining minor points below which are largely graphical/textual:

Points 10-12 — The use of ROC AUCs as your primary metric for model quality is much clearer. I do think providing additional representative examples of these ROCs in the manuscript (as requested and provided in the response to Point #12) would significantly aid others' understanding of the analytical method central to this study.

Point 14 — The scales of these CNA profiles remain somewhat unclear. When the authors plot copy number profiles for chromosomes (Fig 2a, Extended Figs 1 and 5), I assume that they are showing either the full chromosome or full chromosome arm. Providing genomic coordinates on the plot axes or as a scale bar would clarify this.

Figures 2a/2b — It is unclear how the data in Fig 2b were derived and what they represent. Do the distributions these means are derived from represent mean expression per gene for all meningiomas or mean expression per gene averaged across meningiomas? These distributions should be shown. Also, in relating the categories here ('No CNA' or 'Focal copy number deletion' and 'No AUC peak' and 'Focal AUC peak') to Fig 2a, it would be helpful to highlight exactly which regions are being marked as focal CNAs or focal AUC peaks and are therefore included in 2b.

Figure 4a & Extended Fig 9a/b — Indicating (1) the magnitude of the effect, (2) the statistical significance, and/or (3) the actual optimal size threshold for each data point on the plot would aid the impact/take-home of Figure 4. As of now, if I'm interested in a specific cancer type presented here, I have to look across several supplemental tables and figures to access this information.

Manuscript Density — I strongly encourage the authors to break up their results section into major sub-sections to improve readability of this very dense manuscript.

Reviewer #2

(Remarks to the Author)

With regards to my concerns about the background material, and the surrounding body of research, the authors have satisfied all the points I raised, including plenty of references to various different previous studies, and general better grounding of the present research in the extant literature.

The authors have also addressed all of my points about the size-dependent CNAs and their prognostic powers. Overall, the authors have drastically improved the paper, grounding their conclusions in many more relevant analyses. I'd like to highlight in particular the analysis of histologically distinct subtypes of cancer, which I thought to be the most interesting of the results presented.

I would recommend the editor accept the manuscript, with minor revisions:

Focal CNAs:

In the "Binned CNA Analysis" subsection of "Methods" (l.540ff), the authors say "Focal regions of increased prognostic value were selected by manually inspecting AUCs across a chromosome arm and selecting bins with an AUC at least 0.04 greater than the median for that arm." I suppose that the authors mean to say they have a computer script that can do this. "Manual" suggests, to me, "by eye". Actually that whole section feels somewhat repetitive, so the authors might benefit from editing it down.

Further to that point, the authors choose a "resolution criteria" (my words) for deciding if a focal CNA is prognostic. My reading of the text has two conflicting definitions for the resolution criteria: (a) $AUC \geq 0.04 + \text{mean AUC for that arm}$, and (b) $AUC \geq 1.04 * \text{mean AUC for that arm}$. The second definition comes from l.534: "defined as those for which the AUC was at least 4% higher than the median", the first from l.542: "AUC at least 0.04 greater than the median for that arm." The authors should make clear which is the actual case. I assume it is (a).

l.173ff, I don't really understand this segment as it stands in relation to Fig. 2a: "Other than identifying the same regions of focal loss or gains as our sliding window approach (Fig. 2a), this analysis did not reveal clear recurrent regions of focal deep deletion or high level amplification, except for the CDKN2A/B locus on chromosome 9p, which is expected to be

homozygously deleted in some samples, and the HLA locus on chromosome 6p, which is expected to be homozygously deleted or amplified in some samples". Don't I see from the plot what the authors are trying to say. Aren't the "clear recurrent regions of focal deep deletion" the blue regions of the copy-number profiles (bottom panels/row) which have a large amplitude (e.g. near the centromere for chr3, beginning of 7q, end of 8q, etc.)? Also, the authors do not comment on the fact that several of these recurrent copy-number losses do not correlate with changes in the prognostic AUC (or in fact, sometimes anti-correlate). I think this is worth mentioning. The areas I am thinking of are the end of 18q, the spike near the centromere of 10q, some of the losses on 8p, to name a few examples.

Further to that point, in I.547ff: the authors say that "Focal regions of loss were selected by manual inspection of regions along the chromosome arm with a higher proportion of samples demonstrating deletion compared to the surrounding regions." This feels a little vague, and raises two points: (a). Why not automate the procedure? (b). What does higher proportion mean?

Co-occurrence modeling:

Extended Data Fig. 6 - The Kaplan-Meier curve for LFFR (1p/7p/22q) is either missing a line or else it appears that the 1p loss and the 7p loss are overlapping. If so, the authors may benefit from using different line styles or more contrasting colours with a lower transparency. Further to this point, the neighbouring Kaplan-Meier curve for OS (1p/9p/22q) is definitely missing a line, which as far as I can tell is for the 9p/14q co-deletion class.

I.187ff - I am not entirely sure I understand the sentence: "Each of these CNA pairs were prognostic when defining at optimal size thresholds and across thresholds of 5% or 80% in external test cohort 1". What do the authors mean by "across thresholds of 5% or 80% in external test cohort 1"?

I.225ff - The authors may benefit from a small selection of figures (supplementary, extended, or otherwise) to back up this statement: "Unlike meningioma, several TCGA cancers had size-dependent co-occurrent CNA pairs that were associated with better outcomes, including ACC, BLCA, HNSC, LAML, LUSC, and THCA."

Extended Data Fig. 8a - Am I wrong in saying that beyond the 2 year mark, 18p loss is not significantly better than the triple deletion case? Looking at the 50% survival time, the two are comparable. I would say that the authors are overstating their results in the caption of the figure, though I would agree that all the other co-deletion/intact cases have a better patient outcome than the triple deletion case. It is interesting to note that 18q loss has a better PFS than if it is intact.

Discussion:

I.287 - The authors mention their analysis of focal CNAs in chromosome 1, but did not show it in any figures. Figure 2a demonstrates the focal CNAs of interest, fair enough, but if the authors wish to discuss 1p, they should demonstrate this in at least a supplementary or extended data figure.

Other:

Extended Data Fig 6. - The authors may wish to modify the layout of the lower right quadrant of the figure. The two Kaplan-meier curves look a little scattershot at present, and the LASSO Cox regression coefficient boxes are uneven. The network diagrams in the top left should also have a label near them saying LFFR and OS respectively.

Fig 3b - Is the top row (the plot of WHO grade 2, 3 meningiomas) a percentage, i.e. the y-axis ranges from 0.2 to 0.6% or is it from 20% to 60%? If it is the latter, then the label "% WHO grade 2 or 3" is inaccurate.

Reviewer #3

(Remarks to the Author)

Reviewer #4

(Remarks to the Author)

The authors performed a major revision in response to my comments and those of other reviewers, which have improved several aspects of the manuscript. However, the "Results" section has now become exceedingly dense, making it difficult to read and interpret. A major revision is needed to improve clarity, with the following recommendations:

- a. The "Results" section would benefit from subsections that clearly outline individual approaches and findings. Additionally, a study workflow diagram would improve readability and provide an overview of the discovery and test cohorts, methods (e.g., sliding window, incremental percentage), and statistical/machine learning approaches with key findings.
- b. Some methodological details currently included in the "Results" should be moved to the "Methods" section.

Below are my other major and minor comments:

Major Comments:

1. The study promises two main contributions: (i) identifying chromosome-specific variable size thresholds, and (ii) analyzing co-occurrence patterns to better predict tumor control and survival. While the co-occurrence patterns and their predictive

strength for clinical outcomes are intriguing, I have concerns about the introduction of chromosome-specific size thresholds, as detailed below.

- a. Thresholds for CNV sizes have traditionally distinguished between chromosome arm-level events and focal events. However, thresholds in this study range from 0.01 to 0.97, potentially blurring the line between focal and arm-level CNV events. Key points for clarification include:
- b. The study suggests an optimal threshold of 0.01 for chr1p amplification. Is it proposing that a 1% amplification of chr1p should be interpreted as an arm-level CNV event with prognostic value? This requires clarification. If not, the authors should clearly outline their intended interpretation.
- c. Focal vs. Arm-Level Events: A 0.01 threshold for chr18p gain corresponds to ~160kb, similar to the size of focal events identified in 200kb bins. The authors should address this blurring of distinctions between focal and arm-level events.
2. The "integrated grade" approach with a 17% threshold across all chromosomes achieved an AUC of 0.79, while the variable thresholding method reached only 0.65 (with only two significant events). Similar results were observed for the "integrated score" at a 4% standard cutoff. Given these findings, it's unclear how variable thresholding offers an improvement over a uniform cutoff.
3. The authors incorporate CDKN2A loss as a separate variable in the integrated model, even though the size-dependent CNV data theoretically captures chromosome 9. This raises the question: does the model effectively capture CDKN2A loss, or is it necessary to include it as a separate variable?
4. Focal CNVs are defined as prognostic if the AUC is greater than the median AUC of all CNVs for a given chromosome. However, the authors should compare these focal CNVs against the variable threshold approach to determine if they offer better prognostic performance.
5. The absence of chr1p-19q co-deletion in the size-dependent analysis is concerning, as this is a diagnostic marker for oligodendrogliomas.
6. On lines 334-336, the authors suggest that chromosome-specific thresholding is feasible for clinical application. This argument is subjective and should be reconsidered, as focal CNV analysis often leads to well-established clinical markers, such as CDKN2A.
7. The distinction between "sliding window" and "binning" approaches is unclear. The authors should provide further explanation regarding their implementation.
8. The authors should acknowledge the limitations of methylation data in identifying focal CNV events.
9. The figures provided do not adequately illustrate the methods or results. The authors should improve the visual clarity and explanatory power of all figures.

Minor Comments:

10. Line 205: The statement that CNA burden correlates with aggressive behavior is well-established and could be rephrased or removed.
11. Figure 2b: This figure is visually unclear (error bars and dots are not visible) and methodologically ambiguous. Are all genes within the segments combined?
12. Figure 3b: The co-occurrence plot's scale is confusing. With a cohort size of 365, it is unclear why the scale reaches 3000. This plot requires further clarification.

Reviewer #5

(Remarks to the Author)

The authors have thoroughly and diligently addressed all the concerns raised in the previous round of review, specifically by providing a clearer comparison and integration with existing prognostic models, testing generalization on two external datasets, and outlining plans for further validation before clinical application. The revised manuscript demonstrates substantial improvements, all of which contribute to the overall quality and impact of the work.

Version 2:

Reviewer comments:

Reviewer #4

(Remarks to the Author)

I commend the authors for thoroughly addressing all my questions and concerns. The manuscript has been significantly improved, with better articulation of its scope, findings, and limitations. Additionally, the structural changes have enhanced its overall readability.

My few minor recommendations are as follows:

1. Line 583-584: "Focal regions of loss were selected by manual inspection of regions along the chromosome arm with a notably higher proportion of samples.": The authors claim that "manual" inspection is an established method citing CDKN2A studies. This is quite different as those studies were identifying a single prognostic marker. However, here the authors claim a group of focal chromosomal events with clinical implications, also proposes that this method is potentially applicable to other cancer types. To me, this is a limitation of the method. The authors should emphasize this better in the result and/or discussion section.
2. In Methods Section, RNA-sequencing experimental procedures and analysis methods were summarized under "Focal genomic and ontology analyses" section. They should be detailed under the appropriate title, so that readers with specific interests/questions can locate the appropriate method details.

REVIEWER COMMENTS

Reviewer #1 (Remarks to the Author): *Expert in computational cancer genomics, copy-number alterations, and tumour heterogeneity*

Summary and Significance

The authors have developed an approach to improve the prognostic value of copy-number by systematically optimizing the size thresholds used for inclusion in prognostic criteria. Applying this approach in a cohort of meningioma patient samples, the authors identify several optimized prognostic CNAs, as well as co-occurrent CNAs, that are not used in current prognostic criteria for meningioma. They then extend this approach to optimize size thresholds for CNAs across 9 TCGA cohorts. This approach seems actionable and may improve prognostic criteria, thus providing real clinical value.

Thank you for your thorough and thoughtful review of our manuscript.

Major Limitations

While the core analysis of this study has value, this manuscript, and some aspects of the study, are incomplete. The interpretation of the data is insufficient in many areas and discussion of the context, significance, and limitations of this study are missing. Additionally, while the presence of focal alterations is acknowledged, no attempt is made to determine if these, or any other chromosomal region, or any confounding factors (e.g., subtype), are the source of the prognostic value of size-dependent CNAs.

We are grateful to receive this feedback. As described below, we have now undertaken a major revision of our study to address these limitations and the other helpful suggestions for improvement.

Major Issues

Data Analysis and Validity

1. Do size-optimized CNA improve risk stratification models? How much? This study does not directly answer that question aside from identifying how CNA size thresholds change their prognostic value (Fig 1B). Please provide direct comparison of the prognostic value of these model vs the current standard (integrated grade/score). An example would be to provide the ROC curves for the standard vs optimized models.

Thank you for this suggestion for improvement. A direct comparison requires delineation of training and test sets. In order to more fully examine the performance of models based upon size-dependent CNAs, we now delineate a “**discovery cohort**” (N=200, University of California, San Francisco [UCSF]) and an “**external test cohort 1**” (N=365, University of Hong Kong) of meningiomas within our dataset, and compare the validation performance of the size-dependent CNA models we propose with the performance of previously described molecular classification systems for meningiomas (Integrated grade and Integrated score), with or without size-dependent CNAs, using meningiomas that were not used for the development of any of these models (external test cohort 1). Furthermore, we now include a new independent validation cohort (“**external test cohort 2**”) that is comprised of 126 meningiomas from Baylor College of Medicine (BCM) with available DNA methylation profiling and clinical outcome data. LASSO and Elastic Net regularized Cox regression models are now trained on the discovery cohort of meningiomas to identify CNA predictors. Consistent with our initial observation that flexible size thresholds may more sensitively identify prognostic CNAs, these models identified CNAs that were not included in Integrated grade or Integrated score, such as 1q gain, 7p loss, and 12q loss (Fig. 1c). To generate a clinically applicable risk classification system similar to Integrated grade and Integrated score, we combined the presence of size-dependent, prognostic CNAs with WHO 2016 histological grade, mitotic index, and *CDKN2A/B* status (features which were present in either Integrated score or Integrated grade) in order to generate a size-dependent integrated molecular risk score.

Validation of the size-dependent risk score described above in external test cohort 1 (UHK, N=365) yielded a maximum 5-year AUC for LFFR and OS of 0.83 and 0.78, respectively. In external test cohort 2 (BCM, N=126), the size-dependent risk score achieved a maximum 5-year AUC for LFFR and OS of 0.78 and 0.88, respectively. In the external test cohort 1, likelihood ratio tests demonstrated that the addition of a size-dependent model to Integrated grade significantly improved prediction of LFFR (P=0.0055), but reciprocal addition of Integrated grade to the size-dependent model did not improve prediction of LFFR (P=0.32). Addition of the size-dependent model to Integrated score also improved prediction of LFFR (P=0.00088), and the reciprocal was also true, albeit with a more modest P-value (P=0.013). This same pattern was observed for prediction of OS, where adding the size-dependent model significantly improved performance of Integrated grade (P=7.01e-05), but the reciprocal was not true (P=0.18), and adding the size-dependent model significantly improved performance of Integrated

score ($P=5.0e-06$), and the reciprocal was also true, albeit with a more modest P -value ($P=0.014$). Size-dependent CNA models additionally provided further stratification within low and intermediate Integrated grade or Integrated score risk groups (Extended Data Fig. 4). In multivariate regression combining the size-dependent CNA models with patient sex, EOR, primary versus recurrent meningioma presentation, WHO grade, and adjuvant radiotherapy, the size-dependent CNA model remained significantly prognostic ($P<0.0001$) for LFFR and OS in the external test cohorts (Supplementary Table 3; nomograms generated in Extended Data Fig. 3). When incorporated with Integrated grade or Integrated score in multivariate regressions, the size-dependent CNA model remained independently prognostic in both external test cohorts (Supplementary Table 3).

Performance of Integrated grade and Integrated score were also measured in external test cohort 1 using (1) CNAs defined at the thresholds with which these models were trained (e.g. 50% for Integrated grade and 5% for Integrated score), (2) CNAs defined at the threshold with which these models performed best in our Discovery cohort (17% for Integrated grade and 4% for Integrated score, respectively), and (3) our optimized CNA thresholds for LFFR and OS, which ranged between 1% and 97% of chromosome arms. Integrated grade as a continuous score had a maximum 5-year LFFR AUC of 0.82 in external test cohort 1 when defined using either optimized or original thresholds, and Integrated grade as a discrete score had a maximum AUC of 0.79 when defined using optimized or original thresholds. Continuous and discrete Integrated score had maximum 5-year LFFR AUCs of 0.80 and 0.74, respectively, when using either optimized or original thresholds. As suggested, we now provide ROC curves for our size-dependent CNA risk model, Integrated Grade, and Integrated score when defined at optimized thresholds for 5-year LFFR and OS in Extended Data Fig. 2d. Comparison was not possible in external test cohort 2 as mitotic data were unavailable to determine Integrated grade.

In summary, while the aim of this project was not to propose a clinical risk stratification model to replace existing ones, we do demonstrate that incorporating variable size thresholds can identify prognostic CNAs not captured by other systems without sacrificing predictive performance. Moreover, as a proof of concept, a size-dependent CNA model trained using similar histopathologic and clinical features as Integrated grade or Integrated score resulted in a size-dependent CNA risk score that (1) achieved excellent performance in external test sets, (2) provided statistically significant additional prognostic value when added to Integrated grade or Integrated score, resulting in improved risk stratification within molecular risk groups, and (3) remained independently prognostic in multivariate analysis when incorporated alongside Integrated grade and score.

2. Does subtype influence the prognostic value of size dependent CNAs? Is the contribution of single or co-occurring CNAs to prognosis consistent across a cancer type or are these picking up residual signal from subtype-specific CNAs that may not necessarily be prognostic? This should be explored analytically (and discussed) to the extent possible and will significantly improve the clinical significance of these results.

Thank you for this suggestion. We agree that CNA patterns could differ both between and within cancer subtypes, and the degree to which CNA patterns provide prognostic information independent of subtype is an important question. Although meningioma DNA methylation groups are enriched for characteristic CNAs, we found that size-dependent CNA models combined with WHO grade and extent of resection remained prognostic in external test cohort 1 (HKU, $N=365$) across meningioma DNA methylation groups (dotplot left). In addition, incorporation of DNA methylation groups in a multivariate Cox analysis showed that the CNA model remained independently prognostic for LFFR ($P=0.000136$) and OS ($P=7.24e-08$). We have added these new analyses to the Results section of our revised manuscript.

3. The presence of focal alterations within size dependent CNAs in both datasets raises the question that these may be solely responsible for their prognostic value. This study is expanded to characterize which genes are present in these areas, thus additional analysis of their sole prognostic value is warranted.

Thank you for this insightful comment. To address whether minimally altered regions and areas of focal gains or losses may contain oncogenes or tumor suppressors that could contribute to the prognostic value of broader CNAs containing the same regions, we now use RNA sequencing data from the same meningiomas to more fully interrogate the effects of focal CNAs on gene expression (Extended Data Fig. 5a), and the prognostic value of expression of genes located on focal CNAs (Extended Data Fig. 5b). LASSO Cox regression of the expression of genes located on focal CNAs identified several prognostic candidates, including *NUAK2*, *UBE2C*, *CTSC*, *NEK11*, and *HSPB7* as prognostic for LFFR, and *FANCA*, *UBE2C*, *CACHD1*, and *SYNPO2* as prognostic for OS. Narrowing this analysis to genes located only in focally prognostic regions located on CNAs as determined by focal increases in AUC across 200kb bins (see the Methods section of our revised manuscript for a more detailed description of these analyses) identified *NUAK2* as prognostic for LFFR, and *NUAK2*, *DSP*, and *PRKAR2B* as prognostic for OS. We also now examine the prognostic value of CNAs in a more granular fashion using the sliding window approach suggested in comment #4 below, allowing for finer analysis of specific focal CNAs. We have also edited the text and figure legends of our revised manuscript to mention that HLA is a prominent distinguishing marker that is gained or lost in different DNA methylation groups of meningiomas¹, and the approximate size of this locus (200 Kb) was chosen as the bin size for focal CNA analyses. Please see the response below for more details.

4. Along these lines, focusing on the percentage of a chromosome arm is gained/lost overlooks the potential prognostic value of individual regions of the chromosome as well as the magnitude of the CNA (particularly for gains). For example, consider a whole-chromosome gain, which would be included in both a 5% or 75% threshold. In this setting, would a focal amplification on the same chromosome with real, prognostic value be overlooked? Would the magnitude of this amplification be overlooked? Varying additional inclusion criteria, including using a sliding window across each chromosome (e.g., using only CNAs within a 20Mb bin) or copy number-based thresholds (e.g., using only CNAs with signal intensity above 0.1, 0.2, 0.4, etc.) may help identify the source of the prognostic value (e.g., specific regions or cumulative effect?). This additional analysis would significantly improve this study.

Thank you for these helpful suggestions. In order to examine the prognostic value of CNAs in a more granular fashion, we now divide each chromosome arm into bins of 200 Kb. We have also edited the text and figure legends of our revised manuscript to mention that HLA is a prominent distinguishing marker that is gained or lost in different DNA methylation groups of meningiomas¹, and the approximate size of this locus (200 Kb) was chosen as the bin size for focal CNA analyses. Gain or loss of each bin was inputted into univariate Cox proportional hazards models for 5-year LFFR and OS, and the AUCs were plotted along the chromosome arms (Fig. 2a). Focal regions with prognostic value were defined as those for which the AUC was at least 4% higher than the median for the entire chromosome arm. This threshold was selected based on the magnitude of the AUC peak corresponding to the *CDKN2A/B* locus on chromosome 9p, which is known to be a recurrently lost region with significant prognostic value that is now codified as sufficient for diagnosis of anaplastic WHO grade 3 meningioma.

This new analysis identified focal prognostic regions on chromosomes 2q, 3q, 4p/q, 5p/q, 7p, 8p, 10q, 12q, 13q, and 14q, including a focal peak at the *CDKN2A/B* locus on 9p. Of these, only 3q, 9p, 12q, and 14q were identified as prognostic and size-dependent when using percent-arm-altered thresholds alone rather than the sliding window approach, suggesting that a more granular approach could identify additional focal gains and losses of prognostic significance. In addition, not all regions of focal loss identified by visual inspection of CNA profiles were found to have prognostic significance. Ontology analysis of the genes contained in these combined focal prognostic regions identified enrichment of STAT1 signaling and type I interferon signaling pathways (Fig. 2c), which have previously been implicated in cellular responses to genomic instability. As these focal regions were predominately associated with copy number loss rather than gain, acquisition of focal CNAs could reflect a mechanism of downregulation of immune signaling pathways normally associated with innate response to genomic instability. Indeed, suppression of type I interferon signaling may represent a mechanism of immune evasion in the setting of chromosomal instability², which has been shown to induce a cGas-STING mediated type I interferon response. Moreover, loss of the interferon gene cluster on chromosome 9p has been shown to be linked to reduced T cell infiltrate and poorer prognosis in melanoma and HPV-negative head and neck cancer patients^{3,4}, and we have shown T cell infiltrate is reduced in meningiomas that are resistant to standard interventions⁵. These new analyses are now presented in the Results section of our revised manuscript.

Finally, to address the possibility of focal or broad amplifications beyond gain of a single copy, or focal homozygous loss, we generated sample-wise segment plots of CNAs by chromosome position plotted against the amplitude of gain or loss (below, left, example of 1q and 17q gain). Other than identifying the same regions

of focal loss or gain as shown in Fig. 2a, this analysis did not reveal clear recurrent regions of focal deep deletion or high level amplification, except for the *CDKN2A/B* locus on chromosome 9p, which is expected to be homozygously deleted in some samples, and the *HLA* locus on chromosome 6p. Plotting segment mean intensities for the two chromosome arms most associated with gains, 1q and 17q, revealed no clear patterns of focal amplification (below, left), suggesting that a uniform intensity threshold was sufficient to capture prognostic CNA gains. As an additional sensitivity analysis, we examined whether varying the intensity threshold by 0.05 increments significantly altered the prognostic value of each chromosome arm and found that the optimal intensity threshold for calling CNAs appeared to be our selected thresholds of -0.1 for loss and 0.15 for gain, with diminishing AUC with higher thresholds (below, right). These new analyses have been incorporated into the Results section of our revised manuscript.

5. Discussion of the context, significance, and limitations of this study is missing. This manuscript is incomplete. Thank you for bringing this omission to our attention. We now more fully elaborate on these components in the Discussion section of our revised manuscript.

Minor Issues

Data Analysis and Validity

6. Size-dependent and co-occurring CNAs are identified in about 50% of cancers in TCGA. However, characterization of these (within Figure 3 and the text) is very sparse despite additional data being present in the supplement. Please expand the summary of this dataset past GBM and CESC. For example, what prognostic size dependent CNAs were identified that are or are not included in current prognostic criteria? Do any of these align with known, relatively aggressive subtypes as identified in the literature?

We have significantly expanded and revised our analyses regarding CNAs in TCGA samples. We now present the full list of size-dependent arm-level CNAs across 33 cancer types in the Pan-Cancer Atlas (Fig. 4a) and explore prognostic co-occurrent pairs (Fig. 4b, Supplementary Table 8). We also further contextualize these findings in the text of our revised manuscript, and we thank the reviewer for bringing this opportunity for improvement to our attention. With respect to what is known or novel about these CNAs in current prognostic criteria or the literature, ambiguity of optimal size thresholds to define arm-level CNAs is not an issue limited to meningioma and (to our knowledge) has not been answered across any cancer. Among the cancers with size-dependent prognostic CNAs we identified from TCGA, several have been reported in pan-cancer analyses to have CNAs of prognostic significance, including KIRP, STAD, LIHC, OV, and SKCM⁶. For others such as COAD, OV, BRCA, HNSC, PRAD, CESC, and NLSC, aneuploidy in general is associated with worse outcomes⁷. We

now demonstrate that all but 4 of the 33 cancer types in TCGA have prognostic CNAs that are sensitive to changing size thresholds. As we demonstrated for meningioma, consideration of chromosome-specific thresholds across these 29 cancers may identify prognostic CNAs not previously reported in the literature. Likewise, prognostic CNA co-occurrence patterns are rarely reported in the literature, but several have been identified, including chromosome 1p/19q codeletion in LGG⁸, 1q gain and 16q loss in BRCA⁹, gain of chromosome 7 and loss of chromosome 10 in GBM⁸, loss of 18q and gain of 8q in COAD¹⁰, and loss of 8p and gain of 8q in prostate cancer¹¹. In our new analyses, we identify prognostic co-occurrence CNA pairs for most TCGA cancers. Furthermore, by restricting co-occurrence model inputs to size-dependent CNAs, we identified pairs that differed from the few that were previously reported. While each of the co-occurrent CNA pairs remained significantly prognostic when combined in multivariate analysis with CNA burden (e.g. total number of CNAs), CNA burden was still prognostic for many cancers with size-dependent prognostic CNAs, in accordance with prior pan-cancer investigations¹². Thus, as with meningioma, consideration of specific-size thresholds for defining CNAs identifies prognostic patterns of CNA co-occurrence not previously described for the majority of cancers in TCGA.

7. Line 64 — Gene ontology analyses using genes included in focal deletions is not sufficient to support dysregulation of these pathways. This conclusion oversteps what the analysis can support and should be corrected.

We have amended the text to remove mention of dysregulation, as suggested, and incorporated paired RNA sequencing data (as described above) into the new analyses of our revised manuscript.

8. Line 78 — Hierarchical clustering performed in this way is not sufficient to independently gauge the relative timing of CNAs across individual cancers. While it is encouraging that many early CNAs are captured by your “early” cluster, it is important to note here that the original analysis (Magill et al.), as the authors know, was done in spatially distinct samples within the same tumor. Here, the conclusion that this analysis reveals early vs late CNAs seems overstated and this caveat should be addressed. Inclusion of tumor-grade may bolster this claim.

We have amended the text and figures of our revised manuscript to remove mention of temporality in connection to CNA clusters (Fig. 3b). Instead, the clusters are now labeled numerically, and we show that the CNAs in each cluster are enriched in meningiomas with low, intermediate, or high CNA burden. As suggested, we now include tumor grade in this plot and demonstrate that increased CNA burden correlates with higher WHO grade. Although temporality cannot be gauged on the presence of co-occurrences alone, our clustering analysis does identify some interesting observations. For example, some CNAs become gradually more prevalent as the burden of the total number of CNAs per meningioma increases, whereas other CNAs become common only at the highest levels of CNA burden. In addition, recurrent CNA gains that have been previously observed in meningiomas, such as loss of chromosome 1q or 20q, appear to occur predominately in samples with the highest burden of CNAs.

9. The first paragraph of the main section refers to CNA signatures as a “marker of chromosome instability”. CNA signatures alone are not a reliable measure of CIN, but are a measure of aneuploidy, the product of CIN.

We have amended the text throughout our revised manuscript to reflect CNAs as markers of aneuploidy rather than chromosomal instability. We thank the reviewer for bringing this to our attention.

10. It is not immediately evident that AUC is used to assess the quality of the model rather than an orthogonal metric for survival. Please clarify this in the text.

We have now clarified in the text that AUC for 5-year LFFR and OS are the primary measures of prognostic value across all models in our revised study.

11. Please clarify the rationale of the chosen criterion for size dependent CNAs: “a maximum AUC for 5-year LFFR or OS of at least 0.6 that decreased by at least 5% from the maximum AUC as CNV threshold varied”.

We appreciate the need to justify the definition of size-dependence, as it is a central component of our manuscript. Instead of setting an arbitrary drop-off in AUC of 5%, we now have quantified the variance in AUC across thresholds using the standard deviation. To maintain sensitivity in selecting potential CNA inputs for predictive models, a standard deviation of 0.01 was used as criteria for size-dependence among CNAs with a prognostic AUC (at least 0.60) at any threshold. Additionally, only CNAs present in at least 2.5% of samples were considered size-dependent for purposes of analyses. As a comparison, extent of resection, which is a well described prognostic variable for meningioma, achieved a 5-year LFFR AUC of 0.59, and we now provide this

information in the text of our revised manuscript. With this comparison in mind, a univariate AUC of ~0.60 was felt to be a reasonable benchmark for a prognostic variable. The cutoff of a standard deviation of 0.01 was chosen by estimating the false discovery rate of the standard deviation of the AUC across size thresholds using both permutation and bootstrap methods. The sample-wise percent-arm altered was either permuted without replacement or resampled with replacement (bootstrap) within each chromosome arm, maintaining the chromosome arm specific background distribution, but scrambling the relationship between percent-arm alteration and sample. This was repeated 500 times to generate a distribution of standard deviations which serves as an estimate of the false discovery rate (FDR). The standard deviation cutoff of 0.01 corresponded to a 5.3% and 5.4% FDR by permutation or by bootstrap, respectively. A cutoff of presence in 2.5% of samples was used in order to exclude rare CNAs with only a handful of samples with events, to improve the stability of our analyses. We have added descriptions of these analyses to the Methods section of our revised manuscript.

12. Please provide one or more examples of ROC curves from which AUC was calculated and how these curves change with size thresholds.

We have provided ROC curves for performance of our size-dependent CNA models (using CNAs defined at optimal thresholds), Integrated grade (at a threshold of 50%), and Integrated score (at a threshold of 5%) for external test cohort 1 in Extended Data Fig. 2d of our revised manuscript. Comparative AUC analyses from these curves are provided in the Results section of our revised manuscript. Additionally, provided below are ROC curves in which all models were defined at thresholds of 5% (left), 50% (middle), or our optimized thresholds (right) to predict LFFR (top) or OS (bottom).

13. Figure 1F (and Extended Data Figure 4) — The interpretability of these figures is limited by exclusion of the names of the sub-processes involved (blue nodes). Please provide a key.

In order to make the ontology analysis more interpretable, we have converted the results of these analyses into a series of dot plots (Fig. 2c). These were generated from RNA sequencing data for both the genes contained in all regions of recurrent gain/loss, as well as the prognostic regions identified on focal AUC plots, regardless of CNA size-dependence.

14. *Extended Data Figure 1 — As presented, the scales for each plot are different causing some chromosomes to appear larger or smaller. Either show the same chromosomes for each sample or provide the x-axis scale. Also please consider an alternative color scheme as green/red on a grey background is not accessible for some.*

We have revised Extended Data Fig. 1 to show the same chromosome arms for all three samples and have revised the color scheme to be more accessible.

15. *Extended Data Figure 2 — Interpretation of these results is lacking. Also, there appear to be data off the scale in Extended Data Figure 2b which need to be explained and/or the scale needs to be expanded.*

We have added text referencing the coefficient and calibration plots, which demonstrate that the models were well calibrated and identified a variety of important CNA predictors. The data off of the scale in the calibration plots are histograms showing the count of samples falling along the x-axis of predicted values. We have amended the legend for the figures to reflect this.

16. *Please comment on potential sources of differences between CNAs that are prognostic in OS vs LFFR and those identified exclusively using the LASSO or elastic net models. This consensus approach, using two methods has value but should be examined accordingly.*

During the course of this revision, we discovered that some of the CNAs driving our initial models were rare and may have skewed prediction of either LFFR or OS. To address this, we have now limited model inputs to CNAs that (1) were present in at least 2.5% of samples, (2) were size-dependent (standard deviation for AUC of 0.01), and (3) were prognostic (maximum AUC of at least 0.60). These refined analyses improved concordance across models. It is difficult to know why a CNA may be important for LFFR or OS other than through speculation about genes that may be affected, and the importance of response to treatment after local recurrence. When comparing LASSO to Elastic Net regularization, Elastic net tends to identify more predictors as it places less emphasis on identifying a sparse model, and we have made note of this issue in the Discussion section of our revised manuscript. We also generated the requested consensus models in which only size-dependent CNA predictors from both LASSO and Elastic Net models were used (now shown in Fig. 1c), which achieved AUCs for 5-year LFFR and OS of 0.77 and 0.67, respectively, in external test cohort 1, and 0.74 and 0.70, respectively in external cohort 2. Furthermore, we used gradient boosting methods incorporating predictors identified through both methods, which demonstrated superior performance, reported now in the Results section of our revised manuscript.

17. *Though citation of reference #21 is encouraged as it is relevant to this study (e.g., does this study identify the same novel prognostic CNAs?), it appears to be cited incorrectly in the text in reference to the PanCancer Atlas. Please provide the correct citation for the PanCancer Atlas in place of this.*

We have amended the text to show the correct citation for the PanCancer Atlas in this area.

18. *Copy number changes, as described in this manuscript, are more accurately named “copy number alterations” (CNAs) rather than “copy number variants” (CNVs). See PMID: 23412801.*

We have amended the text and figures of our revised manuscript to state copy number alteration (CNA) in place of copy number variant (CNV) throughout.

Reviewer #2 (Remarks to the Author): *Expert in computational cancer genomics, copy-number alterations, tumour heterogeneity, and statistics; co-reviewed with Reviewer #3*

In their article “Pan-cancer copy number variant analysis identifies optimized size thresholds and co-occurrence models for individualized risk-stratification”, Nguyen and Chen et al. present an analysis of meningiomas whereby they test the hypothesis that copy-number variants (CNVs) can have an optimal size threshold for prognostic power and then apply this analysis to The Cancer Genome Atlas (TCGA) cohort to uncover similar cancer-specific size-thresholds.

CNVs are a hallmark of many cancer types and using them as leverage for prognostic power has been an active area of research for many years. However, the difficulty in the problem comes from the fact that whole chromosome arms are rarely fully intact, and instead the amplified or deleted regions tend to be more fragmented, so the question of how we quantify the amount of DNA gained or lost is somewhat sensitive to the resolution one uses. As such, the paper is concerned with an important problem that can have significant clinical implications. However, I have some major concerns about the paper in its current form that should be addressed before deciding about publication.

I have broken down my major comments into categories corresponding to the three main analyses of the paper.

Thank you for your thorough and thoughtful review of our manuscript. As described below, we have now undertaken a major revision of our study that incorporates your helpful suggestions for improvement.

Major:

General Remarks:

1. I have concerns about the paper not including any of the analyses performed. The authors state that they did not develop any software packages or tools in the “Code Availability” section, but clearly their computational analyses are extensive and as such they must have produced some code. In the spirit of transparency and reproducibility, it is vital that the authors include the analysis scripts. As a bare minimum, the code to reproduce the figures in the paper should be included.

2. Similarly, the most important derived datasets must be made publicly available, in particular the copy-number profiles of the meningiomas. Since CNVs are non-identifiable data I would recommend uploading them together with the source code (see point 1 above) to Zenodo.

Thank you for these suggestions. We have uploaded relevant scripts used for the core analyses of this manuscript, along with meningioma copy number segmentation data for our cohort of 565 meningiomas (<https://zenodo.org/doi/10.5281/zenodo.11501566>, <https://zenodo.org/doi/10.5281/zenodo.11501381>). Raw methylation IDAT files for our cohort have been previously published and are available in the NCBI Gene Expression Omnibus under accession number [GSE183656](https://www.ncbi.nlm.nih.gov/geo/query/acc.cgi?acc=GSE183656). IDAT files for external validation cohort 1 are published under accession number [GSE189673](https://www.ncbi.nlm.nih.gov/geo/query/acc.cgi?acc=GSE189673), and IDAT files for external validation cohort 2 are published under access number [GSE189521](https://www.ncbi.nlm.nih.gov/geo/query/acc.cgi?acc=GSE189521). Pan-Cancer clinical and copy number data are available at <https://gdc.cancer.gov/about-data/publications/pancanatlas>. The Data Availability and Code Availability sections of our revised manuscript now include these links.

3. The main body of the article is extremely dense. There is barely any introduction and as such the work is not thoroughly placed in the existing literature. There is little to no discussion of their intermediate results and the summary paragraph feels somewhat short and underdeveloped. Where does their work stand in the field? Do they have any suggestions for future work or for further research directions?

Thank you for this suggestion for improvement. We now more fully use the allowable word count to elaborate on these topics in the text of our revised manuscript.

4. Starting with the TCGA analysis (l.85 ff), there are virtually no references to existing literature. Are the TCGA results novel? Also for the analysis of GBM and CESC samples, there are zero references to the previously-found features of gains and losses (arm-level or otherwise). It is again unclear if the authors’ work is entirely novel or if they have recapitulated previous (unmentioned) results.

We have significantly expanded and revised our analyses regarding CNAs in TCGA samples. We now present the full list of size-dependent arm-level CNAs across 33 cancer types in the Pan-Cancer Atlas (Fig. 4a) and explore prognostic co-occurrent pairs (Fig. 4b, Supplementary Table 8). We also further contextualize these findings in the text of our revised manuscript, and we thank the reviewer for bringing this opportunity for improvement to our attention. In brief, we find that overlap between our identified size-dependent cancer types and those with reports of either prognostic CNAs or aneuploidy^{6,7}. As in the case of meningioma, there remains no consensus on optimal size thresholds for arm-level CNAs to predict clinical outcomes across cancers. Prognostic CNA co-occurrences are described in the literature for a number of cancer types, including chromosome 1p/19q codeletion in LGG⁸, 1q gain and 16q loss in BRCA⁹, gain of chromosome 7 and loss of chromosome 10 in GBM⁸, loss of 18q and gain of 8q in COAD¹⁰, and loss of 8p and gain of 8q in prostate cancer¹¹. Using our size-dependent approach, we identified pairs that differed from the few previously reported.

While each of the co-occurrent CNA pairs remained significantly prognostic when combined in multivariate analysis with CNA burden (e.g. the total number of CNAs per cancer), CNA burden was still prognostic for many size-dependent cancers, in accordance with prior pan-cancer investigations¹². Thus, as with meningioma, consideration of specific size thresholds for defining CNAs identifies prognostic patterns of CNA co-occurrence not previously described for the majority of cancers in TCGA.

Size-dependent CNAs:

5. Fig. 1a demonstrates the prognostic capabilities of individual gains and losses for local freedom from recurrence (LFFR) and overall survival (OS), where the authors claim there are size-dependent CNA thresholds. There are a number of boxes highlighting specific thresholds on a subset of chromosome arms. Given the text and the figure caption, it appeared at first glance that there are no other size-dependent CNAs (though Supp. Table 1 contains a list of the “Optimal CNV threshold[s]”). Are the size-dependent CNVs the same as the optimal CNV thresholds? If so, why not highlight all such optima in Fig. 1a?

Thank you for this comment on the interpretability of the heatmaps in Fig. 1. This was an oversight in the original submission. We now highlight all size-dependent prognostic CNAs using an asterisk located at the location of the optima, as suggested.

6. Further to Fig. 1a, the authors are attempting to understand the size-dependence of CNAs, but does this not depend on the location or contiguity of the CNA? This is hinted at by the authors’ noting that there are focal amplifications and deletions in a number of chromosomes (Fig. 1d). The authors do not motivate why they only consider size in this analysis and not where CNAs can occur. I would expect that a priori 5% gain in areas of high density of oncogenes is (presumably) not the same as 5% gain dispersed across areas of no major function.

Thank you for this insightful comment. We now address this in two ways: (1) we identify size-dependent prognostic CNAs using a finer bin size of 1% per chromosome arm to capture small CNA events, and (2) we include a new analysis examining size, location, and contiguity of CNAs with finer granularity and specificity by dividing each chromosome arm into bins of 200 Kb. We have also edited the text and figure legends of our revised manuscript to mention that *HLA* is a prominent distinguishing marker that is gained or lost in different DNA methylation groups of meningiomas¹, and that the approximate size of this locus (200 Kb) was chosen as the bin size for focal CNA analyses. Gain or loss of each bin was used for univariate Cox proportional hazards models for 5-year LFFR or OS, and AUCs were plotted along chromosome arms (Fig. 2a). Focal regions with prognostic value were defined as those for which the AUC was at least 4% higher than the median for the entire chromosome arm. This threshold was selected based on the magnitude of the AUC peak corresponding to the *CDKN2A/B* locus on chromosome 9p, which is known to be a recurrently lost region with significant prognostic value that is now codified as sufficient for diagnosis of anaplastic WHO grade 3 meningioma.

These new analyses identified focal prognostic regions on chromosomes 2q, 3q, 4p/q, 5p/q, 7p, 8p, 10q, 12q, 13q, and 14q, including a focal peak at the *CDKN2A/B* locus on 9p. Of these, only 3q, 9p, 12q, and 14q were identified as prognostic and size-dependent when using percent-arm-altered thresholds alone rather than the sliding window approach, suggesting that a more granular approach could identify additional focal gains and losses of prognostic significance. Such a granular approach may be too complex to be clinically practical, as compared to use of a percent-arm-altered threshold, which we now mention in the Discussion section of our revised manuscript.

Interestingly, and as anticipated by the reviewer, not all regions of focal loss identified by visual inspection of CNA profiles were found to have prognostic significance. Ontology analysis of the genes contained in these combined focal prognostic regions identified enrichment of STAT1 signaling and type I interferon signaling pathways (Fig. 2c), which have previously been implicated in cellular responses to genomic instability. As these focal regions were predominately associated with copy number loss rather than gain, acquisition of focal CNAs could reflect a mechanism of downregulation of immune signaling pathways normally associated with innate response to genomic instability. Indeed, suppression of type I interferon signaling may represent a mechanism of immune evasion in the setting of chromosomal instability², which has been shown to induce a cGas-STING mediated type I interferon response. Moreover, loss of the interferon gene cluster on chromosome 9p has been shown to be linked to reduced T cell infiltrate and poorer prognosis in melanoma and HPV-negative head and neck cancer patients^{3,4}, and we have shown T cell infiltrate is reduced in meningiomas that are resistant to standard interventions⁵. These new analyses are now presented in the Results section of our revised manuscript.

7. The summary of Fig. 1a to me is that the prognostic power of any single CNV gain or loss on a single chromosome is not particularly good (with the exception of an extremely small select subset). Why do the authors

choose an AUC of ≥ 0.6 as the benchmark? What happens to the analysis if the benchmark is set higher? Also, the area under what curve was even used for the AUC? Precision/recall? Receiver/operator? From the name of the R package I assume the latter, but this needs to be stated explicitly. Further, assuming a ROC curve, an AUC of around 0.6 is not particularly impressive, given that an AUC of 0.5 is no better than random guessing.

We have amended the text to clarify that the area under the curve refers to a time-dependent receiver operating characteristic curve and provided as figures below the ROC curves for our size-dependent CNA models across different thresholds. We selected this AUC cutoff, along with our standard deviation cutoff, in order to balance sensitivity to any potential prognostic size-dependent CNA to include in our models. As a comparison, extent of resection, which is a well described prognostic variable for meningioma, achieved a 5-year LFFR AUC of 0.59, and we now provide this information in the text of our revised manuscript. Thus, a univariate AUC of ~ 0.60 was felt to be a reasonable benchmark for a prognostic variable. When CNAs were combined with additional variables within a larger prognostic model, the resulting model was expected to improve significantly in performance. In support of this hypothesis, our combined CNA + WHO grade + mitoses + CDKN2A/B loss models demonstrated AUCs that were more robust in predicting LFFR (0.83) and OS (0.78). At the same time, use of a higher AUC threshold may prove too high of a benchmark: use of a cutoff of 0.65, for example, results in only 2 prognostic size-dependent CNAs for LFFR and OS, respectively. We have added a description of these analyses and findings to the Methods section of our revised manuscript, and we thank the reviewer for this helpful suggestion for improvement.

8. Why are the thresholds (1.38 / Fig 1a) varied at 5% intervals? There is no justification in the text for this. How stable are the optimal CNV thresholds with respect to the threshold step size? What happens if one picks e.g. 8% or 13% thresholds? Are the specific optima used in the paper simply artifacts of the choice of bin size?

Thank you for raising this point. We have now re-performed this analysis using 1% intervals to improve resolution (Fig. 1a). The optimal thresholds are provided in Supplementary Table 1. Use of 1% intervals (plot

below, left) resulted in similar appearance of AUC distribution by size threshold across arms (Fig. 1a) as 5% (plot below, right). Use of 8% or 13% thresholds also resulted in similar appearance of AUC distribution, albeit with less granularity. As expected, the optima were minimally impacted by the use of a more granular step size of 1% compared to 5% (median change in % threshold of 3%, interquartile range 1.5%-5.5%). We have added a description of these analyses and findings to the Results section of our revised manuscript.

9. Further to this point, for the 6q losses, there is a double peak from what I can tell: a peak at the $\geq 5\%$ threshold (the size-dependent CNA), and then a peak for a pair of bins (at $\geq 25-35\%$ for LFFR and $\geq 15-25\%$ for OS). The authors do not comment on the significance of this, and could it be attributed to the fact that the analysis is binned in 5% increments?

Thank you for pointing this out. This double peak was an error in the generation of this graphic and has been corrected in our revised manuscript (Fig. 1a). We are grateful to the reviewer for bringing this error to our attention.

10. The authors state the performance (i.e. the AUC) for the LASSO and elastic net models for LFFR and OS (1.58 ff.), but do not comment on the fact that these measures seem to have roughly the same performance as the conventional integrated grade and score (judging from Fig 1b). The authors need to include a thorough comparison between the integrated grade and score model and their new model based on optimized CNV thresholds, to underpin the statements made in the title and the rest of the paper. They do note that their trained models identify some chromosome arms excluded by the conventional methods, but do not comment on the fact that some of these arms (such as 5p/q) showed poor individual prognostic power. To me, this could actually make a stronger case for the second part of their analysis: the co-occurrence of certain “size-dependent” CNVs.

Thank you for this helpful suggestion for improvement. As is also described in detail in our response to Reviewer 1 Comment #1, we have improved our model-building and validation workflow to more consistently identify CNAs with individual prognostic value and provided direct comparisons of our size-dependent CNA models to Integrated grade and Integrated score. Model performances using only CNA data were either improved or similar in our analyses, although as before, our size-dependent CNA models identified multiple prognostic CNAs that were not included in other models. While some of the CNAs identified by our models did display marginal individual prognostic power (e.g. 12p loss), the same can also be said of CNAs included in the existing Integrated grade (3p loss, 19p/q loss). We entirely agree with the reviewer that these observations further motivate our subsequent analyses of co-occurrent CNAs and have provided this rationale in the text of our revised manuscript. We have also provided additional exposition of these nuances in the Discussion section of our revised manuscript.

To address the point the reviewer raises regarding a thorough comparison of our approach with existing classification systems, we now also use size-dependent CNAs to develop a comprehensive prognostic model that compares favorably to Integrated grade and Integrated score, and which provides additional prognostic information when added to these models. Moreover, we now include a completely new external test cohort 2

(Table 2) that is comprised of 126 meningiomas with clinical data from Baylor College of Medicine (BCM). In brief, to generate a clinically applicable risk classification system similar to Integrated grade and Integrated score, we combined the presence of size-dependent, prognostic CNAs with WHO 2016 histological grade, mitotic index, and *CDKN2A/B* status -- features which were present in either Integrated score or Integrated grade -- in order to generate a size-dependent integrated molecular risk score.

Validation of the size-dependent risk score described above (which was developed using meningiomas from UCSF) in our external test cohort 1 (UHK, N=365) yielded a maximum 5-year AUC for LFFR and OS of 0.83 and 0.78, respectively. In external test cohort 2 (BCM, N=126), the size-dependent risk score achieved a maximum 5-year AUC for LFFR and OS of 0.78 and 0.88, respectively. In the external test cohort 1, likelihood ratio tests demonstrated that the addition of a size-dependent model to Integrated grade significantly improved prediction of LFFR ($P=0.0055$), but reciprocal addition of Integrated grade to the size-dependent model did not improve prediction of LFFR ($P=0.32$). Addition of the size-dependent model to Integrated score also improved prediction of LFFR ($P=0.00088$), and the reciprocal was also true, albeit with a more modest P-value ($P=0.013$). This same pattern was observed for prediction of OS, where adding the size-dependent model significantly improved performance of Integrated grade ($P=7.01e-05$), but the reciprocal was not true ($P=0.18$), and adding the size-dependent model significantly improved performance of Integrated score ($P=5.0e-06$), and the reciprocal was also true, albeit with a more modest P-value ($P=0.014$). Size-dependent CNA models additionally provided further stratification within the low- and intermediate-risk groups as identified by Integrated grade or Integrated score (Extended Data Fig. 4). In multivariate regression combining the size-dependent CNA models with patient sex, EOR, primary versus recurrent meningioma presentation, WHO grade, and adjuvant radiotherapy, the size-dependent CNA risk scores remained significantly prognostic ($P<0.0001$) for LFFR and OS in the external test cohorts (Supplementary Table 3; nomograms generated in Extended Data Fig. 3). When incorporated with Integrated grade or Integrated score in multivariate regressions, the size-dependent CNA risk score remained independently prognostic in both external test cohorts (Supplementary Table 3).

Co-occurrence in meningiomas:

11. *The size-dependent CNA co-occurrence analysis (1.67 ff / Fig 2) is interesting, though it is unclear if this is novel or a recapitulation of previous results found in the literature. If the results are novel, the authors may benefit from stating this more strongly. I would also like to know the predictive power of non-optimal CNA thresholds for both 1p/22q and 9p/14q curves. Does the size-dependence still have a significant effect as the authors claim?*

12. *The authors show the impressive results for 1p/22q and 9p/14q (Fig 2b), which are the predominant components of the LASSO and elastic net models for LFFR and OS (Extended data figure 3a). However, the authors make no mention of another relatively high weighted component: 22q/9p. Why is this absent?*

Our revised analyses (described in greater detail in response to previous comments) identified 1p/22q loss, 1q gain/22q loss, and 9p/18q loss as highly prognostic for LFFR. To our knowledge, this is the first investigation of co-occurrent pairs of copy number alterations in meningioma, and we have added this information to the text of our revised manuscript, as suggested. We also have provided Kaplan-Meier curves for these pairs across different thresholds (below: left = optimized thresholds, middle = 5%, right = 80% threshold). The c-index varied somewhat among these thresholds, from 0.70 (1p/22q loss), 0.61 (1q gain/22q loss), and 0.64 (9p/18q loss) for optimized thresholds, to 0.69 (1p/22 loss), 0.61 (1q gain/22q loss), and 0.62 (9p/18q loss) for 5%, and 0.62 (1p/22q loss), 0.59 (1q gain/22q loss), and 0.60 (9p/18q loss) for 80%. Notably, the individual chromosome arms identified in these pairs were not the novel CNAs identified using size-dependent thresholds. Thus, the magnitude of effect of size-dependence may be smaller in comparison to the large effect of the co-occurrences in question. Nevertheless, the findings we present are consistent with the ideas that both size-dependence and co-occurrence of CNAs can have significant prognostic impact, and that this impact can be context dependent. These new analyses are described in the Results section and shown in Fig. 3a, Extended Data 6a, b, and Supplementary Table 6 of our revised manuscript, and we thank the reviewer for inviting these new analyses.

13. The authors do not at all discuss the differences between co-occurrent CNA pairs for LFFR and OS and the two models used (LASSO vs Elastic net, Extended data figure 3a) For example, for LASSO LFFR vs OS, there are a number of pairs for the OS model that do not appear in the LFFR model. Similarly with the elastic net model. Can you posit any reason as to why this would be?

Thank you for bringing this area for improvement to our attention. In our revised analyses, we also identified co-occurrent pairs that were important for OS, namely 4q/13q loss, 12q/14q loss, and 6q/12q loss. As before, these represent different pairs than those identified for LFFR. Unfortunately, despite being present in at least 2.5% of our cohorts, these pairs were rare overall and risk stratification became difficult, especially at higher thresholds. It is possible that these rarer pairs occurred primarily in patients with shorter OS, thus leading models to overestimate their importance. Furthermore, the overall low rate of death in our cohorts and in meningioma cohorts more broadly is a challenge to an analysis involving so many possible predictors without a higher sample size. We have added text mentioning these explanations and limitations to the Discussion section of our revised manuscript.

14. The number of clusters used in the unsupervised clustering is 3 for both LFFR and OS (the evidence the authors use is the “elbow plot” in Extended Data Fig. 3b and d). But why have they picked 3 and not 4? Three clusters already demonstrate some level of distortion of the score. How does the hierarchy look if there are instead 4 clusters and how drastically does the heatmap of Fig. 2c change? If there are no significant changes with 4 clusters, then I would side with the authors and choose the lesser number, but I have no current a priori reason for seeing that it should be 3, other than the authors arguing for “early”, “late”, and “miscellaneous” CNAs.

We now provide heatmaps based on 4 clusters instead of 3 in Extended Data Fig. 7d. The primary change identified with 4 clusters in Extended Data Fig. 7d compared to 3 clusters in Fig. 3b is the separation of 22q loss when thresholds are optimized for LFFR and both 1p and 22q loss when optimized for OS. These CNAs are the two most abundant at low levels of CNA burden. The clusters appear otherwise similar, and as mentioned in the following response, an alternative approach for clustering also identified 3 clusters in our data.

15. A more general point to the k-means clustering: The method has a number of large drawbacks, for example, the assumption that the underlying data can be clustered into separable spherical clusters, and clusters should be of a similar size. Are these assumptions fulfilled? How sensitive is the data to the choice of clustering method? Have the authors tried alternatives, such as DBScan? Is there any other particular reason for using k-means clustering?

Thank you for the suggestion. We have implemented DBScan as an alternative means of clustering, which also identified 3 clusters (Extended Data Fig. 7c).

CNV size-dependence and co-occurrence in other cancers:

16. The finding that some cancers have “size-dependent CNAs” while others don’t (1.85 ff) is interesting, but the authors do not discuss potential reasons for this. In general, there is a lack of discussion about how the authors’ results fit into the current state of the field.

We have used the expanded word count constraints in our revised manuscript to better review the current literature and explore the limitations and implications of this work. In the Discussion of our revised manuscript we now review the overlap between our identified size-dependent cancer types and those with reports of either prognostic CNAs or aneuploidy^{6,7}. As in the case of meningioma, there remains no consensus on optimal size thresholds for defining CNAs to predict clinical outcomes across cancers. Prognostic CNA co-occurrences are described in the literature for a number of cancer types, chromosome 1p/19q codeletion in LGG⁸, 1q gain and 16q loss in BRCA⁹, gain of chromosome 7 and loss of chromosome 10 in GBM⁸, loss of 18q and gain of 8q in COAD¹⁰, and loss of 8p and gain of 8q in prostate cancer¹¹. However, our analyses identify prognostic CNA co-occurrence patterns across most TCGA cancers. We also identify pairs that differed from the few that have been previously reported. While each of the co-occurrent CNA pairs in our study remained significantly prognostic when combined in multivariate analysis with CNA burden (e.g. the total number of CNAs per cancer), CNA burden was still prognostic for many size-dependent cancers, in accordance with prior pan-cancer investigations¹². Thus, as with meningioma, consideration of specific size thresholds for defining CNAs identifies prognostic patterns of CNA co-occurrence not previously described for the majority of cancers in TCGA. We thank the reviewer for inviting these additions to our revised manuscript.

17. There is no mention of two quirky features in the co-occurrence analysis (Fig. 3c): 1. in GBM, 10q loss + 19q gain has a negative coefficient, and 2. CESC only has 2 components, much fewer than the other LASSO models the authors have shown in the rest of the paper. What does the Kaplan-Meier curve look like for 10q loss + 19q gain in GBM? Do the authors have any reason for CESC being well-described by just two covariants?

As part of the expanded Pan-Cancer analysis we performed for this revision, we have limited CNA inputs to co-occurrence models to those that meet our three criteria for size-dependence, as described above: (1) AUC

of at least 0.6 at any threshold, (2) standard deviation of AUC across thresholds of at least 0.01, and (3) presence in at least 2.5% of samples. The number of CNAs meeting these standardized criteria varied considerably across cancer types (Fig. 4a), and cancer types with fewer size-dependent CNAs may have sparser co-occurrence models. In our revised analyses, we do identify a number of co-occurrent pairs across cancer types that carry a negative coefficient, suggesting an association with better outcomes (Supplementary Table 8). None of these have been reported in the literature, to our knowledge, and we have clarified this in the text of our revised manuscript. Moreover, 10q loss and 19q gain in GBM showed worse outcomes compared to having both arms intact, but no difference when compared to the individual events (Kaplan-Meier curves below). In our revised analyses, CESC has several covariants predicting PFS and OS (plots below), and we describe differences in size-dependent CNAs and cooccurrence patterns in histological subtypes within this cancer in the Results section of our revised manuscript.

Minor:

1. In Fig. 1a, the heatmaps should have the same scale for all 4 plots, or at the very least the same scale for gains and for losses.

The revised Fig. 1a heatmaps now have the same scale for gains and for losses.

2. Fig. 1b: The effects of varying CNA thresholds on the integrated grade and integrated score are interesting, but then the authors do not comment on the fact that the integrated score is maximized at its conventional score.

In our revised analyses, Integrated score's optimized threshold (4%) was still quite similar to its conventional training threshold (5%). In direct comparisons across thresholds in external test cohort 1, there was minor improvement in performance at a threshold of 50%, but this difference was not statistically significant (see Fig. 1b as well as the Results section of our revised manuscript). Similar findings were seen for Integrated grade. As a result, we have rephrased our results and conclusions away from stating that either system improves with optimized thresholds, and instead focus on the observation that optimized thresholds can be used to build a model with favorable comparative performance and different CNAs of importance. We also provide additional descriptions of this observation in the Discussion of our revised manuscript, which we have provided below for ease of evaluation:

“Our finding that each of these systems achieved optimal performance using CNA size thresholds close to the values used for training, and that performance degraded with variation of size thresholds, underscores the sensitivity of CNA models to size thresholds. At the level of model training, we find that size thresholds determine which CNAs form model inputs. Thus, the selection of CNA size thresholds represents a crucial branch point that influences the ultimate characteristics of resulting prognostic models. While Integrated grade and Integrated score incorporate overlapping CNAs, the optimal CNA size threshold and performance across thresholds varies significantly between these systems. These models each use uniform CNA size thresholds across chromosomes, but our results indicate that the optimal CNA size threshold can vary from chromosome to chromosome. By allowing for chromosome-specific variation in CNA size thresholds, we show that additional prognostic CNAs can be identified, and that a size-dependent CNA model may provide additional prognostic information in comparison to, or in combination with, Integrated grade and Integrated score.”

3. What does Fig. 1d have to do with the LASSO and elastic net models? The order to the chromosomes is also odd. Why not order the plots by chromosome number? Why is chr20 included in the subfigure (while not being covariate in the models), but chr1 isn't? Why does chr6 have the text "HLA" inscribed, though this is never mentioned in the text?

To improve the clarity of our revised manuscript, we now separate the model-building (Fig. 1) and focality (Fig. 2) portions of our analyses into separate figures. We have also fixed the order of focal loss/gain plots to be by chromosome number, and now show all chromosomes with either a focally recurrent region of loss/gain or a focally prognostic region. Focal regions with prognostic value were defined as those for which the AUC was at least 4% higher than the median for the entire chromosome arm. This threshold was chosen based on the magnitude of the AUC peak corresponding to the *CDKN2A/B* locus, which is well known to be a recurrently lost region with significant prognostic value for meningioma and is now codified as sufficient for diagnosis of anaplastic meningioma, WHO grade 3, by the World Health Organization. We have also edited the text and figure legends of our revised manuscript to mention that HLA is a prominent distinguishing marker that is gained or lost in different DNA methylation groups of meningiomas¹, and the approximate size of this locus (200 Kb) was chosen as the bin size for focal analysis.

4. L.68: "Regularized Cox regression models using co-occurrent CNV pairs identified..." Are all possible pairs of co-occurrent CNV pairs used as possible hazard inputs?

We have clarified in the text of our revised manuscript that co-occurrence model inputs were limited to any pair of CNAs that contained at least one of 13 CNAs from our size-dependent models. These were CNAs that had an individual maximum AUC of at least 0.60, a standard of deviation for AUC across thresholds of at least 0.01 and were present in at least 2.5% of samples (as described in greater detail above). Co-occurrent pair inputs were also limited to those present in at least 2.5% of samples. All possible pairs of co-current CNAs meeting these criteria were included in regularized Cox regression models.

5. L.96: Is CESC the abbreviation used for both subtypes of cervical cancer? Why are both included into the same dataset?

CESC is the abbreviation for both cervical squamous cell carcinoma and endocervical adenocarcinoma, and these are reported jointly in the Pan-Cancer Atlas. In the analyses for our revision, we have separated these two based on available histological information and have also made note of important molecular subtypes within the 33 overarching cancer types in the Pan-Cancer Atlas (Supplementary Table 9). Unfortunately, molecular markers to further distinguish molecular subgroups of individual cancer types is not provided in the Pan-Cancer Atlas, but we address this limitation by analyzing CNAs across molecular groups of meningiomas in our revised manuscript.

6. L.147: In the section "CNV analysis", why are losses and gains defined with a different absolute mean intensity value? Is this conventional?

This comment highlights an important issue in CNA analysis in meningioma and other cancers that is a central message in our manuscript: there currently is no conventional method for defining CNAs. In our manuscript we use commonly available workflows, including the SeSAME package¹³ for DNA methylation profiling data, which uses the circular binary segmentation algorithm to derive segments from methylation probes, and the available SNP based circular binary segmentation data from the PanCancerAtlas¹⁴. We have previously shown that CNA calls using DNA methylation profiling data and the SeSAME package are 99.12% concordant with CNA calls from whole exome sequencing¹. However, utilization of the output of segmentation from SeSAME and SNP pipelines diverges in methodology depending on the analysis being conducted, and the intensity thresholds used to call CNAs from the resulting segments in systems such as Integrated grade or Integrated score are not always reported^{15,16}. Our intensity value thresholds were selected based on manual inspection for local minima among the distribution of mean values across all segments, as described in a previous publication¹ (Density plot below). These values were relatively stable across meningiomas. In addition, we examined whether varying the intensity threshold by 0.05 increments significantly altered the prognostic value of each chromosome arm and found no clear optimal intensity threshold for calling any arm-level CNA (Heatmaps below). Similar manual inspection was performed for the SNP-array data used to generate CNA results from the PanCancerAtlas. As a sensitivity analysis, we repeated the process of calling CNAs using sliding intensity value thresholds for gain and loss and did not identify an intensity that significantly improved prognostic value of individual CNAs. These new analyses are now described in the Methods section of our revised manuscript. We thank the reviewer for inviting these helpful additions to our revised study.

7. In Supplementary Table 5, what does the value “Preserved” mean for 20p loss in “size-dependent cancers”? Do the authors mean “Not Lost”?

We apologize for this error, which was a mistake in the creation of these tables. We mean to convey that 20p was not lost, and we have corrected this error in our revised manuscript.

Reviewer #3 (Remarks to the Author): Expert in computational cancer genomics, copy-number alterations, tumour heterogeneity, and statistics; co-reviewed with Reviewer #2

I co-reviewed this manuscript with one of the reviewers who provided the listed reports. This is part of the Nature Communications initiative to facilitate training in peer review and to provide appropriate recognition for Early Career Researchers who co-review manuscripts

Reviewer #4 (Remarks to the Author): Expert in meningioma and CNS cancer genomics, and bioinformatics

In this manuscript the authors applied statistical methods to determine the optimized size thresholds for arm-level CNV events across meningiomas for individualized risk stratification. They also applied the same method to TCGA cohort composed of various cancer types for a similar stratification.

While the manuscript presents some intriguing insights, several significant concerns detract from its clarity and the main conclusions.

Thank you for your thorough and thoughtful review of our manuscript. As described below, we have now undertaken a major revision of our study that incorporates your helpful suggestions for improvement.

Here are my major concerns:

1. While the manuscript's central argument is valid, it is frequently misrepresented both conceptually and literally throughout the text. Specifically, the manuscript consistently refers to the cutoff for defining arm-level CNV events as "size thresholds for defining CNVs," conflating two distinct terms and potentially leading to misinterpretation.

Thank you for this comment. We have edited the text of our revised manuscript to define cumulative thresholds as the “cutoff for defining CNA events”. At the same time, to account for the importance of smaller copy number alterations, we have further explored the importance of focal binned regions along each chromosome (Fig. 2), termed “focal CNAs”. In brief, we now examine size, location, and contiguity of focal CNAs with finer granularity and specificity by dividing each chromosome arm into bins of 200 Kb. The approximate size of the *HLA* locus (200 Kb) was chosen as the bin size for focal analysis, given that copy number loss or gain of

the *HLA* locus is a prominent distinguishing marker of different DNA methylation groups of meningiomas¹. Gain or loss of each bin was inputted into univariate Cox proportional hazards models for 5-year LFFR and OS, and AUCs were plotted along chromosome arms (Fig. 2a). Focal regions with prognostic value were defined as those for which the AUC was at least 4% higher than the median for the entire chromosome arm. This threshold was chosen based on the magnitude of the AUC peak corresponding to the *CDKN2A/B* locus, which is well known to be a recurrently lost region with significant prognostic value and is sufficient for diagnosis of anaplastic meningioma, WHO grade 3.

These refined analyses identified focal prognostic regions on chromosomes 2q, 3q, 4p/q, 5p/q, 7p, 8p, 10q, 12q, 13q, and 14q, including a focal peak at the *CDKN2A/B* locus on 9p. Of these, only 3q, 9p, 12q, and 14q were identified as prognostic and size-dependent when using percent-arm-altered thresholds alone rather than the sliding window approach, suggesting that a more granular approach could identify additional focal gains and losses of prognostic significance.

Although a granular approach such as this may be more biologically and scientifically stringent, it may be too complex to always be clinically practical, as compared to use of a percent-arm-altered cutoff for defining CNA events. In fact, the use of highly variable definitions of percent-arm-altered cutoffs in the literature, ranging from 5% to 80%, was a major impetus for this manuscript. A 5% threshold, clearly, may risk “conflating” broad arm-level events and focal events. Nevertheless, for practical reasons, models incorporating CNAs such as Integrated grade and Integrated score have used a single threshold to call “arm-level” CNA events. Our findings provide an analysis of the potential impacts of the use of variable cutoffs for defining CNA events, and, more broadly, emphasize the importance of context in the interpretation of prognostic CNAs – whether focal or broad – in meningiomas and other cancers with respect to size, location, and co-occurrence with other CNAs. We now provide additional prose in the Discussion section of our revised manuscript that addresses this topic, and which we have copied here for ease of evaluation:

“Our finding that each of these systems achieved optimal performance using a size threshold close to the value used to train the model, and that performance degraded with variation of the size threshold, is perhaps unsurprising, but these observations further underscore the sensitivity of CNA models to size thresholds used for defining CNAs. At the level of model training, the size threshold determines the CNAs which form the model inputs and is thus a crucial branch point which influences the ultimate characteristics of the resulting model. For example, while both models incorporate overlapping CNAs, the optimal size threshold and performance across thresholds varied significantly, implying model specificity for CNA size thresholds. Furthermore, while uniform size thresholds across chromosome arms are commonly used, our results indicate that the optimal size threshold can vary by arm. By allowing for variation in arm level size thresholds, we show that additional prognostic CNAs can be identified, and that an optimal size-dependent CNA model may provide additional prognostic value as compared to Integrated grade and Integrated score.”

2. The considerable variability in cutoffs identified across both the Meningioma and TCGA cohorts underscores the manuscript's inability to establish a universally applicable cutoff with prognostic value for risk stratification. The identified cut-offs vary between 0.05 -0.71 with the highest 0.95 Optimal AUC. The same variability is observed in the TCGA cohort (5% for chr7p gain vs 95% for chr7q gain in GBMs).*

Thank you for this comment. Our findings underscore the importance of the considerable heterogeneity in cutoffs for defining CNAs in meningiomas across chromosome arms, as well as in most other cancers. We demonstrate that careful consideration of chromosome-specific cutoffs can yield models with strong prognostic performance that capture novel CNAs and co-occurrence patterns of interest, and that future study of CNAs in cancer may benefit from more tailored approaches. Our goal was never to identify a universally applicable cutoff, and our data do not support the existence of such a cutoff. We apologize for the confusion and have clarified these nuances in the Discussion section of our revised manuscript.

3. I find the conclusion to propose a superficial classifier, making it challenging to generalize, especially considering the potential for clinical utility. However, it remains unclear whether the ultimate goal of the study was to define a classifier specifically intended for clinical application. Are the authors suggesting using ~78 distinct cutoffs to define chromosome level events? The manuscript briefly mentions an effort to define a “uniform” threshold by testing their approach against the previously published “integrated grade” and “integrated score” approaches. However even with this approach, the manuscript still comes up a very wide range of cutoff values (20-30% vs. 5%), which is not an improvement on the currently used varying cutoff values.

We agree that a system using a different size cutoff for each chromosome gain or loss (for each cancer type) may be suboptimal in clinical practice, but the purpose of this study was to highlight the variability in prognostic

CNA size and focality across chromosome arms (Figs. 1a, 2a), which challenges the notion of a “uniform” threshold for meningioma and other cancers. We apologize for using this word in our initial submission, and for mistakenly giving the impression that we aimed to identify such a threshold. The available data suggest that an optimized system might take into account CNAs for which broad loss or gain is required for clinical relevance (such as loss of 1p or 22q in meningioma), as well as CNAs for which a focal region has particular prognostic importance (such as loss of 9p in meningioma). By building a size-dependent CNA model that outperforms or adds value to existing models, we propose that such a system is worth exploring and need not sacrifice clinical viability. Our results emphasize the importance of context in the interpretation of prognostic CNAs in meningiomas and other cancers with respect to size, location, and co-occurrence with other CNAs. We also provide additional discussion of this observation in the Discussion of our revised manuscript, which we have provided above in response to Reviewer 4 Comment #1.

4. The identification of potentially new prognostic markers for new chromosome level CNV events are mentioned, but not shown how these specific events are "prognostic". Are these events prognostic alone or together with other events?

Thank you for this question. Please see our response to Reviewer 1 Comment #4, where we report that certain focal regions are individually more prognostic for meningioma outcomes (Fig. 2a). Additionally, in our response to Reviewer 1 Comment #3, we discuss how focally affected genes in recurrently lost regions may have individual or combined prognostic value. In brief, our new analyses in response to this area of critique from Reviewers 1 and 4 identified focal prognostic regions on chromosomes 2q, 3q, 4p/q, 5p/q, 7p, 8p, 10q, 12q, 13q, and 14q, including a focal peak at the *CDKN2A/B* locus on 9p, as expected. Of these, only 3q, 9p, 12q, and 14q were identified as prognostic and size-dependent when using percent-arm-altered thresholds alone rather than the sliding window approach, suggesting that a more granular approach could identify additional focal gains and losses of prognostic significance. Notably, not all regions of focal loss identified by visual inspection of CNA profiles were found to have prognostic significance. Ontology analysis of the genes contained in these combined focal prognostic regions identified enrichment of STAT1 signaling and type I interferon signaling pathways (Fig. 2b), which have previously been implicated in cellular responses to genomic instability. As these focal regions were predominately associated with copy number loss rather than gain, acquisition of focal CNAs could reflect a mechanism of downregulation of immune signaling pathways normally associated with innate response to genomic instability. Indeed, suppression of type I interferon signaling may represent a mechanism of immune evasion in the setting of chromosomal instability², which has been shown to induce a cGas-STING mediated type I interferon response. Moreover, loss of the interferon gene cluster on chromosome 9p has been shown to be linked to reduced T cell infiltrate and poorer prognosis in melanoma and HPV-negative head and neck cancer patients^{3,4}, and we have shown T cell infiltrate is reduced in meningiomas that are resistant to standard interventions. These new analyses are now presented in the Results section of our revised manuscript.

5. It is not clear how clustering based on CNV burden leads to a conclusion about timing of CNV occurrence. The presented analysis merely reveals co-occurrence patterns, rather than "timing" of occurrence. Moreover, it is not clear what authors mean by "burden" and "total number of CNVs". Is this only counting arm-level events determined by in the manuscript? Or does this include focal CNV events as well?

Thank you for this comment. We have removed mention of timing of CNAs to avoid overstating the conclusions that can be drawn from our data, which show that increased CNA burden is associated with higher grade meningiomas (Fig. 3b) and worse outcomes (Fig. 3c). We have also clarified in the text that CNA burden is the total number of CNAs in a given meningioma, defined using the optimal cumulative threshold for CNAs on each chromosome. There can thus be a maximum of one CNA called for each chromosome per meningioma, which may be focal or broad depending on the prognostic significance of such events for each chromosome.

6. The definition and significance of "focal" CNVs require clarification. Since the large-scale/arm level CNV identification varies across the chromosomes in the study, does the cutoff of focal CNVs vary across the chromosomes as well? This is critical and should be better described. Also why did the authors choose to test the expression of genes in focal CNV regions against the other genes on the same chromosome arm rather than comparing to "same" region/genes in samples without the CNV events?

We have clarified in the text of our revised manuscript that focal regions of copy number gain or loss were determined through manual inspection of the proportion of samples with gain and loss across chromosome arms, measured in increments of 30 Kb. Thus, the size of these focal regions was not set or predetermined beyond

the minimum size of 30 Kb. We have also revised the gene expression analysis to compare with expression of the same genes in samples without focal CNAs in that region.

In order to examine the prognostic value of CNAs in a more granular fashion, we now divide each chromosome arm into bins of 200 Kb. We have also edited the text and figure legends of our revised manuscript to mention that HLA is a prominent distinguishing marker that is gained or lost in different DNA methylation groups of meningiomas¹, and the approximate size of this locus (200 Kb) was chosen as the bin size for focal CNA analyses. Gain or loss of each bin was inputted into univariate Cox proportional hazards models for 5-year LFFR and OS, and the AUCs were plotted along the chromosome arms (Fig. 2a). Focal regions with prognostic value were defined as those for which the AUC was at least 4% higher than the median for the entire chromosome arm. This threshold was selected based on the magnitude of the AUC peak corresponding to the *CDKN2A/B* locus on chromosome 9p, which is known to be a recurrently lost region with significant prognostic value that is now codified as sufficient for diagnosis of anaplastic WHO grade 3 meningioma.

This new analysis identified focal prognostic regions on chromosomes 2q, 3q, 4p/q, 5p/q, 7p, 8p, 10q, 12q, 13q, and 14q, including a focal peak at the *CDKN2A/B* locus on 9p. Of these, only 3q, 9p, 12q, and 14q were identified as prognostic and size-dependent when using percent-arm-altered thresholds alone rather than the sliding window approach, suggesting that a more granular approach could identify additional focal gains and losses of prognostic significance. In addition, not all regions of focal loss identified by visual inspection of CNA profiles were found to have prognostic significance. Ontology analysis of the genes contained in these combined focal prognostic regions identified enrichment of STAT1 signaling and type I interferon signaling pathways (Fig. 2c), which have previously been implicated in cellular responses to genomic instability. As these focal regions were predominately associated with copy number loss rather than gain, acquisition of focal CNAs could reflect a mechanism of downregulation of immune signaling pathways normally associated with innate response to genomic instability. Indeed, suppression of type I interferon signaling may represent a mechanism of immune evasion in the setting of chromosomal instability², which has been shown to induce a cGas-STING mediated type I interferon response. Moreover, loss of the interferon gene cluster on chromosome 9p has been shown to be linked to reduced T cell infiltrate and poorer prognosis in melanoma and HPV-negative head and neck cancer patients^{3,4}, and we have shown T cell infiltrate is reduced in meningiomas that are resistant to standard interventions⁵. These new analyses are now presented in the Results section of our revised manuscript.

Finally, to address the possibility of focal or broad amplifications beyond gain of a single copy, or focal homozygous loss, we generated sample-wise segment plots of CNAs by chromosome position plotted against the amplitude of gain or loss (below, left, example of 1q and 17q gain). Other than identifying the same regions of focal loss or gain as shown in Fig. 2a, this analysis did not reveal clear recurrent regions of focal deep deletion or high level amplification, except for the *CDKN2A/B* locus on chromosome 9p, which is expected to be homozygously deleted in some samples, and the *HLA* locus on chromosome 6p. Plotting segment mean intensities for the two chromosome arms most associated with gains, 1q and 17q, revealed no clear patterns of focal amplification (below, left), suggesting that a uniform intensity threshold was sufficient to capture prognostic CNA gains. As an additional sensitivity analysis, we examined whether varying the intensity threshold by 0.05 increments significantly altered the prognostic value of each chromosome arm and found that the optimal intensity threshold for calling arm-level CNAs appeared to be our selected thresholds of -0.1 for loss and 0.15 for gain, with diminishing AUC with higher thresholds (below, right). These new analyses have been incorporated into the Results section of our revised manuscript.

7. Detailed cohort description of both cohorts is missing and is not consistent. While Grade 1-2-3s are included in UCSF cohort, no such information is provided for the Hong Kong cohort? Furthermore, it is now known (and published many times) that higher grade meningiomas have higher CNV burden compared to grade 1 cases. Do the higher grade samples in this cohort indeed have higher CNV burden, when their optimized cut-off approach is applied to define arm-level events? The recurrence prediction in Grade-1 meningiomas vs. higher grade meningiomas should be calculated separately or should be corrected in the multivariate analysis. Furthermore, the treatment status of any of the higher grades is not reported, as they sometimes receive radiation treatment. The study does not take these clinically relevant variables into consideration.

Thank you for providing these important considerations. We have provided more detailed description of our cohort and the new external validation cohort in the text and in Tables 1 and 2. As in our responses to Reviewer 1 Comments #1 and #8, we now demonstrate in our clustered heatmaps that CNA burden defined using optimized size-thresholds correlates with WHO grade (Fig. 3b). To account for the effect of WHO grade and to improve the clinical relevance and comparability of our models to existing ones, we combined CNA and co-occurrence models with WHO grade, mitoses per 10 high-power fields, and CDKN2A/B loss (the additional variables that were included alongside CNAs during the development of Integrated grade). The inclusion of adjuvant radiotherapy as predictor is a clinically relevant suggestion. We have included the rates of postoperative radiotherapy in our descriptions of the cohorts, as well as in multivariate analysis in external test set 1 (Supplementary Table 3). External test cohort 2 was comprised of entirely newly diagnosed meningiomas without prior treatment, allowing for comparison between those that had and had not received radiotherapy. Adjuvant radiotherapy was a significant predictor of LFFR but not OS in external test cohort 1, including in cases where WHO grade was controlled for. Currently, the criteria to prescribe adjuvant radiotherapy for intermediate risk meningiomas are controversial, and models which can better stratify these patients are of considerable value^{17,18}. The models we describe here, along with many prominent meningioma risk stratification systems^{15,16,19} are designed with a focus on guiding postoperative therapy and management based on data obtained from resected or biopsied tissue. We have clarified this in the Discussion section of our revised manuscript.

8. The identified co-occurrence patterns in Meningiomas are quite intriguing and important.

We appreciate the positive consideration to these analyses.

Minor concerns:

1. The lack of descriptive clarity in figures and legends, particularly in Figure 1d-e-f, limits understanding. Clearer annotations would enhance the interpretability of the presented data.

These data are now presented in Fig. 2, where we now show focal regions of loss or lack thereof across all chromosome arms, explore gene expression changes in each region, and provide dotplots demonstrating ontology analysis of genes in key regions. We have also the expanded character count limit in our revised manuscript to increase our annotations in the figures and legends that correspond to these data.

2. It is interesting that chr7p amplification and chr10 deletion is not identified as a co-occurring event in GBMS. Even though it looks like it was identified in Figure 3b, it is not mentioned in the manuscript and the Kaplan-Meier curve was plotted for chr7p gain-chr16q loss, which seems to have a lower co-occurrence score. It is not clear why this co-occurrence pattern was selected.

Thank you for this comment. While concurrent gain of chromosome 7 and loss of chromosome 10 are known to be enriched in GBM, these were not shown to be important for clinical outcomes in our co-occurrence analyses (coefficient plots below), which were limited to pairs including at least one size-dependent CNA. Kaplan-Meier analysis of chromosome 7 amplification and chromosome 10 deletion demonstrated that GBMs with co-occurrence of both events had similar outcomes to having one or the other (Kaplan-Meier curves below). Co-occurrence of 7p loss and 16q gain in GBM was rare and thus analysis of its prognostic impact is limited (Kaplan-Meier curves below). We have clarified this in the Results section of the revised manuscript.

Reviewer #5 (Remarks to the Author): *Clinical expert in meningioma and CNS cancers, genomics, risk prediction and stratification*

The study investigates optimal size thresholds for copy number variations (CNVs) and their co-occurrence patterns in outcome assessment models using DNA methylation arrays across meningiomas and other tumors. By identifying optimal chromosome size thresholds in various tumors, the authors developed predictive models that unveiled prognostically relevant and novel individual and co-occurring CNVs not detected by previous CNV burden-based models.

The subject is relevant, the paper concise, well-written and overall clear, except for the lack of a couple of definitions as delineated below. However, the advantage of their approach remains unclear as the authors do not compare the predictive performance of their novel approach with existing methods or demonstrate its generalizability to an external validation meningioma cohort. Furthermore, the absence of a detailed plan for a potential clinical deployment workflow renders the clinical significance of this work elusive.

Thank you for your thorough and thoughtful review of our manuscript. As described below, we have now undertaken a major revision of our study that incorporates your helpful suggestions for improvement.

Suggestions to strengthen the study include:

1. Comparing prediction performance with other CNV-based models.

Thank you for this suggestion for improvement. A direct comparison requires delineation of training and test sets. In order to more fully examine the performance of models based upon size-dependent CNAs, we now delineate a “**discovery cohort**” (N=200, University of California, San Francisco [UCSF]) and an “**external test cohort 1**” (N=365, University of Hong Kong) of meningiomas within our dataset, and compare the validation performance of the size-dependent CNA models we propose with the performance of previously described molecular classification systems for meningiomas (Integrated grade and Integrated score), with or without size-dependent CNAs, using meningiomas that were not used for the development of any of these models (external test cohort 1). Furthermore, we now include a new independent validation cohort (“**external test cohort 2**”) that is comprised of 126 meningiomas from Baylor College of Medicine (BCM) with available DNA methylation profiling and clinical outcome data. LASSO and Elastic Net regularized Cox regression models are now trained on the discovery cohort of meningiomas to identify CNA predictors. Consistent with our initial observation that flexible size thresholds may more sensitively identify prognostic CNAs, these models identified CNAs that were not included in Integrated grade or Integrated score, such as 1q gain, 7p loss, and 12q loss (Fig. 1c). To generate a clinically applicable risk classification system similar to Integrated grade and Integrated score, we combined the presence of size-dependent, prognostic CNAs with WHO 2016 histological grade, mitotic index, and *CDKN2A/B* status (features which were present in either Integrated score or Integrated grade) in order to generate a size-dependent integrated molecular risk score.

Validation of the size-dependent risk score described above in external test cohort 1 (UHK, N=365) yielded a maximum 5-year AUC for LFFR and OS of 0.83 and 0.78, respectively. In external test cohort 2 (BCM, N=126), the size-dependent risk score achieved a maximum 5-year AUC for LFFR and OS of 0.78 and 0.88, respectively. In the external test cohort 1, likelihood ratio tests demonstrated that the addition of a size-dependent model to Integrated grade significantly improved prediction of LFFR (P=0.0055), but reciprocal addition of Integrated grade to the size-dependent model did not improve prediction of LFFR (P=0.32). Addition of the size-dependent model to Integrated score also improved prediction of LFFR (P=0.00088), and the reciprocal was also true, albeit with a more modest P-value (P=0.013). This same pattern was observed for prediction of OS, where adding the size-dependent model significantly improved performance of Integrated grade (P=7.01e-05), but the reciprocal was not true (P=0.18), and adding the size-dependent model significantly improved performance of Integrated score (P=5.0e-06), and the reciprocal was also true, albeit with a more modest P-value (P=0.014). Size-dependent CNA models additionally provided further stratification within low and intermediate Integrated grade or Integrated score risk groups (Extended Data Fig. 4). In multivariate regression combining the size-dependent CNA models with patient sex, EOR, primary versus recurrent meningioma presentation, WHO grade, and adjuvant radiotherapy, the size-dependent CNA model remained significantly prognostic (P<0.0001) for LFFR and OS in the external test cohorts (Supplementary Table 3; nomograms generated in Extended Data Fig. 3). When incorporated with Integrated grade or Integrated score in multivariate regressions, the size-dependent CNA model remained independently prognostic in both external test cohorts (Supplementary Table 3).

Performance of Integrated grade and Integrated score were also measured in external test cohort 1 using (1) CNAs defined at the thresholds with which these models were trained (e.g. 50% for Integrated grade and 5% for Integrated score), (2) CNAs defined at the threshold with which these models performed best in our Discovery cohort (17% for Integrated grade and 4% for Integrated, respectively), and (3) our optimized CNA thresholds for LFFR and OS, which ranged between 1% and 97% of chromosome arms. Integrated grade as a continuous score had a maximum 5-year LFFR AUC of 0.82 in external test cohort 1 when defined using either optimized or original thresholds, and Integrated grade as a discrete score had a maximum AUC of 0.79 when defined using optimized or original thresholds. Continuous and discrete Integrated score had maximum 5-year LFFR AUCs of 0.80 and 0.74, respectively, when using either optimized or original thresholds. As suggested, we now provide ROC curves for our size-dependent CNA risk model, Integrated Grade, and Integrated score when defined at optimized thresholds for 5-year LFFR and OS in Extended Data Fig. 2d. Comparison was not possible in external test cohort 2 as mitotic data were unavailable to determine Integrated grade.

In summary, while the aim of this project was not to propose a clinical risk stratification model to replace existing ones, we do demonstrate that incorporating variable size thresholds can identify prognostic CNAs not captured by other systems without sacrificing predictive performance. Moreover, as a proof of concept, a size-dependent CNA model trained using similar histopathologic and clinical features as Integrated grade or Integrated score resulted in a size-dependent CNA risk score that (1) achieved excellent performance in external test sets, (2) provided statistically significant additional prognostic value when added to Integrated grade or Integrated score, resulting in improved risk stratification within molecular risk groups, and (3) remained independently prognostic in multivariate analysis when incorporated alongside Integrated grade and score.

2. Conducting a comparative analysis of outcomes between 1p/19q and 7p/16q for gliomas.

Thank you for this suggestion, which highlights a few of the known co-occurrent CNA pairs commonly associated with gliomas. The Pan-Cancer Atlas from which TCGA data was collected differentiates between low-grade glioma (LGG) and glioblastoma (GBM), so our analyses were also separated into these two groups. While 1p/19q codeletion is marker of the generally less aggressive oligodendroglioma subtype of LGG, it was not included in our co-occurrence models as neither 1p nor 19q loss were considered size-dependent by our criteria (AUC ≥ 0.6 , AUC standard deviation across thresholds ≥ 0.01 , CNA present in at least 2.5% of samples). We have included analysis of this pair in LGG, which demonstrates that 1p/19q co-deleted LGG samples had better outcomes as expected (Extended Data Fig. 10a). Co-occurrence of 7p loss and 16q gain in GBM was rare and thus analysis of its prognostic impact was limited (Kaplan-Meier curves below). We have clarified this in the Results section of our revised manuscript.

3. Investigating the influence of factors such as sex, tumor location, primary and recurrence status, and radiotherapy on size-dependent model performance.

Thank you for this suggestion. To better account for the effect of these important variables, we performed multivariate Cox proportional hazards regression combining our size-dependent CNA models with these clinical features, which identified the models, male sex, and recurrent setting as significant predictors of worse LFFR and OS (Supplementary Table 3). We also generated nomograms showing the relative importances of each of these variables (Extended Data Fig. 3). Tumor location data was not available for a large number of tumors in external test cohort 1 (HKU) and so was excluded from these analyses. However, tumor location was available in our new external test set (from Baylor College of Medicine), and our multivariable analyses incorporating this

additional variable (Supplemental Table 3) corroborated the independent prognostic value of size-dependent CNAs when accounting for tumor location in this cohort.

4. Assessing the generalizability of findings to other meningioma cohorts.

Thank you for this important suggestion. As described above in our response to comment #1, we now delineate a “**discovery cohort**” (N=200, University of California, San Francisco [UCSF]) and an “**external test cohort 1**” (N=365, University of Hong Kong) of meningiomas within our dataset, and compare the validation performance of the size-dependent CNA models we propose with the performance of previously described molecular classification systems for meningiomas (Integrated grade and Integrated score), with or without size-dependent CNAs. Furthermore, we now include a new independent validation cohort (“**external test cohort 2**”) that is comprised of 126 meningiomas from Baylor College of Medicine with available DNA methylation profiling and clinical outcome data (Table 2), with comparable model performance in both external test cohorts.

5. Clarifying the definition of "early" and "late" CNV alterations.

To avoid overstating conclusions, we have removed mention of “early” or “late” CNAs, and have instead demonstrated enrichment in samples with low, medium, or high CNA burden (e.g. the total number of CNAs per meningioma), which correlate with increasing WHO grade and worse clinical outcomes.

6. Providing figures illustrating the three k-cluster differences and LFFR and/or OS.

We now show Kaplan-Meier curves stratifying CNA burden according to cutoffs determined through recursive partitioning analysis in Fig. 3c.

7. Clarifying the weight of CNV burden in driving the identified clusters.

While certain CNAs appear enriched in meningiomas with low, medium, or high CNA burden, the clusters that we demonstrate were built on the presence of each CNA, and the binned burden along the x-axis is a visualization method to demonstrate this pattern. Thus, CNA burden had no weight in driving the clusters. We have clarified this in the Methods section of our revised manuscript.

8. Detailing potential plans for CNV validation and scalability.

We have added text detailing efforts for validation and scalability to the Discussion section of our revised manuscript. Moreover, we have now included an external validation cohort from an outside institution that is comprised of 126 meningioma samples with clinical outcomes (Table 2). As described in greater detail in our revised Discussion, further validation will come through testing with additional modern series of meningiomas that better model contemporary clinical cohorts. Moreover, the exploration of CNA size-dependence and co-occurrence in other cancers, as suggested by our TCGA analyses, presents many opportunities to validate our initial findings.

9. Developing a schematic workflow for model deployment and clinical application (from tissue obtention and profiling, model application through clinical utilization).

Thank you for this suggestion. The aim of this project was not to propose a clinical risk stratification model to replace existing ones, but rather to highlight the importance of considering size thresholds for defining prognostic arm-level CNAs. Furthermore, the identification of focal regions of prognostic value and the identification of important co-occurrent CNA pairs highlights the possibility of developing a more tailored system for meningioma or other cancers that incorporates a combination of these factors to maximize clinical risk stratification. We have provided a brief schematic below detailing the process of a possible workflow in which clinical samples are profiled using available molecular characterization processes and the data is used for treatment planning. Created in BioRender. Chen, W. (2025) <https://BioRender.com/b01y590>

① Obtain tumor tissue

② Identify CNAs via NGS, methylation profiling, or other methods

③ Obtain risk stratification using CNA based systems

④ Integrate molecular data with clinical data for patient centered decision making

10. Inclusion of relevant references – Ma et al. 2020, Sybren et al. 2021, Sahm et al. 2017.

Thank you for the suggested references, which we have incorporated into the text of our revised manuscript.

Minor:

11. Downplaying statements suggesting a trend of co-occurrence of prognostic CNVs across all models, as only a few CNVs were common.

Thank you for this suggestion. We have downplayed and tempered the statements suggested in our revised manuscript.

12. Temper the claim regarding the clinical relevance of CNV size-dependence and co-occurrence in human cancer, considering its applicability to only half of them.

Thank you for this comment. In the revised analyses described above, which were suggested by all reviewers, we have now demonstrated patterns of prognostic size-dependent CNA co-occurrence across nearly all 33 cancers in the Pan-Cancer Atlas (Supplementary Table 8). Nevertheless, we have tempered the claims suggested as these data are correlative and mechanistic studies were beyond the scope of this study.

References

1. Choudhury, A. et al. Meningioma DNA methylation groups identify biological drivers and therapeutic vulnerabilities. *Nat Genet* **54**, 649–659 (2022).
2. Li, J. et al. Non-cell-autonomous cancer progression from chromosomal instability. *Nature* **620**, 1080–1088 (2023).
3. William, W. N. et al. Immune evasion in HPV– head and neck precancer–cancer transition is driven by an aneuploid switch involving chromosome 9p loss. *Proceedings of the National Academy of Sciences* **118**, e2022655118 (2021).
4. Linsley, P. S., Speake, C., Whalen, E. & Chaussabel, D. Copy Number Loss of the Interferon Gene Cluster in Melanomas Is Linked to Reduced T Cell Infiltrate and Poor Patient Prognosis. *PLOS ONE* **9**, e109760 (2014).
5. Lucas, C.-H. G. et al. Spatial genomic, biochemical and cellular mechanisms underlying meningioma heterogeneity and evolution. *Nat Genet* **56**, 1121–1133 (2024).
6. Tao, Z. et al. The repertoire of copy number alteration signatures in human cancer. *Briefings in Bioinformatics* **24**, bbad053 (2023).
7. Ben-David, U. & Amon, A. Context is everything: aneuploidy in cancer. *Nat Rev Genet* **21**, 44–62 (2020).
8. Louis, D. N. et al. The 2021 WHO Classification of Tumors of the Central Nervous System: a summary. *Neuro Oncol* **23**, 1231–1251 (2021).
9. Privitera, A. P., Barresi, V. & Condorelli, D. F. Aberrations of Chromosomes 1 and 16 in Breast Cancer: A Framework for Cooperation of Transcriptionally Dysregulated Genes. *Cancers (Basel)* **13**, 1585 (2021).
10. Palin, K. et al. Contribution of allelic imbalance to colorectal cancer. *Nat Commun* **9**, 3664 (2018).
11. Virgin, J. B. et al. Isochromosome 8q formation is associated with 8p loss of heterozygosity in a prostate cancer cell line. *Prostate* **41**, 49–57 (1999).
12. Hieronymus, H. et al. Tumor copy number alteration burden is a pan-cancer prognostic factor associated with recurrence and death. *eLife* **7**, e37294 (2018).

13. Zhou, W., Triche, T. J., Jr, Laird, P. W. & Shen, H. SeSAmE: reducing artifactual detection of DNA methylation by Infinium BeadChips in genomic deletions. *Nucleic Acids Research* **46**, e123 (2018).
14. Weinstein, J. N. *et al.* The Cancer Genome Atlas Pan-Cancer analysis project. *Nat Genet* **45**, 1113–1120 (2013).
15. Driver, J. *et al.* A molecularly integrated grade for meningioma. *Neuro Oncol* **24**, 796–808 (2022).
16. Maas, S. L. N. *et al.* Integrated Molecular-Morphologic Meningioma Classification: A Multicenter Retrospective Analysis, Retrospectively and Prospectively Validated. *J Clin Oncol* **39**, 3839–3852 (2021).
17. Chen, W. C. *et al.* Radiotherapy for meningiomas. *J Neurooncol* **160**, 505–515 (2022).
18. Chen, W. C., Lucas, C.-H. G., Magill, S. T., Rogers, C. L. & Raleigh, D. R. Radiotherapy and radiosurgery for meningiomas. *Neurooncol Adv* **5**, i67–i83 (2023).
19. Chen, W. C. *et al.* Targeted gene expression profiling predicts meningioma outcomes and radiotherapy responses. *Nat Med* (2023) doi:10.1038/s41591-023-02586-z.

Reviewer #1

I commend the authors on thoroughly addressing each point of my review. This manuscript is significantly improved. The study's scope and limitations are now defined, and the authors' have filled major gaps in their original analysis, interpretation, and reporting. Their methods are conveyed more clearly, their conclusions are supported by evidence, and their findings are contextualized by discussion of other studies.

Thank you once again for your thoughtful comments on our manuscript.

Please see the remaining minor points below which are largely graphical/textual:

Points 10-12 — The use of ROC AUCs as your primary metric for model quality is much clearer. I do think providing additional representative examples of these ROCs in the manuscript (as requested and provided in the response to Point #12) would significantly aid others' understanding of the analytical method central to this study.

Thank you for this comment and for the opportunity to further improve the clarity of our methods. ROC curves for the different models being compared are provided in Extended Data Fig. 2d of our revised manuscript, and we mention in our Methods section that “5-year area under the curve (AUC) for a time-dependent receiver operating characteristic (ROC) curve for meningioma LFFR and OS, and for TGCA PFS and OS, were used as primary measures of prognostic value for individual CNAs”.

Point 14 — The scales of these CNA profiles remain somewhat unclear. When the authors plot copy number profiles for chromosomes (Fig 2a, Extended Figs 1 and 5), I assume that they are showing either the full chromosome or full chromosome arm. Providing genomic coordinates on the plot axes or as a scale bar would clarify this.

Thank you for this comment. We have clarified that CNA profiles show the entire length of their respective chromosomes and chromosome arms in the legends for Fig. 3a (formerly Fig. 2a) and Extended Data Fig. 1 and 5.

Figures 2a/2b — It is unclear how the data in Fig 2b were derived and what they represent. Do the distributions these means are derived from represent mean expression per gene for all meningiomas or mean expression per gene averaged across meningiomas? These distributions should be shown. Also, in relating the categories here ('No CNA' or 'Focal copy number deletion' and 'No AUC peak' and 'Focal AUC peak') to Fig 2a, it would be helpful to highlight exactly which regions are being marked as focal CNAs or focal AUC peaks and are therefore included in 2b.

Thank you for this suggestion. We have clarified in the legend of Fig. 3b (formerly Fig. 2b) that gene expression data were derived from sample-matched RNA sequencing of genes mapping to focal regions identified by cross-referencing chromosomal position with the Ensembl database. Furthermore, we have now included asterisks in Fig. 3a (formerly Fig. 2a) to denote focal copy number and prognostic regions from which these genes were identified, which are also provided in Supplementary Tables 4 and 5. Moreover, we have provided the relevant revised text from the Methods section for ease of evaluation below.

“Genes present in focal regions of loss and in prognostic regions identified on binned CNA analysis were identified by cross-referencing positions along the chromosome with the Ensembl (release 109) database using the biomaRt (v2.54.1) package in R. Meningioma gene expression analysis was performed using sample-matched RNA sequencing data, as previously described. Briefly, RNA sequencing was performed on all 200 of the UCSF discovery cohort samples and 302 of the Hong Kong University external test cohort 1 samples meeting quality metrics. For UCSF samples, library preparation was performed using the TruSeq RNA Library Prep Kit v2 (RS-122-2001, Illumina), sequencing was performed on an Illumina HiSeq 4000 to a mean of 42 million reads per sample at the UCSF IHG Genomics Core, quality control of FASTQ files was performed with FASTQC (v0.11.9), and 50 bp single-end reads were mapped to the human reference genome GRCh38 using HISAT2 (v2.1.0) with default parameters. For Hong Kong University samples, library preparation was performed using the TruSeq Standard mRNA Kit (20020595, Illumina) and 150 bp paired-end reads were sequenced on an Illumina NovaSeq 6000 to a mean of 100 million reads per sample at MedGenome Inc. Analysis was performed using a pipeline comprised of FastQC for quality control and Kallisto for reading pseudo alignment and transcript abundance quantification using default settings (v0.46.2). Comparison of gene expression levels was performed using the Wilcoxon rank-sum test on log₂ transformed transcripts per million (TPM).”

Figure 4a & Extended Fig 9a/b — Indicating (1) the magnitude of the effect, (2) the statistical significance, and/or (3) the actual optimal size threshold for each data point on the plot would aid the impact/take-home of Figure 4. As of now, if I'm interested in a specific cancer type presented here, I have to look across several supplemental tables and figures to access this information.

Thank you for this suggestion. We have now provided consolidated data on optimal size thresholds and AUCs for size-dependent TCGA cancers in Supplementary Table 8 and provide a reference to this table in the legend of Fig. 5 (formerly Fig. 4). We have also further clarified the size-dependent criteria within the legend of Extended Data Fig. 9.

Manuscript Density — I strongly encourage the authors to break up their results section into major sub-sections to improve readability of this very dense manuscript.

Thank you for this suggestion. We have now added subsection headings to the Results section of our revised manuscript.

Reviewer #2

With regards to my concerns about the background material, and the surrounding body of research, the authors have satisfied all the points I raised, including plenty of references to various different previous studies, and general better grounding of the present research in the extant literature.

The authors have also addressed all of my points about the size-dependent CNAs and their prognostic powers. Overall, the authors have drastically improved the paper, grounding their conclusions in many more relevant analyses. I'd like to highlight in particular the analysis of histologically distinct subtypes of cancer, which I thought to be the most interesting of the results presented.

Thank you once again for your thoughtful comments on our manuscript.

I would recommend the editor accept the manuscript, with minor revisions:

Focal CNAs:

In the "Binned CNA Analysis" subsection of "Methods" (l.540ff), the authors say "Focal regions of increased prognostic value were selected by manually inspecting AUCs across a chromosome arm and selecting bins with an AUC at least 0.04 greater than the median for that arm." I suppose that the authors mean to say they have a computer script that can do this. "Manual" suggests, to me, "by eye". Actually that whole section feels somewhat repetitive, so the authors might benefit from editing it down.

*Further to that point, the authors choose a "resolution criteria" (my words) for deciding if a focal CNA is prognostic. My reading of the text has two conflicting definitions for the resolution criteria: (a) $AUC \geq 0.04 + \text{mean AUC for that arm}$, and (b) $AUC \geq 1.04 * \text{mean AUC for that arm}$. The second definition comes from l.534: "defined as those for which the AUC was at least 4% higher than the median", the first from l.542: "AUC at least 0.04 greater than the median for that arm." The authors should make clear which is the actual case. I assume it is (a).*

Thank you for this comment. We have removed this confusing terminology, clarified that the definition of a focal prognostic CNA as mean arm AUC + 0.04, and edited the binned CNA analysis subsection of the revised Methods section, with the new prose provided here for ease of evaluation:

"Chromosome arms were divided into bins of 200,000 bases, which was chosen as it approximates the size of the *HLA* locus, a prominent distinguishing marker that is gained or lost in different DNA methylation groups of meningiomas. For each bin a weighted average of the mean intensities of all segments overlapping with the bin was calculated, weighted to account for length of the overlap and the number of methylation probes in each segment. Weighted average values less than -0.1 were defined as lost, and those greater than 0.15 were defined as gained, as described above. The prognostic value of each focal bin was measured using AUC for 5-year LFFR or AUC for 5-year OS. Focal CNAs with prognostic value were defined as those for which the AUC was at least 0.04 higher than the median for the entire chromosome arm, which was chosen as it approximates the magnitude of the AUC peak corresponding to the *CDKN2A/B* locus, another well-known recurrently lost region with significant prognostic value in meningioma."

I.173ff, I don't really understand this segment as it stands in relation to Fig. 2a: "Other than identifying the same regions of focal loss or gains as our sliding window approach (Fig. 2a), this analysis did not reveal clear recurrent regions of focal deep deletion or high level amplification, except for the CDKN2A/B locus on chromosome 9p, which is expected to be homozygously deleted in some samples, and the HLA locus on chromosome 6p, which is expected to be homozygously deleted or amplified in some samples". Don't I see from the plot what the authors are trying to say. Aren't the "clear recurrent regions of focal deep deletion" the blue regions of the copy-number profiles (bottom panels/row) which have a large amplitude (e.g. near the centromere for chr3, beginning of 7q, end of 8q, etc.)? Also, the authors do not comment on the fact that several of these recurrent copy-number losses do not correlate with changes in the prognostic AUC (or in fact, sometimes anti-correlate). I think this is worth mentioning. The areas I am thinking of are the end of 18q, the spike near the centromere of 10q, some of the losses on 8p, to name a few examples.

Thank you for this comment. We have further clarified this section of the text to articulate that these were a separate analyses that were not shown in Fig. 3 (formerly Fig. 2), and provided this revised text below. We now also more clearly label the *CDKN2A/B* and *HLA* loci in Fig. 3.

"To address the potential impact of greater degrees of copy number amplification or deletion on meningioma outcomes, such as trisomy or homozygous loss, we generated sample-wise CNA segment plots by chromosome position plotted against the amplitude of loss or gain as measured by segment intensity. Other than identifying the same regions of focal loss or gains as our sliding window approach (Fig. 3a), this analysis did not reveal clear regions of decreased or increased intensity to suggest focal deep deletion or high level amplification, except for the *CDKN2A/B* locus on chromosome 9p, which is expected to be homozygously deleted in some samples, and the *HLA* locus on chromosome 6p, which is expected to be homozygously deleted or amplified in some samples."

Additionally, we have added text to the Results section mentioning the discrepancies in regions of focally recurrent deletion and focal prognostic value, provided below.

"Interestingly, regions with focally recurrent copy number deletions did not consistently align -- and were sometimes negatively associated -- with regions of focal prognostic value, such as those on chromosomes 8p, 10q, and 18q. This suggests that random copy number changes alone may not worsen clinical outcomes, and that focal prognostic losses could impact chromosomal regions critical for meningioma biology."

Finally, we have also added text discussing the discrepancies in focal recurrent loss regions and regions of focal prognostic value as measured by AUC.

Further to that point, in I.547ff: the authors say that "Focal regions of loss were selected by manual inspection of regions along the chromosome arm with a higher proportion of samples demonstrating deletion compared to the surrounding regions." This feels a little vague, and raises two points: (a). Why not automate the procedure? (b). What does higher proportion mean?

Thank you for this comment. While automation is a valid strategy to capture regions of focal change, we opted for manual identification of focal change (without predetermined numeric cutoffs) to account for subtle variability in the background frequency of deletion or artifactual peaks or valleys due to uneven distribution of methylation probes. Indeed, manual inspection of focal CNAs has been previously reported as a preferred method of identifying hemizygous and homozygous deletion of the *CDKN2A/B* locus (Wang et al. *Acta Neuropathologica* 2023, Ippen et al *Neuro-Oncology* 2024). We have now added this explanation to our revised Methods section, and also provided it below.

"CNA pileup plots demonstrating the proportion of tumors with losses or gains at each position along each chromosome arm were constructed using the *ggplot2* (v3.4.3) package in R. Focal regions of loss were selected by manual inspection of regions along the chromosome arm with a notably higher proportion of samples demonstrating deletion compared to the surrounding regions. This method was chosen to better account for subtle variability in the baseline frequency of loss across each chromosome arm or artifactual peaks or valleys due to uneven distribution of methylation probes. Manual inspection of focal CNAs has been previously reported as a preferred method of identifying hemizygous and homozygous deletion of the *CDKN2A/B* locus. Genes present in focal regions of loss and in prognostic regions identified on binned CNA analysis were identified by

cross-referencing positions along the chromosome with the Ensembl (release 109) database using the *biomaRt* (v2.54.1) package in R.”

Co-occurrence modeling:

Extended Data Fig. 6 - The Kaplan-Meier curve for LFFR (1p/7p/22q) is either missing a line or else it appears that the 1p loss and the 7p loss are overlapping. If so, the authors may benefit from using different line styles or more contrasting colours with a lower transparency. Further to this point, the neighbouring Kaplan-Meier curve for OS (1p/9p/22q) is definitely missing a line, which as far as I can tell is for the 9p/14q co-deletion class.

Thank you for this comment, and for the careful attention to detail in our supplementary figures. The Kaplan-Meier curve in Extended Data Fig. 6 showing different combinations of 1p, 7p, and 22q losses did indeed have overlapping curves for individual 1p or 7p loss. We have addressed this in the revised figure using contrasting line types. For the neighboring curve showing combinations of 1p/9p/14q (mistakenly labeled 1p/9p/22q, now corrected), there were no cases of 9p/14q co-deletion without 1p, and thus no data to plot for that combination. We have removed this label from the plot.

I.187ff - I am not entirely sure I understand the sentence: “Each of these CNA pairs were prognostic when defining at optimal size thresholds and across thresholds of 5% or 80% in external test cohort 1”. What do the authors mean by “across thresholds of 5% or 80% in external test cohort 1”?

We have clarified this sentence in the revised Results section, and provide the amended text below:

“Each of these CNA pairs remained prognostic when defining CNAs at different size thresholds, including the optimal size thresholds identified in Supplementary Table 1, a threshold of 5% uniformly applied to each chromosome arm, or a threshold of 80% uniformly applied to each chromosome arm, using samples in external test cohort 1 (Fig. 4a).”

I.225ff - The authors may benefit from a small selection of figures (supplementary, extended, or otherwise) to back up this statement: “Unlike meningioma, several TCGA cancers had size-dependent co-occurrent CNA pairs that were associated with better outcomes, including ACC, BLCA, HNSC, LAML, LUSC, and THCA.”

We have now added a reference in this sentence to Supplementary Table 9, in which there are numerous co-occurrent pairs associated with improved PFS or OS.

Extended Data Fig. 8a - Am I wrong in saying that beyond the 2 year mark, 18p loss is not significantly better than the triple deletion case? Looking at the 50% survival time, the two are comparable. I would say that the authors are overstating their results in the caption of the figure, though I would agree that all the other co-deletion/intact cases have a better patient outcome than the triple deletion case. It is interesting to note that 18q loss has a better PFS than if it is intact.

Thank you for this comment. It is true that the differences in PFS between 10q/18p/18q triple deletion and 18p loss alone become similar after 2 years, although the lower sample size for 18p loss alone makes estimation of PFS difficult. We have made note of this caveat in the revised legend for Extended Data Fig. 8. The finding of improved PFS in the 18q loss subgroup compared to other subgroups is indeed interesting. It should be noted that these groupings do not account for other CNAs outside of the three included, and cases without 18q loss may in fact be enriched in other alterations that confer worse prognosis.

Discussion:

I.287 - The authors mention their analysis of focal CNAs in chromosome 1, but did not show it in any figures. Figure 2a demonstrates the focal CNAs of interest, fair enough, but if the authors wish to discuss 1p, they should demonstrate this in at least a supplementary or extended data figure.

Thank you for this comment. While there were no obvious “peaks” of focal prognostic value in chromosome 1p, loss of the distal portion of this chromosome arm was more prognostic than that of the proximal end. We have now included a plot demonstrating focal regions of prognostic value on chromosome 1 and 22q (which likewise has no focal prognostic regions) as Extended Data Fig. 5a.

Other:

Extended Data Fig 6. - The authors may wish to modify the layout of the lower right quadrant of the figure. The two Kaplan-meier curves look a little scattershot at present, and the LASSO Cox regression coefficient boxes

are uneven. The network diagrams in the top left should also have a label near them saying LFFR and OS respectively.

Thank you for this suggestion. The layout of Extended Data Fig. 6 was necessitated by the differences in size of the plots in Extended Data Fig. 6b, c and a decision had to be made to maintain visual clarity. We have, however, implemented the suggestion to add labels for LFFR and OS to the network diagrams in Extended Data Fig. 6a.

Fig 3b - Is the top row (the plot of WHO grade 2, 3 meningiomas) a percentage, i.e. the y-axis ranges from 0.2 to 0.6% or is it from 20% to 60%? If it is the latter, then the label "% WHO grade 2 or 3" is inaccurate.

Thank you for bringing this error to our attention. We have corrected Fig. 4b and Extended Data Fig. 7b to denote the proportion of WHO grade 2 or 3 meningiomas rather than the percentage.

Reviewer #3

Thank you once again for your thoughtful comments on our manuscript.

Reviewer #4

The authors performed a major revision in response to my comments and those of other reviewers, which have improved several aspects of the manuscript. However, the "Results" section has now become exceedingly dense, making it difficult to read and interpret. A major revision is needed to improve clarity, with the following recommendations:

a. The "Results" section would benefit from subsections that clearly outline individual approaches and findings. Additionally, a study workflow diagram would improve readability and provide an overview of the discovery and test cohorts, methods (e.g., sliding window, incremental percentage), and statistical/machine learning approaches with key findings.

Thank you for this suggestion. We have now added subsection headings to the Results section of our revised manuscript. We have also added a study design and experimental workflow diagram as Fig. 1 of our revised manuscript.

b. Some methodological details currently included in the "Results" should be moved to the "Methods" section.

Thank you for this suggestion. We have ensured that the Methods section contains all methodological details. However, we have opted to also include brief details of relevant methods in the Results section for clarity in response to feedback from other reviewers.

Below are my other major and minor comments:

Major Comments:

1. The study promises two main contributions: (i) identifying chromosome-specific variable size thresholds, and (ii) analyzing co-occurrence patterns to better predict tumor control and survival. While the co-occurrence patterns and their predictive strength for clinical outcomes are intriguing, I have concerns about the introduction of chromosome-specific size thresholds, as detailed below.

a. Thresholds for CNV sizes have traditionally distinguished between chromosome arm-level events and focal events. However, thresholds in this study range from 0.01 to 0.97, potentially blurring the line between focal and arm-level CNV events. Key points for clarification include:

b. The study suggests an optimal threshold of 0.01 for chr1p amplification. Is it proposing that a 1% amplification of chr1p should be interpreted as an arm-level CNV event with prognostic value? This requires clarification. If not, the authors should clearly outline their intended interpretation.

Thank you for this comment. Amplification of chromosome 1p was uncommon, a weak predictor of meningioma recurrence in this cohort (maximum AUC to predict 5-year LFFR of 0.55), was not considered size-dependent by our criteria, and is not a well-described marker of meningioma behavior overall. Thus, the optimal threshold to call loss of chromosome 1p may be a result of noise in the data. The question of the value of small size thresholds is important. Prior research has demonstrated that loss of as little as 6% of chromosome 1p is

prognostic in meningioma (Maas et al. *Acta Neuropathologica* 2024), and that even loss of 1% could be useful in predicting risk of recurrence after resection. We have revised our Discussion section and provided the relevant passage below. The aim of this work was not to redefine the criteria for defining arm-level events. Instead, our finding that each chromosome arms exhibits a different optimal size threshold highlights the need to better understand the biology driving each arm's prognostic importance.

"More broadly, the optimal size thresholds for defining chromosome-specific CNAs ranged from 1-32% of individual chromosome arms, suggesting that for some chromosomes, such as loss of 1p (optimal LFFR threshold 23%) or loss of 12q (optimal LFFR threshold 18%), smaller size thresholds may not be ideal for identifying meningiomas with poor clinical outcomes. These thresholds may reflect the cutoff needed to separate broader amplification or deletion from sporadic CNAs that are not associated with a worse clinical prognosis. However, recent work has demonstrated that loss of as little as 6% of chromosome 1p is prognostic in meningioma, and that even loss of 1% could be useful in predicting risk of recurrence after resection, suggesting that any deletion of a critical chromosome arm may be sufficient to affect clinical outcomes."

c. Focal vs. Arm-Level Events: A 0.01 threshold for chr18p gain corresponds to ~160kb, similar to the size of focal events identified in 200kb bins. The authors should address this blurring of distinctions between focal and arm-level events.

Thank you for this suggestion. The identification of optimal size thresholds shown in Fig. 2 and Supplementary Tables 1 and 2 allowed for thresholds as small as 1% and as large as 99% of a chromosome arm, and thus could capture important focal or entire arm-level events depending on the arm of interest. In fact, our findings demonstrate that some CNAs are most prognostic as arm-level events, and others are most prognostic when looking at a focal region of loss. This underscores the importance of chromosome arm-specific considerations when building prognostic models for risk stratification. We have provided relevant text from our revised Discussion section below for ease of evaluation.

"These findings emphasize the importance of context when interpreting prognostic CNAs in meningiomas with respect to size, location, and co-occurrence with other CNAs. For example, interpretation of 10% loss of chromosome 1p in a proximal location, in the absence of any other CNAs, differs dramatically from 30% loss of distal chromosome 1p, co-occurring with loss of chromosomes 22q, 9p, or 14q, or co-occurring with gain of chromosomes 1q. Our data suggest the optimal prognostic threshold for loss of chromosome 1p is approximately 20%. One possible explanation for this threshold is that smaller losses tended to occur sporadically throughout chromosome 1p in our cohorts, whereas broader losses preferentially impacted the distal chromosome arm. Concordantly, our analysis of focal prognostic CNAs showed that AUC peaked in the distal region of chromosome 1p. Indeed, loss of distal chromosome 1p has long been shown to be among the most prognostic CNAs in meningioma. More broadly, the optimal size thresholds for defining chromosome-specific CNAs ranged from 1-32% of individual chromosome arms, suggesting that for some chromosomes, such as loss of 1p (optimal LFFR threshold 23%) or loss of 12q (optimal LFFR threshold 18%), smaller size thresholds may not be ideal for identifying meningiomas with poor clinical outcomes. These thresholds may reflect the cutoff needed to separate broader amplification or deletion from sporadic CNAs that are not associated with a worse clinical prognosis. However, recent work has demonstrated that loss of as little as 6% of chromosome 1p is prognostic in meningioma, and that even loss of 1% could be useful in predicting risk of recurrence after resection, suggesting that any deletion of a critical chromosome arm may be sufficient to affect outcomes."

2. The "integrated grade" approach with a 17% threshold across all chromosomes achieved an AUC of 0.79, while the variable thresholding method reached only 0.65 (with only two significant events). Similar results were observed for the "integrated score" at a 4% standard cutoff. Given these findings, it's unclear how variable thresholding offers an improvement over a uniform cutoff.

Thank you for this comment. Integrated score was trained on CNAs defined with a universal size threshold of 5% and performed best in our dataset at a similar size threshold (4%). This further supports the importance of size thresholds when building prognostic models, and that changing such thresholds can impact model performance (sometimes for the worse). Integrated grade was trained using a size threshold of 50%, and achieved optimal performance in our dataset at a uniform threshold of 17%. While these two thresholds are not as similar as in the case of Integrated score, they still differ from that of Integrated score and the arm-specific thresholds derived from our dataset, underscoring the importance of consideration of size thresholds when

developing and implementing clinical models based upon CNAs. This change in model performance is shown in Fig. 2b, and we have provided relevant text from our Discussion section below for ease of evaluation.

“Recent meningioma molecular classification systems have incorporated CNAs to enhance clinical risk stratification. These include Integrated grade, which is based on copy number losses of chromosomes 1p, 3p, 4p/q, 6p/q, 10p/q, 14q, 18p/q, and 19p/q, defined at a uniform threshold of 50% of each chromosome arm plus *CDKN2A* loss and mitotic count from histology. The second, Integrated score, is based on copy number losses of chromosomes 1p, 6q, and 14q, defined at a uniform threshold of 5% of each chromosome arm plus DNA methylation family and WHO histological grade. Our finding that each of these systems achieved optimal performance using CNA size thresholds close to the values used for training, and that performance degraded with variation of size thresholds, underscores the sensitivity of CNA models to size thresholds. At the level of model training, we find that size thresholds determine which CNAs form model inputs. Thus, the selection of CNA size thresholds represents a crucial branch point that influences the ultimate characteristics of resulting prognostic models. While Integrated grade and Integrated score incorporate overlapping CNAs, the optimal CNA size threshold and performance across thresholds varies significantly between these systems. These models each use uniform CNA size thresholds across chromosomes, but our results indicate that the optimal CNA size threshold can vary from chromosome to chromosome. By allowing for chromosome-specific variation in CNA size thresholds, we show that additional prognostic CNAs can be identified, and that a size-dependent CNA model may provide additional prognostic information in comparison to, or in combination with, Integrated grade and Integrated score.”

3. The authors incorporate *CDKN2A* loss as a separate variable in the integrated model, even though the size-dependent CNV data theoretically captures chromosome 9. This raises the question: does the model effectively capture *CDKN2A* loss, or is it necessary to include it as a separate variable?

Thank you for this comment. Homozygous *CDKN2A* deletion is a well-described marker of aggressive meningioma behavior, and as such is often treated as a standalone predictor of poor outcomes in prognostic models, including those incorporating CNAs such as Integrated grade (Driver et al. *Neuro Oncology* 2021). At the same time, hemizygous loss of *CDKN2A*, usually as part of a broader CNA involving 9p, has also been previously shown to be a marker of poorer outcomes, possibly to an intermediate degree (Khan et al. *Acta Neuropathologica* 2023). In both the previously published example and in our model, the addition of *CDKN2A* deletion as a separate variable improved risk stratification. Likely, the use of a single binary variable of 9p loss cannot fully capture the additive prognostic value of broader hemizygous loss of a 9p region encompassing *CDKN2A*, and focal homozygous loss of the *CDKN2A* locus, which can occur in addition to, or in the absence of, broader hemizygous loss in 9p. Additionally, our focal prognostic analyses identified loss of additional genes beyond *CDKN2A*, such as interferon-related genes located in 9p21, adjacent to the *CDKN2A* locus, as being important in meningioma outcomes. This locus has been implicated in immune evasion and a cold tumor immune microenvironment in other tumor contexts. We have provided the amended prose from the revised Methods section of our manuscript below for ease of evaluation.

“To build a model with improved prognostic performance, size-dependent CNAs were used as inputs into a gradient boosting model (XGBoost), either alone or in iterative combinations with WHO 2016 histological grade, *CDKN2A/B* loss, and mitoses per 10 high-power fields, using the *xgboost* (v1.7.7.1) package in R. These variables have been previously demonstrated to boost prognostic performance of risk stratification models incorporating meningioma CNAs”.

4. Focal CNVs are defined as prognostic if the AUC is greater than the median AUC of all CNVs for a given chromosome. However, the authors should compare these focal CNVs against the variable threshold approach to determine if they offer better prognostic performance.

Thank you for this suggestion. To assess the prognostic value of focal CNAs in our dataset, LASSO Cox models were built on gene expression data for the genes contained within focal regions of recurrent deletion (Extended Data Fig. 5b) and prognostic importance (Extended Data Fig. 5c). Performance of these models in external testing cohort 1 was still strong, with AUCs ranging from 0.61-0.85 for 5-year LFFR and 0.72-0.76 for 5-year OS. We have amended the Results section of our manuscript to mention these values and provided the revised text below for ease of evaluation.

“LASSO Cox regression of the expression of genes located on recurrent focally lost CNAs identified several as being prognostic for LFFR, such as *NUAK2*, *UBE2C*, *CTSC*, *NEK11*, and *HSPB7* (AUC 0.85 for 5-year LFFR in external test cohort 1), and several as being prognostic for OS, such as *FANCA*, *UBE2C*, *CACHD1*, and *SYNPO2* (AUC 0.76 for 5-year OS) (Extended Data Fig. 5c). Narrowing this analysis to genes located only in regions with focal prognostic value as determined by focal increases in AUC across 200 Kb bins identified *NUAK2* as important for LFFR (AUC 0.61), and *NUAK2*, *DSP*, and *PRKAR2B* as most important for OS (AUC 0.72) (Extended Data Fig. 5d), suggesting that these genes may drive the prognostic value of their respective CNAs in meningioma.”

5. The absence of chr1p-19q co-deletion in the size-dependent analysis is concerning, as this is a diagnostic marker for oligodendrogliomas.

Thank you for this observation. It is of course true that 1p/19q co-deletion is a defining molecular marker for oligodendrogliomas, and in fact did serve as a marker of better clinical outcomes among low-grade gliomas in TCGA data (Extended Data Fig. 10a). However, in this dataset, neither loss of 1p nor 19q met our criteria for size-dependence: (1) an AUC of at least 0.60 at any CNA threshold when using univariate Cox proportional hazards models to predict progression-free survival (PFS) or OS, (2) an AUC standard deviation of at least 0.01 across CNA size thresholds, and (3) prevalence in at least 2.5% of samples when defined using optimal size thresholds for each chromosomes. The relevant passage from our revised manuscript is provided below.

“Although low-grade glioma (LGG) can be stratified by co-deletion of chromosomes 1p and 19q to identify less aggressive oligodendrogliomas, neither of these CNAs met size-dependent criteria and this pair was excluded from co-occurrence models. Nevertheless, chromosome 1p/19q co-deletion was associated with better clinical outcomes, as expected (Extended Data Fig. 10a).”

6. On lines 334-336, the authors suggest that chromosome-specific thresholding is feasible for clinical application. This argument is subjective and should be reconsidered, as focal CNV analysis often leads to well-established clinical markers, such as *CDKN2A*.

Thank you for this comment. We agree that it is important to highlight differences in focal vs. arm-level copy number events in predicting clinical outcomes. Ideally, a combined approach would take into account both prognostic arm-level and focal CNAs, which is highlighted by the differences in optimal size thresholds across chromosome arms and diseases. We have revised the text addressing this topic in our Discussion section and have provided this prose below for ease of evaluation.

“In meningioma, we identified focal prognostic regions on chromosomes 2q, 3q, 5p, 5q, 8p, 9p, 10q, 12q, 13q, and 14q, including a focal peak at the *CDKN2A/B* and interferon gene locus on 9p. Of these, only 3q, 9p, 12q, and 14q were identified as prognostic and size-dependent when using percent-arm-altered thresholds rather than a sliding window approach, suggesting that in the cases of some chromosome arms, consideration should be given to both arm-level and focal CNAs of prognostic significance. However, a granular approach such as this may be too complex to be implemented and routinely used in clinical practice as compared to a percent-arm-altered threshold.”

7. The distinction between "sliding window" and "binning" approaches is unclear. The authors should provide further explanation regarding their implementation.

Thank you for this comment. We have used the terms “sliding window” and “binned CNA analysis” interchangeably. We have edited the text to be consistent in using the term “sliding window” throughout.

8. The authors should acknowledge the limitations of methylation data in identifying focal CNV events.

Thank you for this suggestion. We have added the following text to the Discussion section of the revised manuscript.

“We assess copy number changes in meningioma using DNA methylation profiling, as opposed to SNP array or whole-exome sequencing, and probe-based approaches are inherently limited by probe density and distribution compared to whole genome sequencing. However, CNAs identified using DNA methylation profiling have previously been demonstrated to have 99.12% concordance with those identified through exome sequencing approaches across multiple platforms (Choudhury et al. Nature Genetics 2022).”

9. The figures provided do not adequately illustrate the methods or results. The authors should improve the visual clarity and explanatory power of all figures.

To better clarify the methods and findings of this work, we have now included a study workflow diagram as Fig. 1 of our revised manuscript.

Minor Comments:

10. Line 205: The statement that CNA burden correlates with aggressive behavior is well-established and could be rephrased or removed.

Thank you for this suggestion. We have edited this line to the following: “Thus, CNA burden as a marker of aneuploidy, which has been demonstrated to be important in other cancers, and enrichment of key CNAs may be a useful biomarker of high-risk meningiomas.”

11. Figure 2b: This figure is visually unclear (error bars and dots are not visible) and methodologically ambiguous. Are all genes within the segments combined?

Thank you for this suggestion. We have clarified in the legend for revised Fig. 3b (formerly Fig. 2b) that gene expression data were derived from sample-matched RNA sequencing of all genes mapping to focal regions identified by cross-referencing chromosomal position with the Ensembl database. Furthermore, we have now included asterisks in Fig. 3a (formerly Fig. 3a) to denote focal copy number and prognostic regions from which these genes were identified, which are also provided in Supplementary Tables 4 and 5. We provide an enlarged version of Fig. 3b (right) for visualization of the means and error bars.

12. Figure 3b: The co-occurrence plot's scale is confusing. With a cohort size of 365, it is unclear why the scale reaches 3000. This plot requires further clarification.

We have edited the legend of Figure 4b to clarify that the scale of the co-occurrence bar plots reflects the total number of co-occurrent pairs including each of the CNAs, across all samples containing that CNA. Thus, the total number may greatly exceed the number of samples.

Reviewer #5

The authors have thoroughly and diligently addressed all the concerns raised in the previous round of review, specifically by providing a clearer comparison and integration with existing prognostic models, testing generalization on two external datasets, and outlining plans for further validation before clinical application. The revised manuscript demonstrates substantial improvements, all of which contribute to the overall quality and impact of the work.

Thank you very much for your positive consideration of our revised manuscript.

Reviewer #4

I commend the authors for thoroughly addressing all my questions and concerns. The manuscript has been significantly improved, with better articulation of its scope, findings, and limitations. Additionally, the structural changes have enhanced its overall readability.

Thank you very much for your positive consideration of our revised manuscript.

My few minor recommendations are as follows:

1. Line 583-584: "Focal regions of loss were selected by manual inspection of regions along the chromosome arm with a notably higher proportion of samples.": The authors claim that "manual" inspection is an established method citing CDKN2A studies. This is quite different as those studies were identifying a single prognostic marker. However, here the authors claim a group of focal chromosomal events with clinical implications, also proposes that this method is potentially applicable to other cancer types. To me, this is a limitation of the method. The authors should emphasize this better in the result and/or discussion section.

Thank you for this comment. We have amended the Discussion section of the revised manuscript to mention this limitation and provided the revised text below for ease of evaluation.

"While manual inspection of focal CNAs is the preferred method of identifying deletion of the CDKN2A/B locus, this was previously only done in the context of identifying a single prognostic marker, rather than a generalized approach across chromosomes."

2. In Methods Section, RNA-sequencing experimental procedures and analysis methods were summarized under "Focal genomic and ontology analyses" section. They should be detailed under the appropriate title, so that readers with specific interests/questions can locate the appropriate method details.

Thank you for this suggestion. We have added a new subsection heading within the Methods section for meningioma gene expression analysis.